# ONLINE CONFORMAL PREDICTION WITH ADVERSARIAL SEMI-BANDIT FEEDBACK VIA REGRET MINIMIZATION

**Junyoung Yang**[*]
CSE
POSTECH

**Kyungmin Kim**[*]
GSAI
POSTECH

**Sangdon Park**
GSAI & CSE
POSTECH

## ABSTRACT

Uncertainty quantification is crucial in safety-critical systems, where decisions must be made under uncertainty. In particular, we consider the problem of online uncertainty quantification, where data points arrive sequentially. Online conformal prediction is a principled online uncertainty quantification method that dynamically constructs a prediction set at each time step. While existing methods for online conformal prediction provide long-run coverage guarantees without any distributional assumptions, they typically assume a *full feedback* setting in which the true label is always observed. In this paper, we propose a novel learning method for online conformal prediction with *partial feedback* from an adaptive adversary—a more challenging setup where the true label is revealed only when it lies inside the constructed prediction set. Specifically, we formulate online conformal prediction as an adversarial bandit problem by treating each candidate prediction set as an arm. Building on an existing algorithm for adversarial bandits, our method achieves a long-run coverage guarantee by explicitly establishing its connection to the regret of the learner. Finally, we empirically demonstrate the effectiveness of our method in both independent and identically distributed (i.i.d.) and non-i.i.d. settings, showing that it successfully controls the miscoverage rate while maintaining a reasonable size of the prediction set.

## 1 INTRODUCTION

Uncertainty quantification is essential in safety-critical domains such as autonomous driving (Lindemann et al., 2023), healthcare (Lin et al., 2022), and finance (Park & Cho, 2025), where uncertainty-aware decision making is required. Unlike point prediction methods that return the most likely outcome, conformal prediction (Vovk et al., 2005) is a promising uncertainty quantification method that constructs a *conformal set* for a given input, a set of outcomes that is guaranteed to contain the true label with a user-specified probability. We refer to the guarantee as a *coverage guarantee*. Here, the size of the conformal set quantifies the uncertainty in terms of making a prediction.

Moreover, the coverage guarantee is model-agnostic in the sense that the guarantee holds irrespective of the choice of the prediction model. Exchangeability assumption on the data generating process (Vovk et al., 2005) is the only requirement for the guarantee, where a typical independent and identically distributed (i.i.d.) scenario is the case that satisfies such assumption. Specifically, under the exchangeability assumption, the coverage guarantee of the conformal set constructed from training samples holds for an unseen test sample (Vovk, 2013). Therefore, since the coverage guarantee holds for any prediction models, conformal prediction has been applied to complex and large-scale models such as large language models (Mohri & Hashimoto, 2024; Cherian et al., 2024; Lee et al., 2024).

However, the exchangeability assumption is easily violated under scenarios such as distribution shift, where the training and test distributions differ. A number of conformal prediction methods have been proposed to provide coverage guarantees under such settings (Tibshirani et al., 2019; Podkopaev & Ramdas, 2021; Park et al., 2022; Gendler et al., 2022; Si et al., 2024). In contrast to the aforementioned

---

[*]Equal contribution

batch conformal prediction methods which require a batch of samples for training, methods for online conformal prediction are proposed to tackle online uncertainty quantification problems, where data points arrive sequentially (Gibbs & Candès, 2021; Bastani et al., 2022; Angelopoulos et al., 2023; 2024). Even in adversarial settings, where no distributional assumptions are made on the data stream or on the functional form of the scoring functions, these methods provide a long-run coverage guarantee such that the empirical coverage reaches the target level after sufficiently many time steps.

Meanwhile, existing methods for online conformal prediction typically assume a *full feedback* scenario, where the true label is revealed every time step. Indeed, these methods are tailored to the full feedback setting, since they require a scoring function evaluated on the true label either for its quantile estimation or for the evaluation of the miscoverage loss over multiple conformal set candidates. Recently, Ge et al. (2025) proposed a method for online conformal prediction with *partial feedback*, specifically feedback referred to as *semi-bandit feedback*, where the true label is revealed only when it lies within the chosen conformal set. While it is a more challenging learning setting compared to the full feedback scenario, their coverage guarantee holds only under i.i.d. data streams. Figure 1 provides an illustration of different feedback settings in online conformal prediction.

In this paper, we present the first study of online conformal prediction with adversarial partial feedback. Specifically, by discretizing the continuous hypothesis space of thresholds that parameterize a conformal set, and then treating each candidate conformal set as an arm, we formulate the problem as a multi-armed adversarial bandit problem. A bandit problem is a sequential game between a learner and an environment. In each round, the learner chooses an arm, and the environment provides feedback on the chosen arm. In particular, the adversarial bandit problem removes almost all assumptions about how the environment provides feedback, where the environment is often called the adversary accordingly. The performance of the learner is evaluated based on the regret, which typically quantifies how well the learner performs with respect to the best arm in hindsight (Bubeck et al., 2012; Lattimore & Szepesvári, 2020). There is a rich literature on adversarial bandit problems, devising algorithms with sublinear regret. `EXP3.P` algorithm Auer et al. (2002) is one of the algorithms that provides a high-probability sublinear regret even under the adaptive adversary, an adversary that can generate feedback based on the previous history.

Building on the `EXP3.P` algorithm, we propose a novel algorithm for online conformal prediction with adversarial partial feedback, in which we consider a semi-bandit feedback scenario, similar to Ge et al. (2025). Specifically, by devising a loss function tailored to conformal prediction, we explicitly establish a connection between the regret of the learner and a long-run coverage (Lemma 1), which in turn provides a long-run coverage guarantee of our algorithm. We further improve the performance in terms of the speed approaching the target coverage, by fully exploiting the monotonicity property of the miscoverage loss with respect to the threshold parameterizing a conformal set. Specifically, it enables partial inference of feedback from candidate conformal sets that are not chosen, even when the true label is unavailable. We empirically demonstrate the efficacy of our method on both classification and regression tasks, conducting experiments in i.i.d. and non-i.i.d. settings for each task. In particular, we show that our method approaches long-run coverage while maintaining a moderate average conformal set size, achieving performance comparable to Bastani et al. (2022), an online conformal prediction method with adversarial full feedback.

## 2 RELATED WORK

### 2.1 ONLINE CONFORMAL PREDICTION

Gibbs & Candès (2021) first proposed an online conformal prediction method for arbitrary data streams. Based on online gradient descent, their method provides a long-run coverage guarantee over arbitrary sequences. While the method relies only on a single step size parameter, the optimal parameter requires knowledge of the degree of distribution shift, which is an unrealistic assumption. The same authors have resolved the issue by aggregating results from multiple experts, running in parallel with different step sizes, making the method adaptive to the type of distribution shift in a data-driven manner Gibbs & Candès (2024). While providing a biased result in terms of the long-run coverage, they provide a local coverage guarantee over all time intervals of a given width, under mild assumptions on the smoothness of the distribution of the scoring function and its quantile estimates. Building on the strongly adaptive online learning method, Bhatnagar et al. (2023) further improved the method by providing a simultaneous coverage guarantee over all local intervals of arbitrary

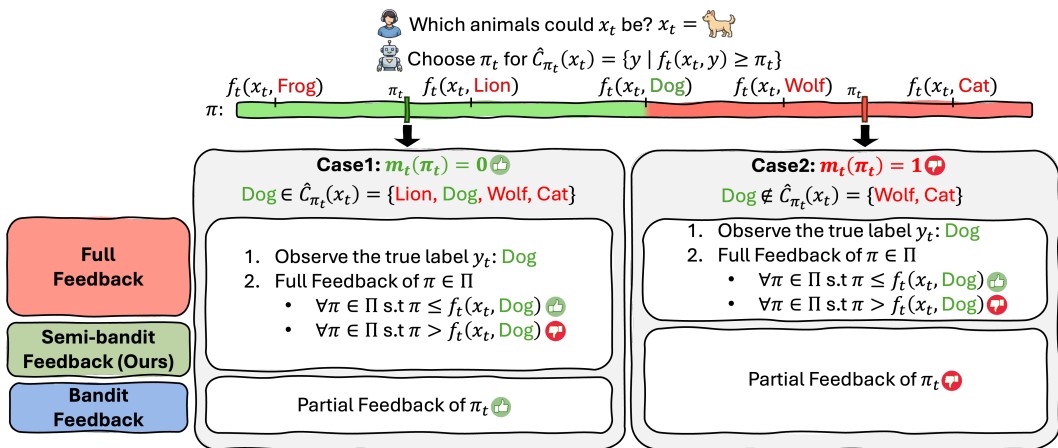

Figure 1: Online conformal prediction under different feedback scenarios characterized by the amount of feedback received at each time step

window size. Unlike Gibbs & Candès (2024), they considered a dynamic set of experts, where each expert is active only for a specific period of time. Inspired by control theory, Angelopoulos et al. (2023) extended existing online gradient descent-based methods by incorporating online gradient descent steps, which they refer to as quantile tracking, as one of the components for the online quantile update.

Besides, there have been works to provide stronger theoretical guarantees. Bastani et al. (2022) proposed a method with a threshold-calibrated multivalid coverage guarantee, a group- and threshold-conditional coverage guarantee where a set of groups can be arbitrarily defined. Angelopoulos et al. (2024) proposed a simple online gradient-descent method that has simultaneous guarantees both on the adversarial and i.i.d. settings. Recently, Zhang et al. (2025) devised an online conformal prediction algorithm, providing both privacy and coverage guarantees under arbitrary data streams.

In this paper, we also consider online conformal prediction under an arbitrary data stream. Specifically, we consider an adaptive adversary that can generate data based on the learner's past actions.

## 2.2 Online Conformal Prediction with Partial Feedback

Existing methods for online conformal prediction with adversarial feedback typically assume the full feedback setting, where the true label is revealed at every time step. One exceptional case is Angelopoulos et al. (2024), where the algorithm itself only requires the feedback on whether a chosen conformal set contains the true label. However, the authors are basically considering a full feedback scenario, and some of their theoretical results assume a problem setup where the scoring function is trained online, using the labeled data pairs from previous time steps.

On the other hand, there have been few papers addressing online conformal prediction with partial feedback, a scenario where access to the true label is limited. Wang & Qiao (2024) considered a bandit feedback scenario, where the true label is observed only when the predicted label corresponds to the true label. Recently, Ge et al. (2025) proposed a method under a semi-bandit feedback scheme, a less rigid partial feedback scenario where the true label is revealed as the true label lies within a chosen conformal set. Although partial feedback is inherently more challenging than full feedback, prior works still rely on the i.i.d. data-generating assumption, which restricts their applicability to real-world, non-i.i.d. data streams.

As such, we consider an online conformal prediction with adversarial partial feedback, where data streams deviate from the i.i.d. process and at the same time the true label is difficult to obtain.

## 3 Online Conformal Prediction with Adversarial Feedback

We consider online conformal prediction with adversarial partial feedback. Let $\mathcal{X}$ be a set of examples and $\mathcal{Y}$ be a set of labels. At each time step $t \in [T]$, a learner chooses a conformal set $\hat{C}_{\pi_t} : \mathcal{X} \to 2^{\mathcal{Y}}$,

which is parameterized by the threshold parameter $\pi_t \in [0, 1]$ as follows:

$$\hat{C}_{\pi_t}(x) := \{\tilde{y} \in \mathcal{Y} \mid f_t(x, \tilde{y}) \geq \pi_t\}. \tag{1}$$

Here, $f_t : \mathcal{X} \times \mathcal{Y} \rightarrow [0, 1]$ is a scoring function that measures the conformity of a label for a given input. Note that the functional form of $f_t(\cdot, \cdot)$ may evolve over time.

Specifically, we consider the following learning protocol: an example $(x_t, y_t) \in \mathcal{X} \times \mathcal{Y}$ is chosen by an adversary, where only the input $x_t$ is revealed to a learner. Here, we assume that the adversary can be adaptive, in the sense that it may select a sample based on the learner's previous decisions. Once $x_t$ is given, the learner outputs a conformal set $\hat{C}_{\pi_t}(x_t)$. The threshold parameter $\pi_t$ is selected according to the learner's current stochastic strategy, which is updated online based on past interactions with the adversary. Then, the learner receives a feedback from the adversary on whether $\hat{C}_{\pi_t}(x_t)$ contains the true label $y_t$, which we denote as $m_t(\pi_t) := \mathbb{1}(y_t \notin \hat{C}_{\pi_t}(x_t))$. We refer to the term as *miscoverage* henceforth.

Here, we consider a *semi-bandit feedback* scenario (Ge et al., 2025), one of the partial feedback settings where a true label $y_t$ is additionally revealed when $m_t(\pi_t) = 0$. In other words, each instance $x_t$ has an associated true label $y_t$, which exists but is not labeled at the time the adversary chooses the sample. Having an access to $y_t$ enables the learner to evaluate the miscoverage $m_t(\pi)$ for all $\pi \in [0, 1]$, since the scoring function $f_t(x_t, y_t)$ of the true label—a quantity sufficient to evaluate the miscoverage of a threshold-parameterized conformal set—can be computed. Although we have coined the term partial feedback, in contrast to full feedback, to encompass both bandit and semi-bandit feedback settings, we will use it to refer exclusively to the semi-bandit feedback scenario in the following sections for simplicity.

Under such online conformal prediction problem with adversarial partial feedback, our goal is to design a learner that provides a long-run coverage guarantee by controlling the miscoverage rate defined as follows:

$$\mathbf{MC}(T) := \frac{1}{T} \sum_{t=1}^{T} m_t(\pi_t), \tag{2}$$

where $T$ is the time horizon. Specifically, given a target miscoverage level $\alpha \in (0, 0.5)$, we aim to upper bound the miscoverage rate as $\mathbf{MC}(T) \leq \alpha + \varepsilon(T)$ such that $\varepsilon(T) \rightarrow 0$ as $T \rightarrow \infty$. Additionally, we aim to minimize the average size of constructed conformal sets at the same time, which is defined as

$$\mathbf{Ineff}(T) := \frac{1}{T} \sum_{t=1}^{T} |\hat{C}_{\pi_t}(x_t)|, \tag{3}$$

since a learner that always returns the vacuous conformal set $\mathcal{Y}$ by choosing $\pi_t = 0$ for all $t \in [T]$ trivially achieves the target miscoverage level.

## 4 ONLINE CONFORMAL PREDICTION AS ADVERSARIAL BANDIT PROBLEM

We formulate the online conformal prediction problem with adversarial partial feedback as a multi-armed adversarial bandit problem, by treating each candidate conformal set as an arm (Section 4.1). Defining a finely discretized subset $\Pi$ of the continuous hypothesis space $[0, 1]$ as an action space, we improve the `EXP3.P` algorithm (Auer et al., 2002), which is an algorithm that provides a high-probability sublinear regret under adversarial bandit environments, for conformal prediction.

To this end, we first design a loss function tailored to conformal prediction (Section 4.2), which in turn provides an explicit learner-agnostic relationship between a regret from a learner and its miscoverage rate $\mathbf{MC}(T)$ (Section 4.3). This relationship ensures the long-run coverage to achieve the target level $1 - \alpha$, for any learner that achieves a sublinear regret. However, directly applying an existing learner, *e.g.,* `EXP3.P`, does not make full use of the available information under the semi-bandit feedback setting. Therefore, we further improve the algorithm by fully exploiting such additional information and the monotonicity property of a threshold-parameterized conformal set with respect to the miscoverage (Section 4.4).

## 4.1 PROBLEM RE-FORMULATION

Discretization of the hypothesis space $[0, 1]$ into $\Pi$ allows us to reformulate online conformal prediction with adversarial feedback (Section 3) as a multi-armed adversarial bandit problem. Each arm corresponds to a threshold parameter $\pi \in \Pi$ that specifies a threshold-parameterized conformal set $\hat{C}_\pi$. Specifically, for each time step $t$, a learner chooses a threshold parameter $\pi_t$ from $K$ possible candidates, where $K := |\Pi|$ is defined as the discretization level of the original hypothesis space $[0, 1]$. As we consider a partial feedback scenario, we observe the true label $y_t \in \mathcal{Y}$ in addition to the miscoverage feedback $m_t(\pi_t)$ when $m_t(\pi_t) = 0$. This reformulation enables us to leverage existing learners under multi-armed adversarial bandit environments, where each learner's performance is measured by its regret relative to the best arm in hindsight, defined as

$$\mathbf{Reg}(T) := \sum_{t=1}^{T} \ell_t(\pi_t) - \min_{\pi \in \Pi} \sum_{t=1}^{T} \ell_t(\pi). \tag{4}$$

Here, $\ell_t := (\ell_t(\pi))_{\pi \in \Pi} \in [\ell_{\min}, \ell_{\max}]^K$ is a loss vector chosen by an adversary at time $t$, with each entry lying in the bounded interval $[\ell_{\min}, \ell_{\max}]$. Note that the online conformal prediction problem with adversarial feedback (Gibbs & Candès, 2021; 2024; Angelopoulos et al., 2024) assumes that the environment controls the choice of $(x_t, y_t)$ instead of the loss vector $\ell_t$, where $m_t(\pi)$ is fully determined by

Table 1: Comparison of online conformal prediction with adversarial feedback and typical multi-armed adversarial bandit problems

| **Problem** | Online Conformal Prediction with Adversarial Feedback | Multi-armed Adversarial Bandit |
|---|---|---|
| **Adversary Controls** | Sample: $(x_t, y_t)$, where $y_t$ is unlabeled | Loss Vector: $(\ell_t(\pi))_{\pi \in \Pi}$ |
| **Feedback** | Miscoverage: $m_t(\pi_t)$ True Label: $y_t$ if $m_t(\pi_t) = 0$ | Loss: $\ell_t(\pi_t)$ |
| **Loss Defined by** | Learner and Adversary | Adversary |
| **Objective** | Miscoverage Rate: $\mathbf{MC}(T)$ Inefficiency: $\mathbf{Ineff}(T)$ | Regret: $\mathbf{Reg}(T)$ |

$(x_t, y_t)$ for all $\pi \in \Pi$. Unlike the typical adversarial bandit problem, the learner has control over the design of the loss function $\ell_t : \Pi \mapsto [\ell_{\min}, \ell_{\max}]$. Nevertheless, while the adversary does not directly control the loss sequence, the adaptive adversary we consider can choose $(x_t, y_t)$ based on the previous interaction with the learner. See Table 1 for details on the problem reformulation and the following section for details on the loss design.

## 4.2 DESIGN OF THE LOSS FUNCTION

In online conformal prediction with adversarial feedback, the only feedback we observe at each time step $t$ is on the miscoverage $m_t(\pi_t)$ of the chosen threshold $\pi_t$. However, additional feedback on the size of a conformal set is necessary to keep the constructed set compact and informative, since a lazy learner that always returns the label space $\mathcal{Y}$ as a conformal set by letting $\pi_t = 0$ for all $t \in [T]$ achieves $\mathbf{MC}(T) = 0$. Therefore, we define a loss function composed of a miscoverage term and an inefficiency term, capturing the miscoverage and the size of a conformal set, respectively, as follows:

$$\ell_t(\pi; \alpha) := d_t(\pi; \alpha) + a_t(\pi), \tag{5}$$

where $\alpha \in (0, 0.5)$ is a target miscoverage level. The *miscoverage term* $d_t(\pi; \alpha)$ is defined in terms of the miscoverage feedback $m_t(\pi)$ as

$$\begin{aligned} d_t(\pi; \alpha) &:= m_t(\pi) + \mathbb{1}(m_t(\pi) = 0)g(\alpha) + \mathbb{1}(m_t(\pi) = 1)h(\alpha) \\ &= m_t(\pi) + \mathbb{1}(m_t(\pi) = 0)(\alpha + \alpha(1 - \alpha)) + \mathbb{1}(m_t(\pi) = 1)(-\alpha(1 - \alpha)). \end{aligned} \tag{6}$$

Here, $g(\alpha) > 0$ and $h(\alpha) < 0$ jointly control the miscoverage term in an $\alpha$-adaptive manner. Taking the miscoverage control as a primary objective, the miscoverage term satisfies $d_t(\pi_1; \alpha) \leq d_t(\pi_2; \alpha)$ for all $\pi_1 \in \Pi_t^*$ and $\pi_2 \in (\Pi_t^*)^c$, where $\Pi_t^* := \{\pi \in \Pi : m_t(\pi) = 0\}$. In particular, $g(\alpha)$ and $h(\alpha)$ are designed such that $d_t(\pi_2; \alpha) - d_t(\pi_1; \alpha) = (1 + h(\alpha)) - g(\alpha)$ decreases to 0 as $\alpha$ increases, prioritizing the miscoverage for smaller $\alpha$. We henceforth write $d_t(\pi)$ for simplicity, indicating the dependence on $\alpha$ only when needed. Additionally, we consider the *inefficiency term* $a_t(\pi)$ of an arbitrary functional form, as long as it penalizes large conformal sets.

### 4.3 From Regret Bounds to Coverage Guarantees

By reformulating online conformal prediction with adversarial feedback as a multi-armed adversarial bandit problem, we reduce our problem to minimizing regret with respect to the loss defined in Eq. 5. However, while both the miscoverage and the size of a conformal set are incorporated into the loss design (Section 4.2), minimizing regret (Eq. 4) does not necessarily guarantee control of the miscoverage rate (Eq. 2) at the target level $\alpha$, which is our primary objective.

Given a target miscoverage level $\alpha \in (0, 0.5)$, we therefore specify the condition that yields a learner-agnostic guarantee that bounds the deviation of the miscoverage rate $\mathbf{MC}(T)$ from $\alpha$ in terms of the per-round regret $\frac{1}{T}\mathbf{Reg}(T)$ (Lemma 1), inspired by prior work that derives analogous regret-based bounds for false discovery rate control in selective generation (Lee et al., 2025). The proof is deferred to Appendix B.

**Lemma 1.** *For any $T \in \mathbb{N}, \alpha \in (0, 0.5)$, under the design choice that $a_t(\pi)$ is of order $o(T)^{-1}$ for all $\pi \in \Pi$, any learner incurring regret $\mathbf{Reg}(T)$ with respect to the loss defined in Eq. 5 satisfies*

$$\mathbf{MC}(T) - \alpha \le \frac{1}{T}\mathbf{Reg}(T) + C_{\mathbf{MC}}(T),$$

*where $C_{\mathbf{MC}}(T)$ is a constant term of order $o(T)^{-1}$.*

Specifically, $C_{\mathbf{MC}}(T) := \frac{Ta_t(0) - \sum_{t=1}^{T} a_t(\pi_t)}{T} + \frac{\alpha C_{\mathrm{gap}}(T)\sum_{t=1}^{T}\mathbb{1}(m_t(\pi_t)=1)}{T}o(T)^{-1}$, where $C_{\mathrm{gap}}(T)$ is a constant that characterizes the tightness of the learner that satisfies the relationship $C_{\mathrm{gap}}(T)o(T)^{-1} = \frac{1-\alpha}{\alpha} - \frac{\sum_{t=1}^{T}\mathbb{1}(m_t(\pi_t)=0)}{\sum_{t=1}^{T}\mathbb{1}(m_t(\pi_t)=1)}$. Note that the empirical analysis reveals that $C_{\mathrm{gap}}(T)$ decreases as $T$ increases (Appendix L). Roughly speaking, any learner with sublinear regret achieves long-run coverage at least $1 - \alpha$. For example, we can employ the EXP3.P algorithm (Auer et al., 2002) which provides a high-probability sublinear regret even under an adaptive adversary.

### 4.4 Tailoring EXP3.P for Online Conformal Prediction

We propose modifications to the EXP3.P algorithm tailored to online conformal prediction with adversarial partial feedback. First, we consider the following inefficiency term $a_t(\pi; \alpha, c) \in [0, 1]$ for all baselines:

$$a_t(\pi; \alpha, c) :=$$
$$\mathbb{1}(m_t(\pi) = 0)\left(-c\alpha\left(1 + \frac{\pi^2}{1-\alpha}\right)o(T)^{-1}\right) + \mathbb{1}(m_t(\pi) = 1)\left(-\frac{(c\alpha)o(T)^{-1}}{1 + \alpha(1-2\alpha)\pi}\right), \quad (7)$$

where $c > 0$. Here, $c$ balances the trade-off between miscoverage and inefficiency terms. We henceforth write $a_t(\pi)$ for simplicity, indicating the dependence on $\alpha$ and $c$ only when needed. See Appendix H for details on the choice of $c$. For $\pi \in \Pi_t^*$, $a_t(\pi)$ is a monotonically increasing function in the size of the corresponding conformal set $|\hat{C}_\pi(x_t)|$, since the set size decreases as $\pi$ increases (Eq. 1). This encourages smaller conformal sets as long as $m_t(\pi) = 0$. In contrast, for $\pi \in (\Pi_t^*)^c$, $a_t(\pi)$ decreases as $|\hat{C}_\pi(x_t)|$ increases. When $m_t(\pi) = 1$, the primary role of $a_t(\pi)$ is to penalize the miscoverage, while the set size is of secondary importance. The design leverages the monotonic structure of single-thresholded conformal sets (Eq. 1), in which the miscoverage term $m_t(\pi)$ monotonically decreases as $|\hat{C}_\pi(x_t)|$ increases. Since EXP3.P-based algorithms are formulated in terms of gains, we additionally introduce a normalized gain term $g_t(\pi) \in [0, 1]$, defined as

$$g_t(\pi) := \frac{\ell_{\max} - \ell_t(\pi)}{\ell_{\max} - \ell_{\min}}. \quad (8)$$

**OCP-Bandit.** It is a direct modification of the EXP3.P algorithm (Algorithm 2) for online conformal prediction with adversarial partial feedback (Algorithm 3). OCP-Bandit provides a sublinear regret with probability at least $1 - \delta$, where $\delta \in (0, 1)$ is a confidence parameter (Theorem 3). Following EXP3.P, OCP-Bandit adopts a biased gain estimator $\tilde{g}_t(\pi|\{\pi_t\})$ for all $\pi \in \Pi$ to update the learner's strategy $p_t \in [0, 1]^K$ as:

$$\tilde{g}_t(\pi|\{\pi_t\}) := \mathbb{1}(\pi = \pi_t)\frac{g_t(\pi) + \beta}{p_t(\pi)} + \mathbb{1}(\pi \ne \pi_t)\frac{\beta}{p_t(\pi)}, \quad (9)$$

where $p_t$ lies in the probability simplex defined over the action space $\Pi$. It is an importance-weighted estimator of the corresponding normalized gain $g_t(\pi)$, augmented with an additional exploration term $\beta > 0$, to ensure high-probability sublinear regret against an adaptive adversary. Note that the normalized gain can be computed only for the chosen arm $\pi_t$.

However, unlike the standard bandit feedback setting assumed by `EXP3.P`, we consider a partial feedback scenario in which the true label $y_t$ is additionally observed when $m_t(\pi_t) = 0$. In this case, the scoring function $f_t(x_t, y_t)$ of the true label can be computed, which allows us to evaluate $m_t(\pi)$ for all $\pi \in \Pi$. Moreover, exploiting the monotonicity of $m_t(\pi)$ with respect to $\pi$, we can partially evaluate miscoverage terms even when $m_t(\pi_t) = 1$. This additional information, which is ignored by `OCP-Bandit`, can be exploited to achieve the target coverage level with fewer time steps.

**OCP-Unlock+.** We propose a novel algorithm that fully exploits the additional information available when $m_t(\pi_t) = 0$ and uses the monotonicity of the miscoverage term $m_t(\pi)$ with respect to the threshold $\pi \in \Pi$ to obtain additional feedback when $m_t(\pi_t) = 1$ (Algorithm 1). Specifically, if $m_t(\pi_t) = 1$, then $m_t(\pi) = 1$ for all $\pi \in \Pi$ such that $\pi \geq \pi_t$, since $m_t(\pi)$ in non-decreasing as $\pi$ increases. Therefore, we can evaluate $m_t(\pi)$ for the subset $\Pi_t(\pi_t) \subset \Pi$, which we refer to as *unlocking set*, defined as follows:

$$\Pi_t(\pi_t) := \begin{cases} \Pi & \text{if } m_t(\pi_t) = 0 \\ \{\pi \in \Pi : \pi \geq \pi_t\} & \text{if } m_t(\pi_t) = 1 \end{cases}. \tag{10}$$

As such, we propose a new biased gain estimator $\tilde{g}_t(\pi | \Pi_t(\pi_t))$ that is designed in accordance with the amount of additional feedback available, as characterized by the unlocking set $\Pi_t(\pi_t)$. In particular, since the degree of additional feedback varies with $m_t(\pi_t)$, we design separate estimators for each case where $m_t(\pi_t) = 0$ and $m_t(\pi_t) = 1$, *i.e.,*

$$\tilde{g}_t\left(\pi \mid \Pi_t(\pi_t)\right) := \underbrace{\mathbb{1}(m_t(\pi_t) = 0) \times (A)}_{\text{full unlocking}} + \underbrace{\mathbb{1}(m_t(\pi_t) = 1) \times (B)}_{\text{partial unlocking}}. \tag{11}$$

In what follows, we provide a high-level description of our estimator and refer the reader to Appendix C for further details.

First, when $m_t(\pi_t) = 0$, we design the biased gain estimator as follows:

$$(A) := \mathbb{1}(\pi \in \Pi_t^*)\left\{\frac{g_t(\pi)}{\sum_{\tilde{\pi} \in \Pi_t^*} p_t(\tilde{\pi})} + \left(1 + \frac{1}{\sum_{\tilde{\pi} \in \Pi_t^*} p_t(\tilde{\pi})}\right)\beta\right\} + \mathbb{1}(\pi \notin \Pi_t^*)\left\{g_t(\pi) + \frac{\beta}{\sum_{\tilde{\pi} \leq \pi} p_t(\tilde{\pi})}\right\},$$

where we can compute the normalized gain $g_t(\pi)$ for all $\pi \in \Pi$ due to semi-bandit feedback. The estimator is designed to have different importance weights for normalized gains of thresholds from two groups, $\Pi_t^*$ and its complement, where larger weight is assigned to those from the former. This is consistent with our design rationale for the loss function (Sec. 4.2 and Eq. 7), which prioritizes miscoverage control. Also, for $\pi \notin \Pi_t^*$, we further strengthen this miscoverage-first rationale by placing less weight on the exploration term $\beta$ for larger conformal sets via the denominator $\sum_{\tilde{\pi} \leq \pi} p_t(\tilde{\pi})$.

On the other hand, when $m_t(\pi_t) = 1$, feedback can be evaluated only on the subset $\Pi_t(\pi_t) = \{\pi \in \Pi : \pi \geq \pi_t\}$ of $(\Pi_t^*)^c$. In this case, we define the estimator as follows:

$$(B) := \mathbb{1}(\pi \notin \Pi_t(\pi_t))\left\{\tilde{g}_t(\pi) + \left(1 + \frac{1}{p_t(\pi)}\right)\beta\right\} + \mathbb{1}(\pi \in \Pi_t(\pi_t))\left\{g_t(\pi) + \frac{\beta}{\sum_{\tilde{\pi} \leq \pi} p_t(\tilde{\pi})}\right\}.$$

Here, the term $\tilde{g}_t(\pi)$ is introduced for all $\pi \notin \Pi_t(\pi_t)$, since $g_t(\pi)$ cannot be computed, as follows:

$$\tilde{g}_t(\pi) = \frac{\ell_{\max} - \tilde{\ell}_t(\pi)}{\ell_{\max} - \ell_{\min}}, \tag{12}$$

with $\tilde{\ell}_t(\pi) := 1 - \alpha(1 - \alpha) - \frac{c\alpha}{1 + \alpha(1 - 2\alpha)\pi}o(T)^{-1}$. We refer to $\tilde{g}_t(\pi)$ as a *pseudo-gain*, since it serves as an approximation of the true normalized gain $g_t(\pi)$. Unlike the $m_t(\pi_t) = 0$ case where $\Pi_t^*$ can be explicitly characterized, $\Pi_t(\pi_t)^c$ is decomposed as $\Pi_t(\pi_t)^c = \{\Pi_t(\pi_t)^c \cap \Pi_t^*\} \cup \{\Pi_t(\pi_t)^c \cap (\Pi_t^*)^c\} = \Pi_t^* \cup \{\Pi_t(\pi_t)^c \cap (\Pi_t^*)^c\}$ when $m_t(\pi_t) = 1$. Since $\Pi_t^*$ cannot be correctly

---

**Algorithm 1** `OCP-Unlock+`

---

1: **procedure** OCP-UNLOCK+$(\Pi, T, \eta, \gamma, \beta, \alpha, c, (f_t)_{t=1}^{T})$
2:      Initialize cumulative estimated gain: $\tilde{G}_0(\pi) \leftarrow 0$ for all $\pi \in \Pi$
3:      **for** $t \in \{1, \ldots, T\}$ **do**
4:          Update strategy: $p_t(\pi) \leftarrow (1-\gamma)\frac{\exp(\eta \tilde{G}_{t-1}(\pi))}{\sum_{\tilde{\pi} \in \Pi} \exp(\eta \tilde{G}_{t-1}(\tilde{\pi}))} + \gamma \frac{1}{K}$, where $K = |\Pi|$
5:          Sample arm: $\pi_t \sim p_t$
6:          Receive $m_t(\pi_t) \leftarrow \mathbb{1}\left(y_t \notin \hat{C}_{\pi_t}(x_t)\right)$, where $\hat{C}_{\pi_t}(x_t)$ is defined as Eq. 1
7:          **if** $m_t(\pi_t) = 0$ **then**                        ($\triangleright$) *Full unlocking*: Observe the true label $y_t$
8:             $\Pi_t(\pi_t) \leftarrow \Pi$
9:          **else**                                             ($\triangleright$) *Partial unlocking*
10:             $\Pi_t(\pi_t) \leftarrow \{\pi \in \Pi \mid \pi \geq \pi_t\}$
11:          **for** $\pi \in \Pi_t(\pi_t)$ **do**               ($\triangleright$) Evaluate $m_t(\pi)$ for all $\pi \in \Pi_t(\pi_t)$
12:             $\ell_t(\pi) \leftarrow$ COMPUTELOSS$(\pi, m_t(\pi), \alpha, c)$
13:             Compute normalized gain $g_t(\pi)$, defined as Eq. 8
14:          **for** $\pi \in \Pi$ **do**
15:             Construct biased gain estimator $\tilde{g}_t(\pi | \Pi_t(\pi_t))$, defined as Eq. 11
16:             Update cumulative gain: $\tilde{G}_t(\pi) \leftarrow \tilde{G}_{t-1}(\pi) + \tilde{g}_t(\pi)$
17: **procedure** COMPUTELOSS$(\pi, m, \alpha, c)$
18:      **return** $d_t(\pi; \alpha) + a_t(\pi; \alpha, c)$, defined as Eq. 6 and Eq. 7

---

specified, we do not upweight the pseudo-gain as we did for the normalized gain in the first term of $(A)$. Nevertheless, the construction of the pseudo-gain is once again aligned with our miscoverage-first rationale. Specifically, $\tilde{g}_t(\pi)$ is defined so that $\tilde{g}_t(\pi_1) \geq g_t(\pi_2)$ for all $\pi_1 \in \Pi_t(\pi_t)^c$ and $\pi_2 \in \Pi_t(\pi_t)$, thereby incorporating the monotonicity property that the miscoverage term is non-decreasing as $\pi$ increases. For $\pi \in \Pi_t(\pi_t)$, the biased gain estimator is defined in the same manner as the second term of $(A)$.

Building on the biased gain esitmator as defined in Eq. 11, our `OCP-Unlock+` algorithm provides a long-run coverage guarantee with high probability, by controlling the deviation of the miscoverage rate **MC**$(T)$ from $\alpha$ as in the following theorem.

**Theorem 1.** *For any $\alpha \in (0, 0.5)$, $T \in \mathbb{N}$, $K \geq 5$, and $\delta \in (0, 1)$, run* `OCP-Unlock+` *(Algorithm 1) with $\beta = \sqrt{\frac{\ln K}{KT}}, \gamma = 1.05\sqrt{\frac{K \ln K}{T}}$, and $\eta = 0.95\sqrt{\frac{\ln K}{KT}}$. Then, the deviation of the miscoverage rate* **MC**$(T)$ *from $\alpha$ satisfies*

$$\boldsymbol{MC}(T) - \alpha \leq \ell_{diff}\left(\sqrt{\frac{C \ln K}{T}} + 4.15\sqrt{\frac{K \ln K}{T}} + \sqrt{\frac{K}{T \ln K}}\ln(\delta^{-1}) + 2o(T)^{-1}\right) + C_{\boldsymbol{MC}}(T),$$

*with probability at least $1 - \delta$, where $C$ is the constant defined in Eq. 43 and $\ell_{diff} := \ell_{max} - \ell_{min}$.*

Note that this results from Lemma 1, where the upper bound of **MC**$(T) - \alpha$ is the sum of $C_{\mathbf{MC}}(T)$ and the per-round regret $\frac{1}{T}\mathbf{Reg}(T)$ of `OCP-Unlock+` with respect to the loss defined in Eq. 5 and Eq. 7 (Theorem 5). See Appendix G for proof and additional details, including the introduction of 1 as offset terms in Eq. 11.

## 5 EXPERIMENT

In this section, we empirically evaluate the performance of constructed conformal sets from `OCP-Unlock+` (Algorithm 1) in online conformal prediction with semi-bandit feedback, both in terms of the long-run miscoverage (Eq. 2) and inefficiency (Eq. 3). In addition to the above algorithm, we additionally consider the following baselines: *Semi-bandit Prediction Sets* (`SPS`), and *MultiValid Prediction* (`MVP`). While `SPS` (Ge et al., 2025) provides a long-run miscoverage guarantee in the partial feedback setting, the guarantee is valid under the i.i.d. assumption. In particular, we consider `MVP` (Bastani et al., 2022) as an oracle baseline, since the algorithm is constructed under the *full feedback* setting. See Section 2 for detail.

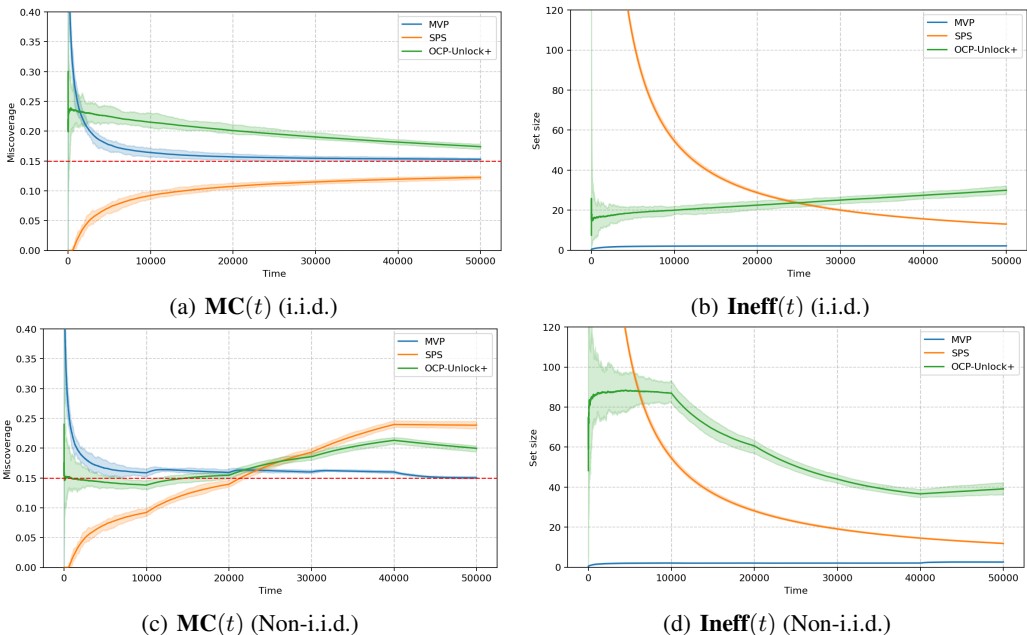

(a) $\mathbf{MC}(t)$ (i.i.d.)

(b) $\mathbf{Ineff}(t)$ (i.i.d.)

(c) $\mathbf{MC}(t)$ (Non-i.i.d.)

(d) $\mathbf{Ineff}(t)$ (Non-i.i.d.)

Figure 2: Long-run miscoverage $\mathbf{MC}(t)$ and inefficiency $\mathbf{Ineff}(t)$ as a function of time $t \in [T]$ on the ImageNet dataset under both the i.i.d. and non-i.i.d. settings, averaged over 50 independent runs

We conduct experiments on both classification and regression tasks, where both i.i.d. and non-i.i.d. settings are considered for each task. In **Experiment 1** (Section 5.1), we consider classification task using ImageNet dataset. We simulate the non-i.i.d. setting by evolving the functional form of scoring function $f_t(\cdot, \cdot)$ over time. In **Experiment 2** (Section 5.2), we consider regression task using the Airfoil Self-Noise dataset (Dua & Graff, 2017) from UCI repository. For the non-i.i.d. scenario, we consider a covariate shift setup where the input distribution changes after the first one-third of the data stream. Moreover, in **Experiment 3** (Appendix I), we present an ablation study on the discretization level $K \in \mathbb{N}$. In **Experiment 4** (Appendix J), we further compare OCP-Unlock+ with its variants, OCP-Bandit (Algorithm 3) and OCP-Unlock (Algorithm 4). In addition, **Experiment 5** (Appendix K) examines the sensitivity of OCP-Unlock+ to the target miscoverage level $\alpha$. Finally, in **Experiment 6** (Appendix L), we empirically analyze the additional term $C_{\mathbf{MC}}(T)$ that appears in the upper bound on the deviation of the miscoverage rate $\mathbf{MC}(T)$ from $\alpha$ (Lemma 1). See Appendix H for additional experimental details.

## 5.1 EXPERIMENT 1: COVERAGE VS INEFFICIENCY ON IMAGENET

For both i.i.d. and non-i.i.d. settings, we evaluate average miscoverage rate $\mathbf{MC}(t)$ and inefficiency $\mathbf{Ineff}(t)$ as a function of time $t \in [T]$, which is averaged over 50 random trials for $\alpha = 0.15$, $K = 200$, and $T = 50,000$ (Figure 2). $\Pi$ is a uniformly discretized set over $[0, 1]$, consisting of $K$ candidate thresholds.

Under the i.i.d. setting, $1 - \mathbf{MC}(t)$ of MVP and SPS closely approach the target coverage level $1 - \alpha$, where MVP has the lowest $\mathbf{Ineff}(t)$ for all $t \in [T]$. Smaller conformal sets from MVP compared to the other baselines are the result of the different amount of feedback available, where MVP can evaluate the miscoverage $m_t(\pi)$ for all $\pi \in \Pi$ at every time step. SPS has $\mathbf{MC}(t) \leq \alpha$ for all $t \in [T]$, which is aligned with its high-probability any-time coverage guarantee under the i.i.d. setting (Ge et al., 2025). Since OCP-Unlock+ exploits additional feedback when $m_t(\pi_t) = 0$, it highlights the importance of incorporating the miscoverage-first rationale in the design of biased gain estimator $\tilde{g}_t(\pi | \Pi_t(\pi_t))$, as illustrated in Section 4.4.

Meanwhile, under the non-i.i.d. setting, both $\mathbf{MC}(t)$ and $\mathbf{Ineff}(t)$ show fluctuating patterns over time across all baselines. Notably, $\mathbf{MC}(t)$ of SPS consistently decreases, having difficulty adapting to the dynamic environment in which the long-run coverage guarantee no longer holds. Similar to the i.i.d. setting, $\mathbf{MC}(t)$ of MVP and OCP-Unlock+ approach the target coverage level, where MVP once again consistently achieves the lowest $\mathbf{Ineff}(t)$.

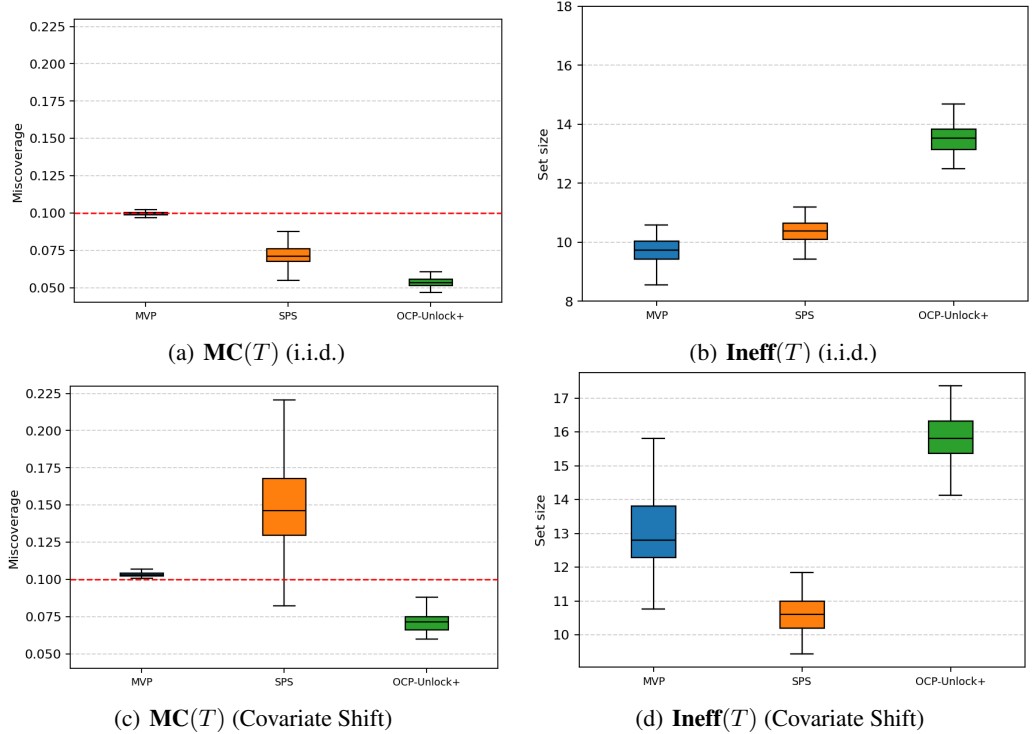

Figure 3: Long-run miscoverage **MC**($T$) and inefficiency **Ineff**($T$) on the Airfoil Self-Noise dataset under the i.i.d. and covariate shift settings, averaged over 50 independent runs

## 5.2 EXPERIMENT 2: COVERAGE VS INEFFICIENCY ON AIRFOIL SELF-NOISE

For both the i.i.d. and covariate-shift settings, we evaluate the long-run miscoverage **MC**($T$) and inefficiency **Ineff**($T$) on the Airfoil Self-Noise dataset, averaged over 50 independent runs with $\alpha = 0.1$, $K = 20$, and $T = 50,000$ (Figure 3). The policy class $\Pi$ is a uniformly discretized subset of $[0, 1]$ with $K$ candidate thresholds.

In the i.i.d. setting, MVP is closest to the target miscoverage level and also attains the smallest inefficiency. SPS and OCP-Unlock+ both produce miscoverage below the target level, with OCP-Unlock+ exhibiting a larger inefficiency. Under covariate shift, MVP continues to remain near the target miscoverage level, whereas SPS becomes less reliable and OCP-Unlock+ remains conservative with a larger set size.

## 6 CONCLUSION

We propose the first algorithms for online conformal prediction under adversarial semi-bandit feedback. By formulating the problem as a multi-armed adversarial bandit problem and designing a loss function tailored to online conformal prediction, we show that any learner achieving sublinear regret with respect to this loss also guarantees long-run coverage at the target level. In particular, our algorithm, OCP-Unlock+, fully exploits the feedback available in the semi-bandit setting and leverages the monotonicity property of single-threshold conformal sets with respect to the miscoverage. As a result, it not only achieves sublinear regret but also demonstrates performance comparable to an oracle baseline with full feedback in both i.i.d. and non-i.i.d. settings. Unlike most online conformal prediction methods that assume full feedback, our approach supports more realistic human-in-the-loop workflows in which the true label is observed only when it lies within the conformal set. Finally, while our algorithms are context-free and do not leverage additional information, extending them to contextual semi-bandit settings presents a promising direction for future research. We also believe that the bound in Theorem 1 may not be tight and that sharper analysis is possible.

ACKNOWLEDGEMENTS

We appreciate constructive feedback by anonymous reviewers. This work was supported by Institute of Information & communications Technology Planning & Evaluation (IITP) and the National Research Foundation of Korea (NRF) grant funded by the Korea government (MSIT) (RS-2019-II191906, Artificial Intelligence Graduate School Program (POSTECH) (5%); RS-2024-00457882, National AI Research Lab Project (15%); No. RS-2024-00509258 and No. RS-2024-00469482, Global AI Frontier Lab (50%); RS-2025-00560062 (30%)).

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

# A  PRELIMINARY

## A.1  ADVERSARIAL BANDIT PROBLEM

The multi-armed bandit problem is a sequential game between a learner and an environment (Slivkins, 2019; Lattimore & Szepesvári, 2020), where the game proceeds over $T \in \mathbb{N}$ rounds. At each round $t \in [T]$, the learner chooses an arm $\pi_t$ from a given set $\Pi$, where we refer to the problem as $K$-armed bandit problem when $|\Pi| = K$. Then, the environment reveals a loss $\ell_t(\pi_t) \in [\ell_{\min}, \ell_{\max}]$, from which the learner updates its strategy to choose an arm in the next round.

In contrast to the *full feedback* setting where a learner observes a whole loss vector $\ell_t \in \mathbb{R}^K$ at each round, the multi-armed bandit problem assumes the *bandit feedback* setting such that only the loss of a chosen arm is observed. Specifically, the adversarial bandit problem makes almost no assumptions on the generation of loss vectors. We often refer to the environment as the adversary in the adversarial bandit problem, where two different types of adversaries are usually considered. First, an oblivious adversary chooses loss vectors $\ell_t$ for all $t \in [T]$ at the start of the game. On the other hand, an adaptive adversary chooses a loss vector $\ell_t$ at round $t$ based on the previous interaction $\ell_1, \pi_1, \ldots, \ell_{t-1}, \pi_{t-1}$ with a learner.

In the adversarial bandit problem, the performance of a learner is measured by its expected regret, where the regret is defined as the difference between the cumulative loss of the learner and that of the best arm in hindsight as

$$\mathbf{Reg}(T) \coloneqq \sum_{t=1}^{T} \ell_t(\pi_t) - \min_{\pi \in \Pi} \sum_{t=1}^{T} \ell_t(\pi).$$

Under the adaptive adversary setup, the expectation is taken with respect to two sources of randomness: the learner's randomized strategy and the adversary's choice of loss vectors. The main goal of adversarial bandit algorithms is to design a learner that provides a sublinear expected regret. In the next section, we provide a simple algorithm under the adaptive adversary that not only achieves sublinear expected regret, but also guarantees sublinear regret with high probability.

## A.2 EXP3.P

EXP3.P (Algorithm 2) is a variant of EXP3 (Auer et al., 2002) that achieves sublinear expected regret under both oblivious and adaptive adversaries. While both algorithms consider an importance-weighted gain estimator $\tilde{g}_t(\pi)$ of $g_t(\pi)$ (Eq. 8) to update their strategy, EXP3.P introduces an additional bonus term $\beta > 0$, thereby making it a biased gain estimator. This enables EXP3.P to further achieve sublinear regret with high probability (Theorem 2) by controlling the variance of regret.

**Theorem 2** (Auer et al. (2002)). *For any $\delta \in (0, 1)$ and $\ell_t(\pi) \in [\ell_{min}, \ell_{max}]$, run EXP3.P (Algorithm 2) with $\beta = \sqrt{\frac{\ln K}{KT}}, \gamma = 1.05\sqrt{\frac{K \ln K}{T}}$, and $\eta = 0.95\sqrt{\frac{\ln K}{KT}}$. Then, with probability at least $1 - \delta$, the regret $\textbf{Reg}(T)$ satisfies*

$$\textbf{Reg}(T) \leq (\ell_{max} - \ell_{min}) \left( \sqrt{\frac{TK}{\ln K}} \ln(\delta^{-1}) + 5.15\sqrt{TK \ln K} \right).$$

---

**Algorithm 2** EXP3.P

1: **procedure** EXP3.P$(\Pi, T, \eta, \gamma, \beta)$
2:     Initialize cumulative estimated gain: $\tilde{G}_0(\pi) \leftarrow 0$ for all $\pi \in \Pi$
3:     **for** $t \in \{1, \dots, T\}$ **do**
4:         Update strategy: $p_t(\pi) \leftarrow (1 - \gamma)\frac{\exp(\eta \tilde{G}_{t-1}(\pi))}{\sum_{\tilde{\pi} \in \Pi} \exp(\eta \tilde{G}_{t-1}(\tilde{\pi}))} + \gamma\frac{1}{K}$, where $K = |\Pi|$
5:         Sample arm: $\pi_t \sim p_t$
6:         Observe loss: $\ell_t(\pi_t)$
7:         Compute normalized gain $g_t(\pi_t)$, defined as Eq. 8
8:         Construct biased gain estimator: $\tilde{g}_t(\pi) \leftarrow \frac{g_t(\pi)\mathbb{1}(\pi_t = \pi) + \beta}{p_t(\pi)}$ for all $\pi \in \Pi$
9:         Update cumulative gain: $\tilde{G}_t(\pi) \leftarrow \tilde{G}_{t-1}(\pi) + \tilde{g}_t(\pi)$ for all $\pi \in \Pi$

---

## B    PROOF OF LEMMA 1

Let

$$
\begin{aligned}
\ell_t(\pi) &= d_t(\pi) + a_t(\pi) \\
&= m_t(\pi) + \mathbb{1}(m_t(\pi) = 0)\underbrace{(\alpha + \alpha(1-\alpha))}_{g(\alpha)} + \mathbb{1}(m_t(\pi) = 1)\underbrace{(-\alpha(1-\alpha))}_{h(\alpha)} + a_t(\pi) \\
&= \underbrace{(m_t(\pi) + \mathbb{1}(m_t(\pi) = 0)g(\alpha) + \mathbb{1}(m_t(\pi) = 1)h(\alpha))}_{\text{Miscoverage Term (Discrepancy decreases as } \alpha \uparrow 0.5)} + \underbrace{a_t(\pi)}_{\text{Inefficiency Term}},
\end{aligned}
$$

where Lemma 1 assumes that $a_t(\pi)$ is of order $o(T)^{-1}$ for all $\pi \in \Pi$.

Then,

$$
\begin{aligned}
\mathbf{Reg}\,(T) &= \sum_{t=1}^{T} \ell_t(\pi_t) - \min_{\pi \in \Pi} \sum_{t=1}^{T} \ell_t(\pi) \\
&= \left( \sum_{m_t(\pi_t)=0} \ell_t(\pi_t) + \sum_{m_t(\pi_t)=1} \ell_t(\pi_t) \right) - \min_{\pi \in \Pi} \sum_{t=1}^{T} \ell_t(\pi) \\
&\geq \left( \sum_{m_t(\pi_t)=0} \ell_t(\pi_t) + \sum_{m_t(\pi_t)=1} \ell_t(\pi_t) \right) - \sum_{t=1}^{T} \ell_t(0) \quad (\because \text{ Simply choose } \pi = 0) \\
&= \left( \sum_{t=1}^{T} (m_t(\pi_t) + a_t(\pi_t)) + \sum_{m_t(\pi_t)=0} g(\alpha) + \sum_{m_t(\pi_t)=1} h(\alpha) \right) - T(\alpha + \alpha(1-\alpha) + a_t(0)) \\
&= \left( \sum_{t=1}^{T} (m_t(\pi_t) + a_t(\pi_t)) + \sum_{m_t(\pi_t)=1} (-\alpha - 2\alpha(1-\alpha)) \right) - Ta_t(0) \\
&= \left( \sum_{t=1}^{T} m_t(\pi_t) + \sum_{m_t(\pi_t)=1} (-\alpha - 2\alpha(1-\alpha)) \right) - \underbrace{\left( Ta_t(0) - \sum_{t=1}^{T} a_t(\pi_t) \right)}_{TC_1} \\
&= \left( \sum_{t=1}^{T} m_t(\pi_t) - \sum_{m_t(\pi_t)=0} \alpha + \sum_{m_t(\pi_t)=0} \alpha + \sum_{m_t(\pi_t)=1} (-\alpha - 2\alpha(1-\alpha)) \right) - TC_1 \\
&= \left( \sum_{t=1}^{T} m_t(\pi_t) - T\alpha \right) + \left( \sum_{m_t(\pi_t)=0} \alpha + \sum_{m_t(\pi_t)=1} (-2\alpha(1-\alpha)) \right) - TC_1.
\end{aligned}
$$

Note that $C_1$ is of order $o(T)^{-1}$, as $a_t(\pi)$ is of order $o(T)^{-1}$ for all $\pi \in \Pi$.

For a given learner, there exists a constant $C_{\text{gap}}(T)$ that satisfies the following relationship:

$$
C_{\text{gap}}(T)o(T)^{-1} = \frac{1-\alpha}{\alpha} - \frac{\sum_{t=1}^{T} \mathbb{1}(m_t(\pi_t) = 0)}{\sum_{t=1}^{T} \mathbb{1}(m_t(\pi_t) = 1)}.
$$

Then,

$$
\begin{aligned}
\mathbf{Reg}(T) &\geq \left( \sum_{t=1}^{T} m_t(\pi_t) - T\alpha \right) + \left( \underbrace{\sum_{m_t(\pi_t)=1} (1-2\alpha)(1-\alpha)}_{\geq 0} - \alpha \sum_{m_t(\pi_t)=1} C_{\text{gap}}(T)o(T)^{-1} \right) - TC_1 \\
&\geq \left( \sum_{t=1}^{T} m_t(\pi_t) - T\alpha \right) - TC_1 - \alpha \sum_{m_t(\pi_t)=1} C_{\text{gap}}(T)o(T)^{-1}.
\end{aligned}
$$

Dividing each side by $T$, we have

$$
\mathbf{MC}(T) - \alpha \leq \frac{1}{T}\mathbf{Reg}(T) + \underbrace{C_1 + \alpha \frac{\sum_{t=1}^{T} \mathbb{1}(m_t(\pi_t) = 1)}{T} C_{\text{gap}}(T)o(T)^{-1}}_{C_{\mathbf{MC}}(T)}.
$$

## C  EXP3.P-STYLE ALGORITHMS FOR ONLINE CONFORMAL PREDICTION WITH ADVERSARIAL PARTIAL FEEDBACK

We propose three modifications to the EXP3.P algorithm tailored to the setting of online conformal prediction with adversarial partial feedback: OCP-Bandit (Algorithm 3), OCP-Unlock (Algorithm 4), and OCP-Unlock+ (Algorithm 1). While sharing the same algorithmic structure, the three variants differ in their specification of the unlocking set $\Pi_t(\pi_t)$ and in the definition of the biased gain estimator $\tilde{g}_t(\pi|\Pi_t(\pi_t))$.

Before describing each algorithm in detail, Table 2 and Table 3 provide a high-level overview of these differences. The difference in the design of $\tilde{g}_t(\pi|\Pi_t(\pi_t))$ between OCP-Bandit and OCP-Unlock is highlighted in red, while the additional difference between OCP-Unlock and OCP-Unlock+, on top of the difference between the first two baselines, is highlighted in blue. The offset term 1 of $\tilde{g}_t(\pi \mid \Pi_t(\pi_t))$ in OCP-Unlock+ (Eq. 11) is omitted from the tables solely to highlight the differences among the three methods.

Table 2: Comparison of the unlocking set $\Pi_t(\pi_t)$ and the biased gain estimator $\tilde{g}_t(\pi \mid \Pi_t(\pi_t))$ for each method when $m_t(\pi_t) = 0$

| Feedback Type | Method | $\Pi_t(\pi_t)$ | $\tilde{g}_t(\pi|\Pi_t(\pi_t))$ |
|---|---|---|---|
| Bandit Feedback | OCP-Bandit | $\{\pi_t\}$ | $\mathbb{1}(\pi = \pi_t)\left\{\frac{g_t(\pi)}{p_t(\pi)} + \frac{\beta}{p_t(\pi)}\right\} + \mathbb{1}(\pi \neq \pi_t)\frac{\beta}{p_t(\pi)}$ |
| Semi-bandit Feedback | OCP-Unlock | $\Pi = \Pi_t^\star \cup (\Pi_t^\star)^c$ | $\mathbb{1}(\pi \in \Pi_t^\star)\left\{\frac{g_t(\pi)}{\sum_{\tilde{\pi} \in \Pi_t^\star} p_t(\tilde{\pi})} + \frac{\beta}{p_t(\pi)}\right\} + \mathbb{1}(\pi \in (\Pi_t^\star)^c)\frac{\beta}{p_t(\pi)}$ |
|  | OCP-Unlock+ |  | $\mathbb{1}(\pi \in \Pi_t^\star)\left\{\frac{g_t(\pi)}{\sum_{\tilde{\pi} \in \Pi_t^\star} p_t(\tilde{\pi})} + \frac{\beta}{\sum_{\tilde{\pi} \in \Pi_t^\star} p_t(\tilde{\pi})}\right\} + \mathbb{1}(\pi \in (\Pi_t^\star)^c)\left\{g_t(\pi) + \frac{\beta}{\sum_{\tilde{\pi} \leq \pi} p_t(\tilde{\pi})}\right\}$ |

Table 3: Comparison of the unlocking set $\Pi_t(\pi_t)$ and the biased gain estimator $\tilde{g}_t(\pi \mid \Pi_t(\pi_t))$ for each method when $m_t(\pi_t) = 1$

| Feedback Type | Method | $\Pi_t(\pi_t)$ | $\tilde{g}_t(\pi|\Pi_t(\pi_t))$ |
|---|---|---|---|
| Bandit Feedback | OCP-Bandit | $\{\pi_t\}$ | $\mathbb{1}(\pi = \pi_t)\left\{\frac{g_t(\pi)}{p_t(\pi)} + \frac{\beta}{p_t(\pi)}\right\} + \mathbb{1}(\pi \neq \pi_t)\frac{\beta}{p_t(\pi)}$ |
| Semi-bandit Feedback | OCP-Unlock | $\{\pi \in \Pi : \pi \geq \pi_t\}$ | $\mathbb{1}(\pi \in \Pi_t(\pi_t))\left\{\frac{g_t(\pi)}{\sum_{\tilde{\pi} \in \Pi_t(\pi_t)} p_t(\tilde{\pi})} + \frac{\beta}{p_t(\pi)}\right\} + \mathbb{1}(\pi \in \Pi_t(\pi_t)^c)\frac{\beta}{p_t(\pi)}$ |
|  | OCP-Unlock+ |  | $\mathbb{1}(\pi \in \Pi_t(\pi_t))\left\{g_t(\pi) + \frac{\beta}{\sum_{\tilde{\pi} \leq \pi} p_t(\tilde{\pi})}\right\} + \mathbb{1}(\pi \in \Pi_t(\pi_t)^c)\left\{\tilde{g}_t(\pi) + \frac{\beta}{p_t(\pi)}\right\}$ |

### C.1  OCP-BANDIT

As illustrated in Section 4.4, OCP-Bandit (Algorithm 3) is a direct modification of EXP3.P (Algorithm 2) for online conformal prediction with adversarial partial feedback. Accordingly, irrespective of the miscoverage feedback $m_t(\pi_t)$ from the chosen arm, the unlocking set is defined as a singleton set containing the chosen arm $\pi_t$:

$$\Pi_t(\pi_t) := \{\pi_t\}. \tag{13}$$

The biased gain estimator $\tilde{g}_t(\pi|\Pi_t(\pi_t))$ is then defined as in Eq. 9:

$$\tilde{g}_t\left(\pi \mid \Pi_t(\pi_t)\right) := \underbrace{\mathbb{1}(m_t(\pi_t) = 0)}_{\text{no unlocking}} \times (A) + \underbrace{\mathbb{1}(m_t(\pi_t) = 1)}_{\text{no unlocking}} \times (B), \tag{14}$$

where

$$(A) = (B) = \mathbb{1}(\pi = \pi_t)\left\{\frac{g_t(\pi)}{p_t(\pi)} + \frac{\beta}{p_t(\pi)}\right\} + \mathbb{1}(\pi \neq \pi_t)\frac{\beta}{p_t(\pi)}.$$

While applicable to the partial feedback setting, OCP-Bandit is tailored to the bandit feedback setting as EXP3.P, thereby ignoring additional information available. Under this design, OCP-Bandit satisfies the following high-probability bound.

**Theorem 3** (Auer et al., 2002). *For any $\alpha \in (0, 0.5), T \in \mathbb{N}$, and $\delta \in (0, 1)$, run OCP-Bandit (Algorithm 3) with $\beta = \sqrt{\frac{\ln K}{KT}}, \gamma = 1.05\sqrt{\frac{K \ln K}{T}}$, and $\eta = 0.95\sqrt{\frac{\ln K}{KT}}$. Then, with probability at least $1 - \delta$, the regret $\mathbf{Reg}(T)$ with respect to the loss defined in Eq. 5 and Eq. 7 satisfies*

$$\mathbf{Reg}(T) \leq (\ell_{max} - \ell_{min})\left(5.15\sqrt{TK \ln K} + \sqrt{\frac{TK}{\ln K}}\ln(\delta^{-1})\right).$$

See Appendix D for proof and additional details.

---

**Algorithm 3** `OCP-Bandit`

---

1: **procedure** OCP-BANDIT($\Pi, T, \eta, \gamma, \beta, \alpha, c, (f_t)_{t=1}^T$)
2:     Initialize cumulative estimated gain: $\tilde{G}_0(\pi) \leftarrow 0$ for all $\pi \in \Pi$
3:     **for** $t \in \{1, \dots, T\}$ **do**
4:         Update strategy: $p_t(\pi) \leftarrow (1 - \gamma) \frac{\exp(\eta \tilde{G}_{t-1}(\pi))}{\sum_{\tilde{\pi} \in \Pi} \exp(\eta \tilde{G}_{t-1}(\tilde{\pi}))} + \gamma \frac{1}{K}$, where $K = |\Pi|$
5:         Sample arm: $\pi_t \sim p_t$
6:         Receive $m_t(\pi_t) \leftarrow \mathbb{1}\left( y_t \notin \hat{C}_{\pi_t}(x_t) \right)$, where $\hat{C}_{\pi_t}(x_t)$ is defined as Eq. 1
7:         $\Pi_t(\pi_t) \leftarrow \{\pi_t\}$                                        ($\triangleright$) *No unlocking*
8:         **for** $\pi \in \Pi_t(\pi_t)$ **do**                         ($\triangleright$) Evaluate $m_t(\pi)$ for all $\pi \in \Pi_t(\pi_t)$
9:             $\ell_t(\pi) \leftarrow$ COMPUTELOSS($\pi, m_t(\pi), \alpha, c$)
10:            Compute normalized gain $g_t(\pi)$, defined as Eq. 8
11:        **for** $\pi \in \Pi$ **do**
12:            Construct biased gain estimator $\tilde{g}_t(\pi | \Pi_t(\pi_t))$, defined as Eq. 9
13:            Update cumulative gain: $\tilde{G}_t(\pi) \leftarrow \tilde{G}_{t-1}(\pi) + \tilde{g}_t(\pi)$
14: **procedure** COMPUTELOSS($\pi, m, \alpha, c$)
15:    **return** $d_t(\pi; \alpha) + a_t(\pi; \alpha, c)$, defined as Eq. 6 and Eq. 7

---

## C.2  OCP-UNLOCK

In contrast to `OCP-Bandit`, `OCP-Unlock` (Algorithm 4) utilizes additional information available when $m_t(\pi_t) = 0$, and leverages the monotonicity of the miscoverage term $m_t(\pi)$ with respect to the threshold $\pi \in \Pi$ to obtain additional feedback when $m_t(\pi_t) = 1$. As such, the unlocking set $\Pi_t(\pi_t)$ is defined as in Eq. 10:

$$\Pi_t(\pi_t) := \begin{cases} \Pi & \text{if } m_t(\pi_t) = 0 \\ \{\pi \in \Pi : \pi \geq \pi_t\} & \text{if } m_t(\pi_t) = 1 \end{cases}.$$

The biased gain estimator $\tilde{g}_t(\pi | \Pi_t(\pi_t))$ is then defined as

$$\tilde{g}_t\left(\pi \mid \Pi_t(\pi_t)\right) := \underbrace{\mathbb{1}(m_t(\pi_t) = 0)}_{\text{full unlocking}} \times (A) + \underbrace{\mathbb{1}(m_t(\pi_t) = 1)}_{\text{partial unlocking}} \times (B), \tag{15}$$

where

$$(A) := \mathbb{1}(\pi \in \Pi_t^*) \left\{ \frac{g_t(\pi)}{\sum_{\tilde{\pi} \in \Pi_t^*} p_t(\tilde{\pi})} + \frac{\beta}{p_t(\pi)} \right\} + \mathbb{1}(\pi \in (\Pi_t^*)^c) \frac{\beta}{p_t(\pi)}$$

and

$$(B) := \mathbb{1}(\pi \in \Pi_t(\pi_t)) \left\{ \frac{g_t(\pi)}{\sum_{\tilde{\pi} \in \Pi_t(\pi_t)} p_t(\tilde{\pi})} + \frac{\beta}{p_t(\pi)} \right\} + \mathbb{1}(\pi \in \Pi_t(\pi_t)^c) \frac{\beta}{p_t(\pi)}.$$

As highlighted in red in Table 2 and Table 3, note that `OCP-Unlock` is designed such that it reduces to `OCP-Bandit` under a bandit feedback setting, where the unlocking set is defined as in Eq. 13. The following theorem establishes a high-probability bound for `OCP-Unlock`.

**Theorem 4.** *For any $\alpha \in (0, 0.5), T \in \mathbb{N}, \delta \in (0, 1)$, and $\lambda > 0$, run* `OCP-Unlock` *(Algorithm 4) with $\beta = \sqrt{\frac{\ln K}{KT}}, \gamma = 1.05\sqrt{\frac{K \ln K}{T}}$, and $\eta = 0.95\sqrt{\frac{\ln K}{KT}}$. Then, with probability at least $1 - \delta$, the regret* $\textbf{Reg}(T)$ *with respect to the loss defined in Eq. 5 and Eq. 7 satisfies*

$$\textbf{Reg}(T) \leq (\ell_{max} - \ell_{min}) \left( 5.15\sqrt{TK \ln K} + \sqrt{\frac{TK}{\ln K}} \ln(\delta^{-1}) + o(T)^{-1}T \right).$$

See Appendix F for proof and additional details.

---

**Algorithm 4** `OCP-Unlock`

---

1: **procedure** OCP-UNLOCK($\Pi, T, \eta, \gamma, \beta, \alpha, c, (f_t)_{t=1}^T$)
2:     Initialize cumulative estimated gain: $\tilde{G}_0(\pi) \leftarrow 0$ for all $\pi \in \Pi$
3:     **for** $t \in \{1, \ldots, T\}$ **do**
4:         Update strategy: $p_t(\pi) \leftarrow (1 - \gamma) \frac{\exp(\eta \tilde{G}_{t-1}(\pi))}{\sum_{\tilde{\pi} \in \Pi} \exp(\eta \tilde{G}_{t-1}(\tilde{\pi}))} + \gamma \frac{1}{K}$, where $K = |\Pi|$
5:         Sample arm: $\pi_t \sim p_t$
6:         Receive $m_t(\pi_t) \leftarrow \mathbb{1}\left(y_t \notin \hat{C}_{\pi_t}(x_t)\right)$, where $\hat{C}_{\pi_t}(x_t)$ is defined as Eq. 1
7:         **if** $m_t(\pi_t) = 0$ **then**                 ($\triangleright$) *Full unlocking*: Observe the true label $y_t$
8:             $\Pi_t(\pi_t) \leftarrow \Pi$
9:         **else**                                         ($\triangleright$) *Partial unlocking*
10:             $\Pi_t(\pi_t) \leftarrow \{\pi \in \Pi \mid \pi \geq \pi_t\}$
11:         **for** $\pi \in \Pi_t(\pi_t)$ **do**             ($\triangleright$) Evaluate $m_t(\pi)$ for all $\pi \in \Pi_t(\pi_t)$
12:             $\ell_t(\pi) \leftarrow$ COMPUTELOSS$(\pi, m_t(\pi), \alpha, c)$
13:             Compute normalized gain $g_t(\pi)$, defined as Eq. 8
14:         **for** $\pi \in \Pi$ **do**
15:             Construct biased gain estimator $\tilde{g}_t(\pi|\Pi_t(\pi_t))$, defined as Eq. 15
16:             Update cumulative gain: $\tilde{G}_t(\pi) \leftarrow \tilde{G}_{t-1}(\pi) + \tilde{g}_t(\pi)$
17: **procedure** COMPUTELOSS$(\pi, m, \alpha, c)$
18:     **return** $d_t(\pi; \alpha) + a_t(\pi; \alpha, c)$, defined as Eq. 6 and Eq. 7

---

## C.3 OCP-UNLOCK+

`OCP-Unlock+` (Algorithm 1) uses the same definition of the unlocking set $\Pi_t(\pi_t)$ (Eq. 10) as `OCP-Unlock`:

$$\Pi_t(\pi_t) := \begin{cases} \Pi & \text{if } m_t(\pi_t) = 0 \\ \{\pi \in \Pi : \pi \geq \pi_t\} & \text{if } m_t(\pi_t) = 1 \end{cases}.$$

However, we further strengthen the *miscoverage-first* rationale in the design of the biased gain estimator $\tilde{g}_t(\pi|\Pi_t(\pi_t))$ (Eq. 11), defined as follows:

$$\tilde{g}_t(\pi \mid \Pi_t(\pi_t)) := \underbrace{\mathbb{1}(m_t(\pi_t) = 0)}_{\text{full unlocking}} \times (A) + \underbrace{\mathbb{1}(m_t(\pi_t) = 1)}_{\text{partial unlocking}} \times (B),$$

where

$$(A) = \mathbb{1}(\pi \in \Pi_t^*) \left\{ \frac{g_t(\pi)}{\sum_{\tilde{\pi} \in \Pi_t^*} p_t(\tilde{\pi})} + \frac{\beta}{\sum_{\tilde{\pi} \in \Pi_t^*} p_t(\tilde{\pi})} \right\} + \mathbb{1}(\pi \in (\Pi_t^*)^c) \left\{ g_t(\pi) + \frac{\beta}{\sum_{\tilde{\pi} \leq \pi} p_t(\tilde{\pi})} \right\}$$

and

$$(B) = \mathbb{1}(\pi \in \Pi_t(\pi_t)) \left\{ g_t(\pi) + \frac{\beta}{\sum_{\tilde{\pi} \leq \pi} p_t(\tilde{\pi})} \right\} + \mathbb{1}(\pi \in \Pi_t(\pi_t)^c) \left\{ \tilde{g}_t(\pi) + \frac{\beta}{p_t(\pi)} \right\}.$$

The main difference from `OCP-Unlock` is that `OCP-Unlock+` additionally introduces a pseudo-gain $\tilde{g}_t(\pi)$ when $m_t(\pi_t) = 1$. Unlike `OCP-Unlock`, the introduction of $\tilde{g}_t(\pi)$ ensures $\tilde{g}_t(\pi_1|\Pi_t(\pi_t)) \geq \tilde{g}_t(\pi_2|\Pi_t(\pi_t))$ for all $\pi_1 \in \Pi_t(\pi_t)^c$ and $\pi_2 \in \Pi_t(\pi_t)$ when $m_t(\pi_t) = 1$, which aligns with the monotonicity property that the miscoverage term decreases as $|\hat{C}_\pi(x_t)|$ increases. Despite the use of additional information under a partial feedback setting, the lack of the pseudo-gain results in the misalignment with the monotonicity property, which may be the cause of sub-optimal empirical performance of `OCP-Unlock` in terms of approaching the target coverage level (Section 5).

In addition, for both cases when $m_t(\pi_t) = 0$ and $m_t(\pi_t) = 1$, importance weights on both normalized gains (pseudo-gains) and exploration parameter $\beta$ are designed such that both gain and exploration terms for $\pi \in \Pi_t^*$ are larger than those from $\pi \in (\Pi_t^*)^c$. Furthermore, for $\pi \notin \Pi_t^*$, we further strengthen this miscoverage-first rationale by placing more weight on the exploration term $\beta$ for larger conformal sets through the denominator $\sum_{\tilde{\pi} \leq \pi} p_t(\tilde{\pi})$. See Figure 4 for illustration.

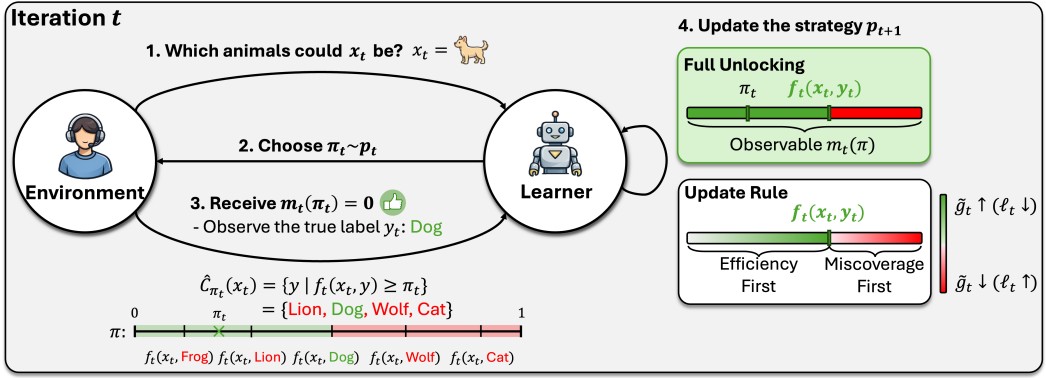

(a) Full unlocking when $m_t(\pi_t) = 0$

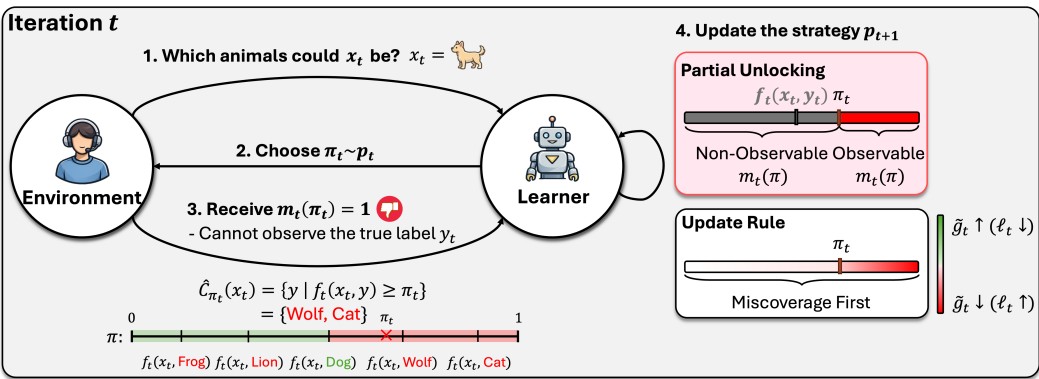

(b) Partial unlocking when $m_t(\pi_t) = 1$

Figure 4: Overview of `OCP-Unlock+` for online conformal prediction with semi-bandit feedback

We now state a high-probability bound for `OCP-Unlock+`.

**Theorem 5.** *For any $\alpha \in (0, 0.5), T \in \mathbb{N}, K \geq 5$, and $\delta \in (0,1)$, run `OCP-Unlock+` (Algorithm 1) with $\beta = \sqrt{\frac{\ln K}{KT}}, \gamma = 1.05\sqrt{\frac{K \ln K}{T}}$, and $\eta = 0.95\sqrt{\frac{\ln K}{KT}}$. Then, with probability at least $1 - \delta$, the regret $\mathbf{Reg}(T)$ with respect to the loss defined in Eq. 5 and Eq. 7 satisfies*

$$\mathbf{Reg}(T) \leq (\ell_{max} - \ell_{min})\left(\sqrt{TC \ln K} + 4.15\sqrt{TK \ln K} + \sqrt{\frac{TK}{\ln K}} \ln(\delta^{-1}) + 2o(T)^{-1}T\right),$$

where $C$ is the constant defined in Eq. 43. See Appendix G for proof and additional details.

Note that `OCP-Unlock+` achieves the tightest bound as long as $(\sqrt{K} - \sqrt{C})\sqrt{T \ln K} > 2o(T)^{-1}T$. In addition, using the same argument based on Lemma 1 and Theorem 5 to derive the high-probability upper bound of $\mathbf{MC}(T) - \alpha$ (Theorem 1), we can show that both long-run coverage results of `OCP-Bandit` and `OCP-Unlock` are at least $1 - \alpha$ with high-probability.

# D   PROOF OF THEOREM 3

The proof is based on that of Theorem 2 (Auer et al., 2002), where we have slightly modified the original proof to accommodate arbitrary bounded loss functions. Specifically, we first provide an arm-wise high-probability bound (Lemma 2), which is then used to derive Theorem 3.

Note that both the proofs of `OCP-Unlock` (Theorem 4) and `OCP-Unlock+` (Theorem 5) follow a similar structure to that of `OCP-Bandit` (Theorem 3), accompanied by additional technical details due to the introduction of the unlocking set $\Pi_t(\pi_t)$ and the modification of the biased gain estimator $\tilde{g}_t(\pi|\Pi_t(\pi_t))$ accordingly. See Appendix E and Appendix G for detail.

First, we show that the biased gain estimator $\tilde{g}_t(\pi|\Pi_t(\pi_t))$ of `OCP-Bandit` (Eq. 14) satisfies the following high-probability inequality for all $\pi \in \Pi$. Note that Eq. 14 is equivalent to Eq. 9, which is re-expressed to clarify the difference between the biased gain estimators of `OCP-Unlock` and `OCP-Unlock+`, as detailed in Appendix C.

**Lemma 2.** *Let $\beta \leq 1$ and $\delta \in (0,1)$. Then, for each $\pi \in \Pi$, the following holds with probability at least $1 - \delta$:*

$$\sum_{t=1}^{T} g_t(\pi) \leq \sum_{t=1}^{T} \tilde{g}_t(\pi|\Pi_t(\pi_t)) + \frac{\ln(\delta^{-1})}{\beta}.$$

*Proof.* Let $\mathbb{E}_t$ be the expectation with respect to the distribution $p_t$, from which $\pi_t$ is sampled. Since $\exp(x) \leq 1 + x + x^2$ for $x \leq 1$, for $\beta \leq 1$, by letting $\Delta_t(\pi) := \beta g_t(\pi) - \frac{\beta g_t(\pi)\mathbb{1}(\pi_t = \pi)}{p_t(\pi)} \leq 1$, we have

$$\mathbb{E}_t \left[ \exp\left( \Delta_t(\pi) - \frac{\beta^2}{p_t(\pi)} \right) \right] \leq \left( 1 + \mathbb{E}_t[\Delta_t(\pi)] + \mathbb{E}_t[\Delta_t(\pi)^2] \right) \exp\left( -\frac{\beta^2}{p_t(\pi)} \right)$$

$$\leq \left( 1 + \frac{\beta^2 g_t(\pi)^2}{p_t(\pi)} \right) \exp\left( -\frac{\beta^2}{p_t(\pi)} \right)$$

$$\leq \left( 1 + \frac{\beta^2}{p_t(\pi)} \right) \exp\left( -\frac{\beta^2}{p_t(\pi)} \right) \ (\because g_t(\cdot) \in [0,1])$$

$$\leq 1 \ (\because 1 + u \leq \exp(u)).$$

By sequentially applying the double expectation rule for $t = T, \ldots, 1$,

$$\mathbb{E} \exp\left[ \sum_{t=1}^{T} \left( \Delta_t(\pi) - \frac{\beta^2}{p_t(\pi)} \right) \right] \leq 1. \tag{16}$$

Moreover, from the Markov's inequality, we have $\mathbb{P}(X > \ln(1/\delta)) = \mathbb{P}(\exp(X) > 1/\delta) \leq \delta \mathbb{E} \exp(X)$. Combined with Eq. 16, we have

$$\beta \sum_{t=1}^{T} g_t(\pi) \leq \beta \sum_{t=1}^{T} \tilde{g}_t(\pi|\Pi_t(\pi_t)) + \ln(\delta^{-1})$$

with probability at least $1 - \delta$. This completes the proof. $\qquad\square$

Now, we derive the the regret bound of `OCP-Bandit` (Algorithm 3), which consists of three steps.

First, our goal is to show that, if $\gamma \leq \frac{1}{2}$ and $(1+\beta)K\eta \leq \gamma$,

$$\mathbf{Reg}(T) \leq (\ell_{\max} - \ell_{\min}) \left( K\beta T + \gamma T + (1+\beta)\eta Kn + \frac{\ln(K\delta^{-1})}{\beta} + \frac{\ln K}{\eta} \right). \tag{17}$$

Irrespective of the hyperparameter setup, note that Theorem 3 always holds if $T < \sqrt{\frac{TK}{\ln K}} \ln(\delta^{-1}) + 5.15\sqrt{TK\ln K}$. If $T \geq \sqrt{\frac{TK}{\ln K}} \ln(\delta^{-1}) + 5.15\sqrt{TK\ln K}$, this implies that $\gamma \leq \frac{1}{2}$ and $(1+\beta)K\eta \leq \gamma$, which makes it suffice to show that Eq. 17 holds for $\gamma \leq \frac{1}{2}$ and $(1+\beta)K\eta \leq \gamma$.

**Step 1: Simple equalities.** For all $\pi \in \Pi$, the following equality holds:

$$\mathbb{E}_{\pi \sim p_t} \tilde{g}_t(\pi | \Pi_t(\pi_t)) = g_t(\pi_t) + \beta K. \tag{18}$$

Then, for all $\pi \in \Pi$, the following equality holds:

$$
\begin{aligned}
R_\pi(T) &:= \sum_{t=1}^{T} \ell_t(\pi_t) - \sum_{t=1}^{T} \ell_t(\pi) \\
&= (\ell_{\max} - \ell_{\min}) \left( \sum_{t=1}^{T} g_t(\pi) - \sum_{t=1}^{T} g_t(\pi_t) \right) \ (\because \text{Eq. 8}) \\
&= (\ell_{\max} - \ell_{\min}) \left( \beta K T + \sum_{t=1}^{T} g_t(\pi) - \sum_{t=1}^{T} \mathbb{E}_{\tilde{\pi} \sim p_t} \tilde{g}_t(\tilde{\pi} | \Pi_t(\pi_t)) \right) \ (\because \text{Eq. 18}).
\end{aligned}
\tag{19}
$$

Using the definition of cumulant generating function and the relationship that $p_t = (1 - \gamma)\omega_t + \gamma u$ where $\omega_t(\pi) = \dfrac{\exp(\eta \tilde{G}_{t-1}(\pi))}{\sum_{\tilde{\pi} \in \Pi} \exp(\eta \tilde{G}_{t-1}(\tilde{\pi}))}$ and $u$ is the uniform distribution over $K$ arms, the following holds:

$$
\begin{aligned}
&-\mathbb{E}_{\tilde{\pi} \sim p_t} \tilde{g}_t(\tilde{\pi} | \Pi_t(\pi_t)) \\
&= -(1 - \gamma) \mathbb{E}_{\tilde{\pi} \sim \omega_t} \tilde{g}_t(\tilde{\pi} | \Pi_t(\pi_t)) - \gamma \mathbb{E}_{\tilde{\pi} \sim u} \tilde{g}_t(\tilde{\pi} | \Pi_t(\pi_t)) \ (\because p_t = (1 - \gamma)\omega_t + \gamma u) \\
&= (1 - \gamma) \left[ \frac{1}{\eta} \ln \mathbb{E}_{\tilde{\pi} \sim \omega_t} \exp \left( \eta (\tilde{g}_t(\tilde{\pi} | \Pi_t(\pi_t)) - \mathbb{E}_{\tilde{\pi} \sim \omega_t} \tilde{g}_t(\tilde{\pi} | \Pi_t(\pi_t))) \right) \right. \\
&\qquad \left. - \frac{1}{\eta} \ln \mathbb{E}_{\tilde{\pi} \sim \omega_t} \exp(\eta \tilde{g}_t(\tilde{\pi} | \Pi_t(\pi_t))) \right] - \gamma \mathbb{E}_{\tilde{\pi} \sim u} \tilde{g}_t(\tilde{\pi} | \Pi_t(\pi_t)).
\end{aligned}
\tag{20}
$$

**Step 2: Bounding the first term of Eq. 20.** Since $\ln x \leq x - 1$, $\exp(x) \leq 1 + x + x^2$ for all $x \leq 1$, and $\eta \tilde{g}_t(\tilde{\pi} | \Pi_t(\pi_t)) \leq \eta \frac{g_t(\tilde{\pi}) + \beta}{p_t(\tilde{\pi})} \leq \eta \frac{1 + \beta}{(1 - \gamma)w_t(\tilde{\pi}) + \gamma \frac{1}{K}} \leq \frac{\gamma \frac{1}{K}}{(1 - \gamma)w_t(\tilde{\pi}) + \gamma \frac{1}{K}} \leq 1 \ (\because (1 + \beta)\eta K \leq \gamma)$,

$$
\begin{aligned}
&\ln \mathbb{E}_{\tilde{\pi} \sim \omega_t} \exp \left( \eta (\tilde{g}_t(\tilde{\pi} | \Pi_t(\pi_t)) - \mathbb{E}_{\tilde{\pi} \sim \omega_t} \tilde{g}_t(\tilde{\pi} | \Pi_t(\pi_t))) \right) \\
&= \ln \mathbb{E}_{\tilde{\pi} \sim \omega_t} \exp \left( \eta \tilde{g}_t(\tilde{\pi} | \Pi_t(\pi_t)) \right) - \eta \mathbb{E}_{\tilde{\pi} \sim \omega_t} \tilde{g}_t(\tilde{\pi} | \Pi_t(\pi_t)) \\
&\leq \mathbb{E}_{\tilde{\pi} \sim \omega_t} \left\{ \exp(\eta \tilde{g}_t(\tilde{\pi} | \Pi_t(\pi_t))) - 1 - \eta \tilde{g}_t(\tilde{\pi} | \Pi_t(\pi_t)) \right\} \ (\because \ln x \leq x - 1) \\
&\leq \mathbb{E}_{\tilde{\pi} \sim \omega_t} \eta^2 \tilde{g}_t(\tilde{\pi} | \Pi_t(\pi_t))^2 \ (\because \exp(x) \leq 1 + x + x^2) \\
&\leq \eta^2 \frac{1 + \beta}{1 - \gamma} \sum_{\tilde{\pi} \in \Pi} \tilde{g}_t(\tilde{\pi} | \Pi_t(\pi_t)) \ \left( \because \frac{w_t(\tilde{\pi})}{p_t(\tilde{\pi})} \leq \frac{1}{1 - \gamma} \right).
\end{aligned}
\tag{21}
$$

**Step 3: Summing.** Let $\tilde{G}_0(\tilde{\pi}) = 0$. Then, combining Eq. 20-Eq. 21 and summing over $t$ yield

$$
\begin{aligned}
&-\sum_{t=1}^{T} \mathbb{E}_{\tilde{\pi} \sim p_t} \tilde{g}_t(\tilde{\pi} | \Pi_t(\pi_t)) \\
&\leq (1 + \beta)\eta \sum_{t=1}^{T} \sum_{\tilde{\pi} \in \Pi} \tilde{g}_t(\tilde{\pi} | \Pi_t(\pi_t)) - \frac{1 - \gamma}{\eta} \sum_{t=1}^{T} \ln \left( \sum_{\tilde{\pi} \in \Pi} w_t(\tilde{\pi}) \exp(\eta \tilde{g}_t(\tilde{\pi} | \Pi_t(\pi_t))) \right) \\
&= (1 + \beta)\eta \sum_{t=1}^{T} \sum_{\tilde{\pi} \in \Pi} \tilde{g}_t(\tilde{\pi} | \Pi_t(\pi_t)) - \frac{1 - \gamma}{\eta} \ln \left( \frac{\sum_{\tilde{\pi} \in \Pi} \exp(\eta \tilde{G}_T(\tilde{\pi}))}{\sum_{\tilde{\pi} \in \Pi} \exp(\eta \tilde{G}_0(\tilde{\pi}))} \right) \ (\because \text{Definition of } \omega_t(\tilde{\pi}), \tilde{G}_t(\tilde{\pi})) \\
&\leq (1 + \beta)\eta K \max_{\tilde{\pi} \in \Pi} \tilde{G}_T(\tilde{\pi}) + \frac{\ln K}{\eta} - \frac{1 - \gamma}{\eta} \ln \left( \sum_{\tilde{\pi} \in \Pi} \exp(\eta \tilde{G}_T(\tilde{\pi})) \right) \ (\because 1 - \gamma \leq 1 \text{ and } \tilde{G}_0(\tilde{\pi}) = 0) \\
&\leq -(1 - \gamma - (1 + \beta)\eta K) \max_{\tilde{\pi} \in \Pi} \tilde{G}_T(\tilde{\pi}) + \frac{\ln K}{\eta} \ (\because \text{Property of log-sum-exponential}) \\
&\leq -(1 - \gamma - (1 + \beta)\eta K) \max_{\tilde{\pi} \in \Pi} \sum_{t=1}^{T} g_t(\tilde{\pi}) + \frac{\ln(K\delta^{-1})}{\beta} + \frac{\ln K}{\eta},
\end{aligned}
\tag{22}
$$

where the last inequality holds due to the Lemma 2 and the initial assumption that $\gamma \leq \frac{1}{2}$ and $(1 + \beta)K\eta \leq \gamma$. Plugging Eq. 22 into Eq. 19, the following holds with probability $1 - \frac{\delta}{K}$ for all $\pi \in \Pi$:

$$R_\pi(T) \leq (\ell_{\max} - \ell_{\min}) \left( K\beta T + \gamma T + (1 + \beta)\eta KT + \frac{\ln(K\delta^{-1})}{\beta} + \frac{\ln K}{\eta} \right).$$

Since $\mathbf{Reg}(T) = \max R_\pi(T)$, this completes the proof by taking the union bound.

# E USEFUL PROPERTIES FOR THE PROOF OF THEOREM 4 AND THEOREM 5

## E.1 MONOTONIC STRUCTURE OF SINGLE-THRESHOLDED CONFORMAL SETS

We consider a class of conformal sets parameterized by a single threshold $\pi \in \Pi$, defined as Eq. 1. Within the model class, the miscoverage term $m_t(\pi)$ is monotonically non-increasing as the size of the corresponding conformal set $|\hat{C}_\pi(x_t)|$ increases. Equivalently, $m_t(\pi)$ is a monotonically non-decreasing function with respect to the threshold parameter $\pi$.

## E.2 PROPERTIES OF THE LOSS FUNCTION

The followings are the properties of the loss function

$$\ell_t(\pi; \alpha, c) = d_t(\pi; \alpha) + a_t(\pi; \alpha, c),$$

as defined in Eq. 5, Eq. 6, and Eq. 7. We henceforth write $\ell_t(\pi)$, $d_t(\pi)$, and $a_t(\pi)$ for simplicity, indicating the dependence on $\alpha$ and $c$ only when needed.

- By the definition of $d_t(\pi; \alpha)$ (Eq. 6), $d_t(\pi; \alpha) = \alpha + \alpha(1 - \alpha)$ for all $\pi \in \Pi_t^*$ and $d_t(\pi; a) = 1 - \alpha(1 - \alpha)$ for all $\pi \in (\Pi_t^*)^c$. Note that for $\alpha \in (0, 0.5)$,

$$\alpha + \alpha(1 - \alpha) < 1 - \alpha(1 - \alpha). \tag{23}$$

- For all $\pi_1, \pi_2 \in \Pi_t^*$ such that $\pi_1 \leq \pi_2$,

$$a_t(\pi_1) \geq a_t(\pi_2).$$

  Therefore, for all $\pi_1, \pi_2 \in \Pi_t^*$ such that $\pi_1 \leq \pi_2$,

$$\ell_t(\pi_1) \geq \ell_t(\pi_2). \tag{24}$$

- For all $\pi_1, \pi_2 \in (\Pi_t^*)^c$ such that $\pi_1 \leq \pi_2$,

$$a_t(\pi_1) \leq a_t(\pi_2).$$

  Therefore, for all $\pi_1, \pi_2 \in \Pi_t^*$ such that $\pi_1 \leq \pi_2$,

$$\ell_t(\pi_1) \leq \ell_t(\pi_2). \tag{25}$$

- Let $\ell_{t,0} := \max_{\pi \in \Pi_t^*} \ell_t(\pi)$ and $\ell_{t,1} := \min_{\pi \in (\Pi_t^*)^c} \ell_t(\pi)$. By letting $\pi_{t,0} := \min_{\pi \in \Pi_t^*} \pi = 0$ and $\pi_{t,1} := \min_{\pi \in (\Pi_t^*)^c} \pi$,

$$\ell_{t,0} = \ell_t(\pi_{t,0}) \ (\because \text{Eq. 24}),$$
$$\ell_{t,1} = \ell_t(\pi_{t,1}) \ (\because \text{Eq. 25}).$$

  Since controlling the miscoverage is the primary objective, the loss estimator will be most satisfactory when $\ell_{t,0} \leq \ell_{t,1}$ for all $t \in [T]$. Indeed,

$$\ell_{t,1} - \ell_{t,0} = (1 - 2\alpha)(1 - \alpha) + (a_t(\pi_{t,1}) - a_t(\pi_{t,0}))$$
$$\geq -(c\alpha)o(T)^{-1} + (c\alpha)o(T)^{-1}$$
$$= 0.$$

  Therefore,

$$\ell_t(\pi_1) \leq \ell_t(\pi_2), \ \forall \pi_1 \in \Pi_t^*, \ \forall \pi_2 \in (\Pi_t^*)^c. \tag{26}$$

- $\ell_{\max} = 1 - \alpha(1 - \alpha) - \frac{c\alpha}{1 + \alpha(1 - 2\alpha)} o(T)^{-1}$

- $\ell_{\min} = \alpha + \alpha(1 - \alpha) - \left(1 + \frac{1}{1 - \alpha}\right)(c\alpha)o(T)^{-1}$

### E.3 PROPERTIES OF THE NORMALIZED GAIN

The followings are properties of the normalized gain $g_t(\pi)$ (Eq. 8) and the pseudo-gain $\tilde{g}_t(\pi)$ (Eq. 12) terms under the partial feedback setting.

- Case 1 ($m_t(\pi_t) = 0$)
    1. $\pi \in \Pi_t^*$ and $\pi \leq \pi_t$: $\ell_t(\pi) \geq \ell_t(\pi_t) \Leftrightarrow g_t(\pi) \leq g_t(\pi_t)$ ($\because$ Eq. 24)
    2. $\pi \in \Pi_t^*$ and $\pi > \pi_t$: $\ell_t(\pi) \leq \ell_t(\pi_t) \Leftrightarrow g_t(\pi) \geq g_t(\pi_t)$. In addition, since $g_t(\pi) - $
    $$g_t(\pi_t) = \frac{c\alpha \frac{\pi^2 - \pi_t^2}{1-\alpha} o(T)^{-1}}{\ell_{\max} - \ell_{\min}} = \frac{c\alpha \frac{\pi^2 - \pi_t^2}{1-\alpha} o(T)^{-1}}{(1-2\alpha)(1-\alpha) + (1 + \frac{1}{1-\alpha} - \frac{1}{1+\alpha(1-2\alpha)})(c\alpha)o(T)^{-1}},$$
    $$g_t(\pi) = g_t(\pi_t) + o(T)^{-1} \ (\because \text{Eq. 24 \& Taylor expansion}).$$

    3. $\pi \in (\Pi_t^*)^c$: $\ell_t(\pi) \geq \ell_t(\pi_t) \Leftrightarrow g_t(\pi) \leq g_t(\pi_t)$ ($\because$ Eq. 26)
        - Additionally, for all $\pi \in (\Pi_t^*)^c$,
        $$\frac{g_t(\pi)}{g_t(\pi_t)} = \frac{\ell_{\max} - \ell_t(\pi)}{\ell_{\max} - \ell_t(\pi_t)}$$
        $$= \frac{c\alpha \left( \frac{o(T)^{-1}}{1+\alpha(1-2\alpha)\pi} - \frac{o(T)^{-1}}{1+\alpha(1-2\alpha)} \right)}{(1-2\alpha)(1-\alpha) + (1 + \frac{\pi_t^2}{1-\alpha} - \frac{1}{1+\alpha(1-2\alpha)})(c\alpha)o(T)^{-1}}$$
        $$= o(T)^{-1} \ (\because \text{Taylor expansion}).$$

    Therefore,
    $$g_t(\pi) = o(T)^{-1} g_t(\pi_t) \ \forall \pi \in (\Pi_t^*)^c.$$

    Therefore,
    $$g_t(\pi) \leq g_t(\pi_t) + o(T)^{-1} \ \forall \pi \in \Pi_t^*, \tag{27}$$
    $$g_t(\pi) \leq o(T)^{-1} g_t(\pi_t) \ \forall \pi \in (\Pi_t^*)^c.$$

- Case 2 ($m_t(\pi_t) = 1$)
    1. $\pi \in \Pi_t(\pi_t) \subset (\Pi_t^*)^c$: $\ell_t(\pi) \geq \ell_t(\pi_t) \Leftrightarrow g_t(\pi) \leq g_t(\pi_t)$ ($\because$ Eq. 25)
    2. $\pi \in \Pi_t(\pi_t)^c$:
    $$\tilde{g}_t(\pi) - g_t(\pi_t) = \frac{c\alpha \left( \frac{o(T)^{-1}}{1+\alpha(1-2\alpha)\pi} - \frac{o(T)^{-1}}{1+\alpha(1-2\alpha)\pi_t} \right)}{(1-2\alpha)(1-\alpha) + (1 + \frac{1}{1-\alpha} - \frac{1}{1+\alpha(1-2\alpha)})(c\alpha)o(T)^{-1}}$$
    $$= o(T)^{-1} \ (\because \text{Taylor expansion})$$

    Therefore,
    $$g_t(\pi) \leq g_t(\pi_t) \ \forall \pi \in \Pi(\pi_t),$$
    $$\tilde{g}_t(\pi) \leq g_t(\pi_t) + o(T)^{-1} \ \forall \pi \in \Pi(\pi_t)^c. \tag{28}$$

# F Proof of Theorem 4

The biased gain estimator $\tilde{g}_t(\pi|\Pi_t(\pi_t))$ of OCP-Unlock (Eq. 15) satisfies the following inequality.

- Case 1 ($m_t(\pi_t) = 0$)

$$
\begin{aligned}
\mathbb{E}_{\pi \sim p_t} \tilde{g}_t(\pi|\Pi_t(\pi_t)) &= \textstyle\sum_{\pi \in \Pi} p_t(\pi)\tilde{g}_t(\pi|\Pi_t(\pi_t)) \\
&= \textstyle\sum_{\pi \in \Pi_t^*} p_t(\pi)\tilde{g}_t(\pi|\Pi_t(\pi_t)) + \sum_{\pi \in (\Pi_t^*)^c} p_t(\pi)\tilde{g}_t(\pi|\Pi_t(\pi_t)) \\
&= \textstyle\sum_{\pi \in \Pi_t^*} p_t(\pi)\left\{\frac{g_t(\pi)}{\sum_{\tilde{\pi} \in \Pi_t^*} p_t(\tilde{\pi})} + \frac{\beta}{p_t(\pi)}\right\} + \sum_{\pi \in (\Pi_t^*)^c} p_t(\pi)\left\{\frac{\beta}{p_t(\pi)}\right\} \\
&\leq g_t(\pi_t) + o(T)^{-1} + K\beta \quad (\because \text{Eq. 27})
\end{aligned}
\tag{29}
$$

- Case 2 ($m_t(\pi_t) = 1$)

$$
\begin{aligned}
\mathbb{E}_{\pi \sim p_t} \tilde{g}_t(\pi|\Pi_t(\pi_t)) &= \textstyle\sum_{\pi \in \Pi} p_t(\pi)\tilde{g}_t(\pi|\Pi_t(\pi_t)) \\
&= \textstyle\sum_{\pi \in \Pi_t(\pi_t)} p_t(\pi)\tilde{g}_t(\pi|\Pi_t(\pi_t)) + \sum_{\pi \in \Pi_t(\pi_t)^c} p_t(\pi)\tilde{g}_t(\pi|\Pi_t(\pi_t)) \\
&= \textstyle\sum_{\pi \in \Pi_t(\pi_t)} p_t(\pi)\left\{\frac{g_t(\pi)}{\sum_{\tilde{\pi} \in \Pi_t(\pi_t)} p_t(\tilde{\pi})} + \frac{\beta}{p_t(\pi)}\right\} + \sum_{\pi \in \Pi_t(\pi_t)^c} p_t(\pi)\left\{\frac{\beta}{p_t(\pi)}\right\} \\
&\leq g_t(\pi_t) + K\beta \quad (\because \text{Eq. 28})
\end{aligned}
\tag{30}
$$

Combining Eq. 29 and Eq. 30, the following upper bound holds irrespective of the choice of $\pi_t$:

$$
\mathbb{E}_{\pi \sim p_t} \tilde{g}_t(\pi|\Pi_t(\pi_t)) \leq g_t(\pi_t) + o(T)^{-1} + K\beta. \tag{31}
$$

**Arm-wise High-probability Bound.** Before moving on to the regret analysis, we provide the following lemma, a variant of Lemma 2.

**Lemma 3.** *Let $\beta \leq 1$ and $\delta \in (0, 1)$. Then, for each $\pi \in \Pi$, the following holds with probability at least $1 - \delta$:*

$$
\sum_{t=1}^{T} g_t(\pi) \leq \sum_{t=1}^{T} \tilde{g}_t(\pi|\Pi_t(\pi_t)) + \frac{\ln(\delta^{-1})}{\beta}.
$$

*Proof.* **Step 1: Useful Decomposition.**
Let $\mathbb{E}_t$ be the expectation with respect to the distribution $p_t$, from which $\pi_t$ is sampled. For each $\pi \in \Pi$, the biased gain estimator can be decomposed as the following:

$$
\begin{aligned}
\tilde{g}_t(\pi \mid \Pi_t(\pi_t)) :=& \underbrace{\mathbb{1}(m_t(\pi_t) = 0)}_{\text{full unlocking}} \times (A) + \underbrace{\mathbb{1}(m_t(\pi_t) = 1)}_{\text{partial unlocking}} \times (B) \\
=& \underbrace{\mathbb{1}(m_t(\pi_t) = 0)}_{\text{full unlocking}} \times \{(A1) + (A2)\} + \underbrace{\mathbb{1}(m_t(\pi_t) = 1)}_{\text{partial unlocking}} \times \{(B1) + (B2)\} \\
=& \underbrace{\{\mathbb{1}(m_t(\pi_t)=0) \times (A1) + \mathbb{1}(m_t(\pi_t)=1) \times (B1)\}}_{(C1)} + \underbrace{\{\mathbb{1}(m_t(\pi_t)=0) \times (A2) + \mathbb{1}(m_t(\pi_t)=1) \times (B2)\}}_{(C2)},
\end{aligned}
$$

where

$$
(A) = \underbrace{\mathbb{1}(\pi \in \Pi_t^*)\frac{g_t(\pi)}{\sum_{\tilde{\pi} \in \Pi_t^*} p_t(\tilde{\pi})}}_{(A1)} + \underbrace{\frac{\beta}{p_t(\pi)}}_{(A2)}
$$

and

$$
(B) = \underbrace{\mathbb{1}(\pi \in \Pi_t(\pi_t))\frac{g_t(\pi)}{\sum_{\tilde{\pi} \in \Pi_t(\pi_t)} p_t(\tilde{\pi})}}_{(B1)} + \underbrace{\frac{\beta}{p_t(\pi)}}_{(B2)} .
$$

**Step 2: Show $\Delta_t(\pi) \leq 1$.**
Next, we show the following claims.

- Claim 1 $(m_t(\pi_t) = 0)$

$$\beta g_t(\pi) - \beta \times (A1) = \beta g_t(\pi) - \beta \frac{g_t(\pi)}{\sum_{\tilde{\pi} \in \Pi_t^*} p_t(\tilde{\pi})} \leq 1$$

- Claim 2 $(m_t(\pi_t) = 1)$

$$\beta g_t(\pi) - \beta \times (B1) = \beta g_t(\pi) - \beta \frac{g_t(\pi)}{\sum_{\tilde{\pi} \in \Pi_t(\pi_t)} p_t(\tilde{\pi})} \leq 1$$

Given $\pi \in \Pi$, Claim 1 and Claim 2 always hold irrespective of the choice of $\pi_t$ by the algorithm. Then, by letting $\Delta_t(\pi) := \beta g_t(\pi) - \beta \times (C1)$, $\Delta_t(\pi) \leq 1$ for all $\pi \in \Pi$.

**Step 3: Show $\mathbb{E} \exp \left[ \sum_{t=1}^{T} (\Delta_t(\pi) - \beta \times (C2)) \right] \leq 1$.**
Therefore, since (1) $\exp(x) \leq 1 + x + x^2$ for $x \leq 1$ and (2) $\Delta_t(\pi) \leq 1$, for $\beta \leq 1$, we have

$$\mathbb{E}_t \left[ \exp\left( \Delta_t(\pi) - \beta \times (C2) \right) \right] \leq \mathbb{E}_t \left[ \left( 1 + \Delta_t(\pi) + \Delta_t(\pi)^2 \right) \times \exp(-\beta \times (C2)) \right]$$

$$= \mathbb{E}_t \left[ \left( 1 + \Delta_t(\pi) + \Delta_t(\pi)^2 \right) \right] \times \exp \left( \frac{-\beta^2}{p_t(\pi)} \right).$$

Since $\pi$ is fixed, we consider each case where $\pi \in \Pi_t^*$ and $\pi \in (\Pi_t^*)^c$.

Now, our goal is to show that

$$\mathbb{E}_t \left[ \exp\left( \Delta_t(\pi) - \beta \times (C2) \right) \right] \leq 1 \ \forall t \in [T].$$

**Step 3-1: $\pi \in \Pi_t^*$.**
**Step 3-1-1: $\mathbb{E}_t \left[ \Delta_t(\pi) \right]$.**

$$\sum_{\tilde{\pi} \in \Pi} p_t(\tilde{\pi}) \Delta_t(\tilde{\pi}) = \beta g_t(\pi) - \sum_{\tilde{\pi} \in \Pi} p_t(\tilde{\pi}) \times \beta(C1)$$

$$= \beta g_t(\pi) - \beta \sum_{\tilde{\pi} \in \Pi_t^*} p_t(\tilde{\pi}) \frac{g_t(\pi)}{\sum_{\pi' \in \Pi_t^*} p_t(\pi')}$$

$$= 0$$

**Step 3-1-2: $\mathbb{E}_t \left[ \Delta_t(\pi)^2 \right]$.**

$$\sum_{\tilde{\pi} \in \Pi} p_t(\tilde{\pi}) \Delta_t(\tilde{\pi})^2 = \sum_{\tilde{\pi} \in \Pi_t^*} p_t(\tilde{\pi}) \Delta_t(\tilde{\pi})^2 + \sum_{\tilde{\pi} \in (\Pi_t^*)^c} p_t(\tilde{\pi}) \Delta_t(\tilde{\pi})^2$$

$$= (\beta g_t(\pi))^2 \sum_{\tilde{\pi} \in \Pi_t^*} p_t(\tilde{\pi}) \left( 1 - \frac{1}{\sum_{\pi' \in \Pi_t^*} p_t(\pi')} \right)^2 + (\beta g_t(\pi))^2 \sum_{\tilde{\pi} \in (\Pi_t^*)^c} p_t(\tilde{\pi})$$

$$\leq \frac{\beta^2}{\sum_{\tilde{\pi} \in \Pi_t^*} p_t(\tilde{\pi})} \ (\because g_t(\pi) \in [0, 1])$$

$$\leq \frac{\beta^2}{p_t(\pi)}$$

**Step 3-1-3: Combine.** Combining the above results and applying the fact that $1 + x \leq \exp(x)$,

$$\mathbb{E}_t \left[ \exp\left( \Delta_t(\pi) - \beta \times (C2) \right) \right] \leq \mathbb{E}_t \left[ \left( 1 + \Delta_t(\pi) + \Delta_t(\pi)^2 \right) \times \exp(-\beta \times (C2)) \right]$$

$$\leq \exp \left( -\frac{\beta^2}{p_t(\pi)} \right) \mathbb{E}_t \left[ \left( 1 + \Delta_t(\pi) + \Delta_t(\pi)^2 \right) \right]$$

$$\leq \exp \left( -\frac{\beta^2}{p_t(\pi)} \right) \left( 1 + \frac{\beta^2}{p_t(\pi)} \right)$$

$$\leq 1 \ (\because 1 + x \leq \exp(x)).$$

**Step 3-2:** $\pi \in (\Pi_t^*)^c$.

**Step 3-2-1:** $\mathbb{E}_t \Big[ \Delta_t(\pi) \Big]$.

$$\sum_{\tilde{\pi} \in \Pi} p_t(\tilde{\pi}) \Delta_t(\tilde{\pi}) = \beta g_t(\pi) - \sum_{\tilde{\pi} \in \Pi} p_t(\tilde{\pi}) \times \beta(C1)$$

$$= \beta g_t(\pi) - \beta \sum_{\tilde{\pi} \in (\Pi_t^*)^c} p_t(\tilde{\pi}) \frac{g_t(\pi)}{\sum_{\pi' \in \Pi_t(\tilde{\pi})} p_t(\pi')}$$

$$\leq \beta g_t(\pi) - \beta \sum_{\tilde{\pi} \in (\Pi_t^*)^c} p_t(\tilde{\pi}) \frac{g_t(\pi)}{\sum_{\pi' \in (\Pi_t^*)^c} p_t(\pi')}$$

$$= 0$$

**Step 3-2-2:** $\mathbb{E}_t \Big[ \Delta_t(\pi)^2 \Big]$.

$$\sum_{\tilde{\pi} \in \Pi} p_t(\tilde{\pi}) \Delta_t(\tilde{\pi})^2 = \sum_{\tilde{\pi} \in \Pi_t^*} p_t(\tilde{\pi}) \Delta_t(\tilde{\pi})^2 + \sum_{\tilde{\pi} \in (\Pi_t^*)^c} p_t(\tilde{\pi}) \Delta_t(\tilde{\pi})^2$$

$$= \sum_{\tilde{\pi} \in \Pi_t^*} p_t(\tilde{\pi}) \Delta_t(\tilde{\pi})^2 + \sum_{\tilde{\pi}:\pi \in \Pi_t(\tilde{\pi})} p_t(\tilde{\pi}) \Delta_t(\tilde{\pi})^2 + \sum_{\tilde{\pi}:\pi \in \Pi_t(\tilde{\pi})^c} p_t(\tilde{\pi}) \Delta_t(\tilde{\pi})^2$$

$$= \sum_{\tilde{\pi} \in \Pi_t^*} p_t(\tilde{\pi}) (\beta g_t(\pi))^2 + \sum_{\tilde{\pi}:\pi \in \Pi_t(\tilde{\pi})} p_t(\tilde{\pi}) (\beta g_t(\pi) - \frac{\beta g_t(\pi)}{\sum_{\pi' \in \Pi_t(\tilde{\pi})} p_t(\pi')})^2$$

$$+ \sum_{\tilde{\pi}:\pi \in \Pi_t(\tilde{\pi})^c} p_t(\tilde{\pi}) (\beta g_t(\pi))^2$$

$$\leq \sum_{\tilde{\pi} \in \Pi_t^*} p_t(\tilde{\pi}) (\beta g_t(\pi))^2 + \sum_{\tilde{\pi}:\pi \in \Pi_t(\tilde{\pi})} p_t(\tilde{\pi}) (\beta g_t(\pi) - \frac{\beta g_t(\pi)}{p_t(\tilde{\pi})})^2$$

$$+ \sum_{\tilde{\pi}:\pi \in \Pi_t(\tilde{\pi})^c} p_t(\tilde{\pi}) (\beta g_t(\pi))^2$$

$$\leq \frac{\beta^2}{p_t(\pi)} \ (\because g_t(\pi) \in [0,1])$$

**Step 3-2-3: Combine.** Combining the above results and applying the fact that $1 + x \leq \exp(x)$,

$$\mathbb{E}_t \left[ \exp\left( \Delta_t(\pi) - \beta \times (C2) \right) \right] \leq \mathbb{E}_t \Big[ (1 + \Delta_t(\pi) + \Delta_t(\pi)^2) \times \exp(-\beta \times (C2)) \Big]$$

$$\leq \exp(-\frac{\beta^2}{p_t(\pi)}) \mathbb{E}_t \Big[ (1 + \Delta_t(\pi) + \Delta_t(\pi)^2) \Big]$$

$$\leq \exp\left( -\frac{\beta^2}{p_t(\pi)} \right) \left( 1 + \frac{\beta^2}{p_t(\pi)} \right)$$

$$\leq 1 \ (\because 1 + x \leq \exp(x)).$$

By sequentially applying the double expectation rule for $t = T, \ldots, 1$,

$$\mathbb{E} \exp \left[ \sum_{t=1}^{T} (\Delta_t(\pi) - \beta \times (C2)) \right] \leq 1. \tag{32}$$

Moreover, from the Markov's inequality, we have $\mathbb{P}(X > \ln(1/\delta)) = \mathbb{P}(\exp(X) > 1/\delta) \leq \delta \mathbb{E} \exp(X)$. Combined with Eq. 32, for every $\pi \in \Pi$,

$$\beta \sum_{t=1}^{T} g_t(\pi) \leq \beta \sum_{t=1}^{T} \tilde{g}_t(\pi | \Pi_t(\pi_t)) + \ln(\delta^{-1})$$

with probability at least $1 - \delta$. This completes the proof. $\qquad \square$

**Proof of Theorem 4.** Now, we derive the regret bound of `OCP-Unlock` (Algorithm 4), which consists of three steps.

First, our goal is to show that, if $\gamma \leq \frac{1}{2}$ and $(1 + \beta)K\eta \leq \gamma$,

$$\mathbf{Reg}(T) \leq (\ell_{\max} - \ell_{\min}) \left( K\beta T + o(T)^{-1}T + \gamma T + (1+\beta)\eta KT + \frac{\ln(K\delta^{-1})}{\beta} + \frac{\ln K}{\eta} \right). \tag{33}$$

Irrespective of the hyperparameter setup, note that Theorem 4 always holds if $T < \sqrt{\frac{TK}{\ln K}} \ln(\delta^{-1}) + 5.15\sqrt{TK\ln K} + \sqrt{T}$. If $T \geq \sqrt{\frac{TK}{\ln K}} \ln(\delta^{-1}) + 5.15\sqrt{TK\ln K} + \sqrt{T}$, this implies that $\gamma \leq \frac{1}{2}$ and $(1+\beta)K\eta \leq \gamma$, which makes it suffice to show that Eq. 33 holds for $\gamma \leq \frac{1}{2}$ and $(1+\beta)K\eta \leq \gamma$.

**Step 1: Simple equalities.** For all $\pi \in \Pi$, the following equality holds:

$$
\begin{aligned}
R_\pi(T) &:= \sum_{t=1}^T \ell_t(\pi_t) - \sum_{t=1}^T \ell_t(\pi) \\
&= (\ell_{\max} - \ell_{\min}) \left( \sum_{t=1}^T g_t(\pi) - \sum_{t=1}^T g_t(\pi_t) \right) \ (\because \text{Eq. 8}) \\
&\leq (\ell_{\max} - \ell_{\min}) \left( K\beta T + o(T)^{-1}T + \sum_{t=1}^T g_t(\pi) - \sum_{t=1}^T \mathbb{E}_{\tilde{\pi} \sim p_t} \tilde{g}_t(\tilde{\pi} \mid \Pi_t(\pi_t)) \right) \ (\because \text{Eq. 31}).
\end{aligned}
$$
(34)

Using the definition of cumulant generating function and the relationship that $p_t = (1-\gamma)\omega_t + \gamma u$ where $\omega_t(\pi) = \frac{\exp\left(\eta \tilde{G}_{t-1}(\pi)\right)}{\sum_{\tilde{\pi} \in \Pi} \exp\left(\eta \tilde{G}_{t-1}(\tilde{\pi})\right)}$ and $u$ is the uniform distribution over $K$ arms, the following holds:

$$
\begin{aligned}
&-\mathbb{E}_{\tilde{\pi} \sim p_t} \tilde{g}_t(\tilde{\pi} \mid \Pi_t(\pi_t)) \\
&= -(1-\gamma)\mathbb{E}_{\tilde{\pi} \sim \omega_t} \tilde{g}_t(\tilde{\pi} \mid \Pi_t(\pi_t)) - \gamma \mathbb{E}_{\tilde{\pi} \sim u} \tilde{g}_t(\tilde{\pi} \mid \Pi_t(\pi_t)) \ (\because p_t = (1-\gamma)\omega_t + \gamma u) \\
&= (1-\gamma) \left[ \frac{1}{\eta} \ln \mathbb{E}_{\tilde{\pi} \sim \omega_t} \exp\left( \eta(\tilde{g}_t(\tilde{\pi} \mid \Pi_t(\pi_t)) - \mathbb{E}_{\tilde{\pi} \sim \omega_t} \tilde{g}_t(\tilde{\pi} \mid \Pi_t(\pi_t))) \right) - \right. \\
&\qquad\qquad \left. \frac{1}{\eta} \ln \mathbb{E}_{\tilde{\pi} \sim \omega_t} \exp(\eta \tilde{g}_t(\tilde{\pi} \mid \Pi_t(\pi_t))) \right] - \gamma \mathbb{E}_{\tilde{\pi} \sim u} \tilde{g}_t(\tilde{\pi} \mid \Pi_t(\pi_t)).
\end{aligned}
$$
(35)

**Step 2: Bounding the first term of Eq. 35.** First, we show that irrespective of the choice of $\pi_t$,

$$\eta \tilde{g}_t(\pi | \Pi_t(\pi_t)) \leq 1 \ \forall \pi \in \Pi.$$

- $m(\pi_t) = 0, \pi \in \Pi_t^*$

$$
\begin{aligned}
\eta \tilde{g}_t(\pi | \Pi_t(\pi_t)) &= \eta \left( \frac{g_t(\pi)}{\sum_{\tilde{\pi} \in \Pi_t^*} p_t(\tilde{\pi})} + \frac{\beta}{p_t(\pi)} \right) \\
&\leq \frac{\eta(g_t(\pi) + \beta)}{p_t(\pi)} \\
&\leq \frac{\eta(1+\beta)}{(1-\gamma)w_t(\pi) + \gamma \frac{1}{K}} \ (\because g_t(\pi) \in [0,1]) \\
&\leq 1 \ (\because (1+\beta)K\eta \leq \gamma)
\end{aligned}
$$

- $m(\pi_t) = 1, \pi \in \Pi_t(\pi_t)$

$$
\begin{aligned}
\eta \tilde{g}_t(\pi | \Pi_t(\pi_t)) &= \eta \left( \frac{g_t(\pi)}{\sum_{\tilde{\pi} \in \Pi_t(\pi_t)} p_t(\tilde{\pi})} + \frac{\beta}{p_t(\pi)} \right) \\
&\leq \frac{\eta(g_t(\pi) + \beta)}{p_t(\pi)} \\
&\leq \frac{\eta(1+\beta)}{(1-\gamma)w_t(\tilde{\pi}) + \gamma \frac{1}{K}} \ (\because g_t(\pi) \in [0,1]) \\
&\leq 1 \ (\because (1+\beta)K\eta \leq \gamma)
\end{aligned}
$$

- $m(\pi_t) = 0, \pi \in (\Pi_t^*)^c$; $m(\pi_t) = 1, \pi \in \Pi_t(\pi_t)^c$

$$\eta \tilde{g}_t(\pi|\Pi_t(\pi_t)) = \eta \frac{\beta}{p_t(\pi)}$$
$$\leq \frac{\eta(1+\beta)}{(1-\gamma)w_t(\pi) + \gamma\frac{1}{K}}$$
$$\leq 1 \; (\because (1+\beta)K\eta \leq \gamma)$$

Since (1) $\ln x \leq x - 1$, (2) $\exp(x) \leq 1 + x + x^2$ for all $x \leq 1$, and (3) $\eta\tilde{g}_t(\tilde{\pi}|\Pi_t(\pi_t)) \leq 1$,

$$
\begin{aligned}
&\ln \mathbb{E}_{\tilde{\pi}\sim\omega_t} \exp\left(\eta(\tilde{g}_t(\tilde{\pi} \mid \Pi_t(\pi_t)) - \mathbb{E}_{\tilde{\pi}\sim\omega_t}\tilde{g}_t(\tilde{\pi} \mid \Pi_t(\pi_t)))\right) \\
&= \ln \mathbb{E}_{\tilde{\pi}\sim\omega_t}\exp(\eta\tilde{g}_t(\tilde{\pi} \mid \Pi_t(\pi_t))) - \eta\mathbb{E}_{\tilde{\pi}\sim\omega_t}\tilde{g}_t(\tilde{\pi} \mid \Pi_t(\pi_t)) \\
&\leq \mathbb{E}_{\tilde{\pi}\sim\omega_t}\left\{\exp(\eta\tilde{g}_t(\tilde{\pi} \mid \Pi_t(\pi_t))) - 1 - \eta\tilde{g}_t(\tilde{\pi} \mid \Pi_t(\pi_t))\right\} \; (\because \ln x \leq x - 1) \\
&\leq \mathbb{E}_{\tilde{\pi}\sim\omega_t}\eta^2\tilde{g}_t(\tilde{\pi} \mid \Pi_t(\pi_t))^2 \; (\because \exp(x) \leq 1 + x + x^2) \\
&\leq \eta^2\frac{1+\beta}{1-\gamma}\sum_{\tilde{\pi}\in\Pi}\tilde{g}_t(\tilde{\pi}|\Pi_t(\pi_t)),
\end{aligned}
\tag{36}
$$

where the last inequality holds due to the following.

- $m(\pi_t) = 0$

$$\mathbb{E}_{\tilde{\pi}\sim\omega_t}\eta^2\tilde{g}_t(\tilde{\pi}|\Pi_t(\pi_t))^2$$

$$=\eta^2\left\{\sum_{\tilde{\pi}\in\Pi_t^*}\omega_t(\tilde{\pi})\left(\frac{g_t(\tilde{\pi})}{\sum_{\pi'\in\Pi_t^*}p_t(\pi')}+\frac{\beta}{p_t(\tilde{\pi})}\right)^2+\sum_{\tilde{\pi}\in(\Pi_t^*)^c}\omega_t(\tilde{\pi})\left(\frac{\beta}{p_t(\tilde{\pi})}\right)^2\right\}$$

$$\leq\eta^2\left\{\sum_{\tilde{\pi}\in\Pi_t^*}\omega_t(\tilde{\pi})\left(\frac{1+\beta}{p_t(\tilde{\pi})}\right)\tilde{g}_t(\tilde{\pi}|\Pi_t(\pi_t))+\sum_{\tilde{\pi}\in(\Pi_t^*)^c}\omega_t(\tilde{\pi})\left(\frac{1+\beta}{p_t(\tilde{\pi})}\right)\tilde{g}_t(\tilde{\pi}|\Pi_t(\pi_t))\right\} \; (\because g_t(\tilde{\pi})\in[0,1])$$

$$\leq\eta^2\frac{1+\beta}{1-\gamma}\left\{\sum_{\tilde{\pi}\in\Pi_t^*}\tilde{g}_t(\tilde{\pi}|\Pi_t(\pi_t))+\sum_{\tilde{\pi}\in(\Pi_t^*)^c}\tilde{g}_t(\tilde{\pi}|\Pi_t(\pi_t))\right\} \; (\because \frac{w_t(\tilde{\pi})}{p_t(\tilde{\pi})}\leq\frac{1}{1-\gamma})$$

$$=\eta^2\frac{1+\beta}{1-\gamma}\sum_{\tilde{\pi}\in\Pi}\tilde{g}_t(\tilde{\pi}|\Pi_t(\pi_t))$$

- $m(\pi_t) = 1$

$$\mathbb{E}_{\tilde{\pi}\sim\omega_t}\eta^2\tilde{g}_t(\tilde{\pi}|\Pi_t(\pi_t))^2$$

$$=\eta^2\left\{\sum_{\tilde{\pi}\in\Pi_t(\pi_t)}\omega_t(\tilde{\pi})\left(\frac{g_t(\tilde{\pi})}{\sum_{\pi'\in\Pi_t(\pi_t)}p_t(\pi')}+\frac{\beta}{p_t(\tilde{\pi})}\right)^2+\sum_{\tilde{\pi}\in\Pi(\pi_t)^c}\omega_t(\tilde{\pi})\left(\frac{\beta}{p_t(\tilde{\pi})}\right)^2\right\}$$

$$\leq\eta^2\left\{\sum_{\tilde{\pi}\in\Pi_t(\pi_t)}\omega_t(\tilde{\pi})\left(\frac{1+\beta}{p_t(\tilde{\pi})}\right)\tilde{g}_t(\tilde{\pi}|\Pi_t(\pi_t))+\sum_{\tilde{\pi}\in\Pi_t(\pi_t)^c}\omega_t(\tilde{\pi})\left(\frac{1+\beta}{p_t(\tilde{\pi})}\right)\tilde{g}_t(\tilde{\pi}|\Pi_t(\pi_t))\right\} \; (\because g_t(\tilde{\pi})\in[0,1])$$

$$\leq\eta^2\frac{1+\beta}{1-\gamma}\sum_{\tilde{\pi}\in\Pi}\tilde{g}_t(\tilde{\pi}|\Pi_t(\pi_t)) \; (\because \frac{w_t(\tilde{\pi})}{p_t(\tilde{\pi})}\leq\frac{1}{1-\gamma})$$

**Step 3: Summing.** Let $\tilde{G}_0(\tilde{\pi}) = 0$. Then, combining Eq. 35-Eq. 36 and summing over $t$ yield

$$-\sum_{t=1}^{T}\mathbb{E}_{\tilde{\pi}\sim p_t}\tilde{g}_t(\tilde{\pi}|\Pi_t(\pi_t))$$

$$\leq (1+\beta)\eta\sum_{t=1}^{T}\sum_{\tilde{\pi}\in\Pi}\tilde{g}_t(\tilde{\pi}|\Pi_t(\pi_t)) - \frac{1-\gamma}{\eta}\sum_{t=1}^{T}\ln\left(\sum_{\tilde{\pi}\in\Pi}w_t(\tilde{\pi})\exp(\eta\tilde{g}_t(\tilde{\pi}\mid\Pi_t(\pi_t)))\right)$$

$$= (1+\beta)\eta\sum_{t=1}^{T}\sum_{\tilde{\pi}\in\Pi}\tilde{g}_t(\tilde{\pi}|\Pi_t(\pi_t)) - \frac{1-\gamma}{\eta}\ln\left(\frac{\sum_{\tilde{\pi}\in\Pi}\exp(\eta\tilde{G}_T(\tilde{\pi}))}{\sum_{\tilde{\pi}\in\Pi}\exp(\eta\tilde{G}_0(\tilde{\pi}))}\right) \; (\because \text{Definition of } \omega_t(\tilde{\pi}), \tilde{G}_t(\tilde{\pi}))$$

$$\leq (1+\beta)\eta K \max_{\tilde{\pi}\in\Pi}\tilde{G}_T(\tilde{\pi}) + \frac{\ln K}{\eta} - \frac{1-\gamma}{\eta}\ln\left(\sum_{\tilde{\pi}\in\Pi}\exp(\eta\tilde{G}_T(\tilde{\pi}))\right) \; (\because 1-\gamma\leq 1 \text{ and } \tilde{G}_0(\tilde{\pi}) = 0)$$

$$\leq -(1-\gamma-(1+\beta)\eta K)\max_{\tilde{\pi}\in\Pi}\tilde{G}_T(\tilde{\pi}) + \frac{\ln K}{\eta} \; (\because \text{Property of log-sum-exponential})$$

$$\leq -(1-\gamma-(1+\beta)\eta K)\max_{\tilde{\pi}\in\Pi}\sum_{t=1}^{T}g_t(\tilde{\pi}) + \frac{\ln(K\delta^{-1})}{\beta} + \frac{\ln K}{\eta},$$

$$\tag{37}$$

where the last inequality holds due to the Lemma 3 and the initial assumption that $\gamma \leq \frac{1}{2}$ and $(1 + \beta)K\eta \leq \gamma$. Plugging Eq. 37 into Eq. 34, the following holds with probability $1 - \frac{\delta}{K}$ for all $\pi \in \Pi$:

$$R_\pi(T) \leq (\ell_{\max} - \ell_{\min}) \left( K\beta T + o(T)^{-1}T + \sum_{t=1}^{T} g_t(\pi) - \sum_{t=1}^{T} \mathbb{E}_{\tilde{\pi} \sim p_t} \tilde{g}_t(\tilde{\pi} \mid \Pi_t(\pi_t)) \right)$$

$$\leq (\ell_{\max} - \ell_{\min}) \left( K\beta T + o(T)^{-1}T + \gamma T + (1 + \beta)\eta KT + \frac{\ln(K\delta^{-1})}{\beta} + \frac{\ln K}{\eta} \right).$$

Since $\mathbf{Reg}(T) = \max_{\pi \in \Pi} R_\pi(T)$, this completes the proof by taking the union bound.

# G  PROOF OF THEOREM 5

The biased gain estimator $\tilde{g}_t(\pi|\Pi_t(\pi_t))$ of `OCP-Unlock+` (Eq. 11) satisfies the following inequality.

- Case 1 ($m_t(\pi_t) = 0$)

$$
\begin{aligned}
\mathbb{E}_{\pi \sim p_t} \tilde{g}_t(\pi|\Pi_t(\pi_t)) &= \textstyle\sum_{\pi \in \Pi} p_t(\pi)\tilde{g}_t(\pi|\Pi_t(\pi_t)) \\
&= \textstyle\sum_{\pi \in \Pi_t^*} p_t(\pi)\tilde{g}_t(\pi|\Pi_t(\pi_t)) + \sum_{\pi \in (\Pi_t^*)^c} p_t(\pi)\tilde{g}_t(\pi|\Pi_t(\pi_t)) \\
&= \textstyle\sum_{\pi \in \Pi_t^*} p_t(\pi)\left\{ \frac{g_t(\pi)+\beta}{\sum_{\tilde{\pi} \in \Pi_t^*} p_t(\tilde{\pi})} + \beta \right\} + \sum_{\pi \in (\Pi_t^*)^c} p_t(\pi)\left\{ g_t(\pi) + \frac{\beta}{\sum_{\tilde{\pi} \le \pi} p_t(\tilde{\pi})} \right\} \qquad (38) \\
&\le (1 + o(T)^{-1})g_t(\pi_t) + o(T)^{-1} + \left( 2 + \frac{\sum_{\tilde{\pi} \in (\Pi_t^*)^c} p_t(\tilde{\pi})}{\sum_{\tilde{\pi} \in \Pi_t^*} p_t(\tilde{\pi})} \right)\beta \quad (\because \text{Eq. 27})
\end{aligned}
$$

- Case 2 ($m_t(\pi_t) = 1$)

$$
\begin{aligned}
\mathbb{E}_{\pi \sim p_t} \tilde{g}_t(\pi|\Pi_t(\pi_t)) &= \textstyle\sum_{\pi \in \Pi} p_t(\pi)\tilde{g}_t(\pi|\Pi_t(\pi_t)) \\
&= \textstyle\sum_{\pi \in \Pi_t(\pi_t)} p_t(\pi)\tilde{g}_t(\pi|\Pi_t(\pi_t)) + \sum_{\pi \in \Pi_t(\pi_t)^c} p_t(\pi)\tilde{g}_t(\pi|\Pi_t(\pi_t)) \\
&= \textstyle\sum_{\pi \in \Pi_t(\pi_t)} p_t(\pi)\left\{ g_t(\pi) + \frac{\beta}{\sum_{\tilde{\pi} \le \pi} p_t(\tilde{\pi})} \right\} + \sum_{\pi \in \Pi_t(\pi_t)^c} p_t(\pi)\left\{ \tilde{g}_t(\pi) + \frac{\beta}{p_t(\pi)} + \beta \right\} \\
&\le g_t(\pi_t) + o(T)^{-1} + \left( 1 + |\Pi_t(\pi_t)^c| + \frac{\sum_{\tilde{\pi} \in \Pi_t(\pi_t)} p_t(\tilde{\pi})}{\sum_{\tilde{\pi} \le \pi} p_t(\tilde{\pi})} \right)\beta \quad (\because \text{Eq. 28})
\end{aligned}
$$
$$(39)$$

Combining Eq. 38 and Eq. 39, the following upper bound holds irrespective of the choice of $\pi_t$:

$$
\mathbb{E}_{\pi \sim p_t} \tilde{g}_t(\pi|\Pi_t(\pi_t)) \le (1 + o(T)^{-1})g_t(\pi_t) + o(T)^{-1}
$$
$$
+ \underbrace{\left( 1 + \{\mathbb{1}(m_t(\pi_t) = 0)1 + \mathbb{1}(m_t(\pi_t) = 1)|\Pi_t(\pi_t)^c|\} + \frac{\sum_{\tilde{\pi} \in (\Pi_t^*)^c} p_t(\tilde{\pi})}{\sum_{\tilde{\pi} \in \Pi_t^*} p_t(\tilde{\pi})} \right)}_{C_t} \beta.
$$
$$(40)$$

**Arm-wise High-probability Bound.**   Before moving on to the regret analysis, we provide the following lemma, a variant of Lemma 2.

**Lemma 4.** *Let $\beta \le 1$ and $\delta \in (0, 1)$. Then, for each $\pi \in \Pi$, the following holds with probability at least $1 - \delta$:*
$$
\sum_{t=1}^{T} g_t(\pi) \le \sum_{t=1}^{T} \tilde{g}_t(\pi|\Pi_t(\pi_t)) + \frac{\ln(\delta^{-1})}{\beta}.
$$

*Proof.* **Step 1: Useful Decomposition.**
Let $\mathbb{E}_t$ be the expectation with respect to the distribution $p_t$, from which $\pi_t$ is sampled. For each $\pi \in \Pi$, the biased gain estimator can be decomposed as the following:

$$
\begin{aligned}
\tilde{g}_t(\pi \mid \Pi_t(\pi_t)) &:= \underbrace{\mathbb{1}(m_t(\pi_t) = 0)}_{\text{full unlocking}} \times (A) + \underbrace{\mathbb{1}(m_t(\pi_t) = 1)}_{\text{partial unlocking}} \times (B) \\
&= \underbrace{\mathbb{1}(m_t(\pi_t) = 0)}_{\text{full unlocking}} \times \{(A1) + (A2)\} + \underbrace{\mathbb{1}(m_t(\pi_t) = 1)}_{\text{partial unlocking}} \times \{(B1) + (B2)\} \\
&= \underbrace{\{\mathbb{1}(m_t(\pi_t)=0) \times (A1) + \mathbb{1}(m_t(\pi_t)=1) \times (B1)\}}_{(C1)} + \underbrace{\{\mathbb{1}(m_t(\pi_t)=0) \times (A2) + \mathbb{1}(m_t(\pi_t)=1) \times (B2)\}}_{(C2)},
\end{aligned}
$$

where

$$
(A) = \underbrace{\mathbb{1}(\pi \in \Pi_t^*)\frac{g_t(\pi)}{\sum_{\tilde{\pi} \in \Pi_t^*} p_t(\tilde{\pi})} + \mathbb{1}(\pi \in (\Pi_t^*)^c)g_t(\pi)}_{(A1)} + \underbrace{\frac{\beta}{\mathbb{1}(\pi \in \Pi_t^*)\sum_{\tilde{\pi} \in \Pi_t^*} p_t(\tilde{\pi}) + \mathbb{1}(\pi \in (\Pi_t^*)^c)\sum_{\tilde{\pi} \le \pi} p_t(\tilde{\pi})} + \mathbb{1}(\pi \in \Pi_t^*)\beta}_{(A2)}
$$

and

$$
(B) = \underbrace{\mathbb{1}(\pi \in \Pi_t(\pi_t))g_t(\pi) + \mathbb{1}(\pi \in \Pi_t(\pi_t)^c)\tilde{g}_t(\pi)}_{(B1)} + \underbrace{\frac{\beta}{\mathbb{1}(\pi \in \Pi_t(\pi_t))\sum_{\tilde{\pi} \le \pi} p_t(\tilde{\pi}) + \mathbb{1}(\pi \in \Pi_t(\pi_t)^c)p_t(\pi)} + \mathbb{1}(\pi \in \Pi_t(\pi_t)^c)\beta}_{(B2)}.
$$

**Step 2: Show $\Delta_t(\pi) \le 1$.**
Next, we show the following claims.

- Claim 1 $(m_t(\pi_t) = 0)$

    - $\pi \in \Pi_t^*$

$$\beta g_t(\pi) - \beta \times (A1) = \beta g_t(\pi) - \beta \frac{g_t(\pi)}{\sum_{\tilde{\pi} \in \Pi_t^*} p_t(\tilde{\pi})} \le 0$$

    - $\pi \in (\Pi_t^*)^c$

$$\beta g_t(\pi) - \beta \times (A1) = \beta g_t(\pi) - \beta g_t(\pi) = 0$$

- Claim 2 $(m_t(\pi_t) = 1)$

    - $\pi \in \Pi_t(\pi_t)$

$$\beta g_t(\pi) - \beta \times (B1) = \beta g_t(\pi) - \beta g_t(\pi) = 0$$

    - $\pi \in \Pi_t(\pi_t)^c$

$$\beta g_t(\pi) - \beta \times (B1) = \beta g_t(\pi) - \beta \tilde{g}_t(\pi) \le 1 \ (\because \ \beta \le 1 \text{ and } g_t(\pi), \tilde{g}_t(\pi) \in [0, 1])$$

Given $\pi \in \Pi$, Claim 1 and Claim 2 always hold irrespective of the choice of $\pi_t$ by the algorithm. Then, by letting $\Delta_t(\pi) := \beta g_t(\pi) - \beta \times (C1)$, $\Delta_t(\pi) \le 1$ for all $\pi \in \Pi$.

**Step 3: Show $\mathbb{E} \exp \left[ \sum_{t=1}^{T} (\Delta_t(\pi) - \beta \times (C2)) \right] \le 1$.**

Therefore, since (1) $\exp(x) \le 1 + x + x^2$ for $x \le 1$ and (2) $\Delta_t(\pi) \le 1$, for $\beta \le 1$, we have

$$\mathbb{E}_t \left[ \exp \left( \Delta_t(\pi) - \beta \times (C2) \right) \right] \le \mathbb{E}_t \left[ (1 + \Delta_t(\pi) + \Delta_t(\pi)^2) \times \exp(-\beta \times (C2)) \right]$$

$$\le \left\{ \mathbb{1}(\pi \in \Pi_t^*) \exp \left( -\frac{\beta^2}{\sum_{\tilde{\pi} \in \Pi_t^*} p_t(\tilde{\pi})} - \beta^2 \right) + \mathbb{1}(\pi \in (\Pi_t^*)^c) \exp \left( -\frac{\beta^2}{\sum_{\tilde{\pi} \le \pi} p_t(\tilde{\pi})} \right) \right\}$$

$$\times \mathbb{E}_t \left[ (1 + \Delta_t(\pi) + \Delta_t(\pi)^2) \right].$$

Since $\pi$ is fixed, we consider each case where $\pi \in \Pi_t^*$ and $\pi \in (\Pi_t^*)^c$.

Now, our goal is to show that

$$\mathbb{E}_t \left[ \exp \left( \Delta_t(\pi) - \beta \times (C2) \right) \right] \le 1 \ \forall t \in [T].$$

**Step 3-1: $\pi \in \Pi_t^*$.**
**Step 3-1-1: $\mathbb{E}_t \left[ \Delta_t(\pi) \right]$.**

$$\sum_{\tilde{\pi} \in \Pi} p_t(\tilde{\pi}) \Delta_t(\tilde{\pi}) = \beta g_t(\pi) - \sum_{\tilde{\pi} \in \Pi} p_t(\tilde{\pi}) \times \beta(C1)$$

$$\le \beta g_t(\pi) - \beta \sum_{\tilde{\pi} \in \Pi_t^*} p_t(\tilde{\pi}) \frac{g_t(\pi)}{\sum_{\pi' \in \Pi_t^*} p_t(\pi')}$$

$$= 0$$

**Step 3-1-2: $\mathbb{E}_t \left[ \Delta_t(\pi)^2 \right]$.**

$$\sum_{\tilde{\pi} \in \Pi} p_t(\tilde{\pi}) \Delta_t(\tilde{\pi})^2 = \sum_{\tilde{\pi} \in \Pi_t^*} p_t(\tilde{\pi}) \Delta_t(\tilde{\pi})^2 + \sum_{\tilde{\pi} \in (\Pi_t^*)^c} p_t(\tilde{\pi}) \Delta_t(\tilde{\pi})^2$$

$$= (\beta g_t(\pi))^2 \sum_{\tilde{\pi} \in \Pi_t^*} p_t(\tilde{\pi}) \left( 1 - \frac{1}{\sum_{\tilde{\pi} \in \Pi_t^*} p_t(\tilde{\pi})} \right)^2 + \sum_{\tilde{\pi} \in (\Pi_t^*)^c} p_t(\tilde{\pi})(\beta g_t(\pi) - \beta \tilde{g}_t(\pi))^2$$

$$\le \frac{\beta^2}{\sum_{\tilde{\pi} \in \Pi_t^*} p_t(\tilde{\pi})} + \beta^2 \ (\because \ g_t(\pi), \tilde{g}_t(\pi) \in [0, 1])$$

**Step 3-1-3: Combine.** Combining the above results and applying the fact that $1 + x \leq \exp(x)$,

$$\mathbb{E}_t \left[ \exp \left( \Delta_t(\pi) - \beta \times (C2) \right) \right] \leq \mathbb{E}_t \left[ (1 + \Delta_t(\pi) + \Delta_t(\pi)^2) \times \exp(-\beta \times (C2)) \right]$$

$$= \exp \left( -\frac{\beta^2}{\sum_{\tilde{\pi} \in \Pi_t^*} p_t(\tilde{\pi})} - \beta^2 \right) \mathbb{E}_t \left[ (1 + \Delta_t(\pi) + \Delta_t(\pi)^2) \right]$$

$$\leq \exp \left( -\frac{\beta^2}{\sum_{\tilde{\pi} \in \Pi_t^*} p_t(\tilde{\pi})} - \beta^2 \right) \left( 1 + \frac{\beta^2}{\sum_{\tilde{\pi} \in \Pi_t^*} p_t(\tilde{\pi})} + \beta^2 \right)$$

$$\leq 1 \ (\because \ 1 + x \leq \exp(x)).$$

**Step 3-2:** $\pi \in (\Pi_t^*)^c$.

**Step 3-2-1:** $\mathbb{E}_t \left[ \Delta_t(\pi) \right]$.

$$\sum_{\tilde{\pi} \in \Pi} p_t(\tilde{\pi}) \Delta_t(\tilde{\pi}) = \sum_{\tilde{\pi} \in \Pi_t^*} p_t(\tilde{\pi}) \Delta_t(\tilde{\pi}) + \sum_{\tilde{\pi} \in (\Pi_t^*)^c} p_t(\tilde{\pi}) \Delta_t(\tilde{\pi})$$

$$= \sum_{\tilde{\pi} \in \Pi_t^*} p_t(\tilde{\pi}) \Delta_t(\tilde{\pi}) + \sum_{\tilde{\pi} : \pi \in \Pi_t(\tilde{\pi})} p_t(\tilde{\pi}) \Delta_t(\tilde{\pi}) + \sum_{\tilde{\pi} : \pi \in \Pi_t(\tilde{\pi})^c} p_t(\tilde{\pi}) \Delta_t(\tilde{\pi})$$

$$= \sum_{\tilde{\pi} \in \Pi_t^*} p_t(\tilde{\pi}) (\beta g_t(\pi) - \beta g_t(\pi)) + \sum_{\tilde{\pi} : \pi \in \Pi_t(\tilde{\pi})} p_t(\tilde{\pi}) (\beta g_t(\pi) - \beta g_t(\pi))$$

$$+ \sum_{\tilde{\pi} : \pi \in \Pi_t(\tilde{\pi})^c} p_t(\tilde{\pi}) (\beta g_t(\pi) - \beta \tilde{g}_t(\pi))$$

$$= \sum_{\tilde{\pi} : \pi \in \Pi_t(\tilde{\pi})^c} p_t(\tilde{\pi}) (\beta g_t(\pi) - \beta g_t(\pi)) \ (\because \text{Eq. 12})$$

$$= 0$$

**Step 3-2-2:** $\mathbb{E}_t \left[ \Delta_t(\pi)^2 \right]$.

$$\sum_{\tilde{\pi} \in \Pi} p_t(\tilde{\pi}) \Delta_t(\tilde{\pi})^2 = \sum_{\tilde{\pi} \in \Pi_t^*} p_t(\tilde{\pi}) \Delta_t(\tilde{\pi})^2 + \sum_{\tilde{\pi} \in (\Pi_t^*)^c} p_t(\tilde{\pi}) \Delta_t(\tilde{\pi})^2$$

$$= \sum_{\tilde{\pi} \in \Pi_t^*} p_t(\tilde{\pi}) \Delta_t(\tilde{\pi})^2 + \sum_{\tilde{\pi} : \pi \in \Pi_t(\tilde{\pi})} p_t(\tilde{\pi}) \Delta_t(\tilde{\pi})^2 + \sum_{\tilde{\pi} : \pi \in \Pi_t(\tilde{\pi})^c} p_t(\tilde{\pi}) \Delta_t(\tilde{\pi})^2$$

$$= \sum_{\tilde{\pi} \in \Pi_t^*} p_t(\tilde{\pi}) (\beta g_t(\pi) - \beta g_t(\pi))^2 + \sum_{\tilde{\pi} : \pi \in \Pi_t(\tilde{\pi})} p_t(\tilde{\pi}) (\beta g_t(\pi) - \beta g_t(\pi))^2$$

$$+ \sum_{\tilde{\pi} : \pi \in \Pi_t(\tilde{\pi})^c} p_t(\tilde{\pi}) (\beta g_t(\pi) - \beta \tilde{g}_t(\pi))^2$$

$$= \sum_{\tilde{\pi} : \pi \in \Pi_t(\tilde{\pi})^c} p_t(\tilde{\pi}) (\beta g_t(\pi) - \beta g_t(\pi))^2 \ (\because \text{Eq. 12})$$

$$= 0$$

**Step 3-2-3: Combine.** Combining the above results,

$$\mathbb{E}_t \left[ \exp \left( \Delta_t(\pi) - \beta \times (C2) \right) \right] \leq \mathbb{E}_t \left[ (1 + \Delta_t(\pi) + \Delta_t(\pi)^2) \times \exp(-\beta \times (C2)) \right]$$

$$= \exp(-\frac{\beta^2}{\sum_{\tilde{\pi} \leq \pi} p_t(\tilde{\pi})}) \mathbb{E}_t \left[ (1 + \Delta_t(\pi) + \Delta_t(\pi)^2) \right]$$

$$= \exp(-\frac{\beta^2}{\sum_{\tilde{\pi} \leq \pi} p_t(\tilde{\pi})})$$

$$\leq 1.$$

By sequentially applying the double expectation rule for $t = T, \ldots, 1$,

$$\mathbb{E} \exp \left[ \sum_{t=1}^{T} (\Delta_t(\pi) - \beta \times (C2)) \right] \leq 1. \tag{41}$$

Moreover, from the Markov's inequality, we have $\mathbb{P}(X > \ln(1/\delta)) = \mathbb{P}(\exp(X) > 1/\delta) \leq \delta \mathbb{E} \exp(X)$. Combined with Eq. 41, we have

$$\beta \sum_{t=1}^{T} g_t(\pi) \leq \beta \sum_{t=1}^{T} \tilde{g}_t(\pi | \Pi_t(\pi_t)) + \ln(\delta^{-1})$$

with probability at least $1 - \delta$. This completes the proof. $\qquad \square$

**Proof of the Theorem 5.** First, we re-express Eq. 40 for the simplicity of proof as the following:

$$-g_t(\pi_t) \leq \frac{1}{1+o(T)^{-1}} \Bigg\{ -\mathbb{E}_{\pi \sim p_t} \tilde{g}_t(\pi | \Pi_t(\pi_t)) + o(T)^{-1}$$
$$+ \underbrace{\left(1 + \{\mathbb{1}(m_t(\pi_t) = 0)1 + \mathbb{1}(m_t(\pi_t) = 1)|\Pi_t(\pi_t)^c|\} + \frac{\sum_{\tilde{\pi} \in (\Pi_t^*)^c} p_t(\tilde{\pi})}{\sum_{\tilde{\pi} \in \Pi_t^*} p_t(\tilde{\pi})}\right)}_{C_t} \beta \Bigg\}. \quad (42)$$

Now, we derive the regret bound of `OCP-Unlock+` (Algorithm 1), which consists of three steps. First, our goal is to show that, if $\gamma \leq \frac{1}{2}$ and $(1+2\beta)K\eta \leq \gamma$,

$$\mathbf{Reg}(T) \leq \frac{\ell_{\max} - \ell_{\min}}{1+o(T)^{-1}} \left( C\beta T + 2o(T)^{-1}T + \gamma T + (1+2\beta)\eta KT + \frac{\ln(K\delta^{-1})}{\beta} + \frac{\ln K}{\eta} \right), \quad (43)$$

where $C = \min\left(\frac{\sum_{t=1}^T C_t}{T}, K\right)$.

Irrespective of the hyperparameter setup, note that Theorem 5 always holds if $T < \sqrt{\frac{TK}{\ln K}} \ln(\delta^{-1}) + \sqrt{TC\ln K} + 4.15\sqrt{TK\ln K} + 2\sqrt{T}$. If $T \geq \sqrt{\frac{TK}{\ln K}} \ln(\delta^{-1}) + \sqrt{TC\ln K} + 4.15\sqrt{TK\ln K} + 2\sqrt{T}$, this implies that $\gamma \leq \frac{1}{2}$ and $(1+2\beta)K\eta \leq \gamma$, which makes it suffice to show that Eq. 43 holds for $\gamma \leq \frac{1}{2}$ and $(1+2\beta)K\eta \leq \gamma$.

**Step 1: Simple equalities.** For all $\pi \in \Pi$, the following equality holds:

$$R_\pi(T) := \sum_{t=1}^T \ell_t(\pi_t) - \sum_{t=1}^T \ell_t(\pi)$$

$$= (\ell_{\max} - \ell_{\min}) \left( \sum_{t=1}^T g_t(\pi) - \sum_{t=1}^T g_t(\pi_t) \right) \quad (\because \text{Eq. 8})$$

$$\leq \frac{\ell_{\max} - \ell_{\min}}{1+o(T)^{-1}} \left( C\beta T + o(T)^{-1}T + (1+o(T)^{-1}) \sum_{t=1}^T g_t(\pi) - \sum_{t=1}^T \mathbb{E}_{\tilde{\pi} \sim p_t} \tilde{g}_t(\tilde{\pi} | \Pi_t(\pi_t)) \right) \quad (\because \text{Eq. 42}). \quad (44)$$

Using the definition of cumulant generating function and the relationship that $p_t = (1-\gamma)\omega_t + \gamma u$ where $\omega_t(\pi) = \frac{\exp(\eta \tilde{G}_{t-1}(\pi))}{\sum_{\tilde{\pi} \in \Pi} \exp(\eta \tilde{G}_{t-1}(\tilde{\pi}))}$ and $u$ is the uniform distribution over $K$ arms, the following holds:

$$-\mathbb{E}_{\tilde{\pi} \sim p_t} \tilde{g}_t(\tilde{\pi} | \Pi_t(\pi_t))$$
$$= -(1-\gamma)\mathbb{E}_{\tilde{\pi} \sim \omega_t} \tilde{g}_t(\pi_i | \Pi_t(\pi_t)) - \gamma \mathbb{E}_{\tilde{\pi} \sim u} \tilde{g}_t(\tilde{\pi} | \Pi_t(\pi_t)) \quad (\because p_t = (1-\gamma)\omega_t + \gamma u)$$

$$= (1-\gamma)\left[ \frac{1}{\eta} \ln \mathbb{E}_{\tilde{\pi} \sim \omega_t} \exp\left(\eta(\tilde{g}_t(\tilde{\pi} | \Pi_t(\pi_t)) - \mathbb{E}_{\tilde{\pi} \sim \omega_t} \tilde{g}_t(\tilde{\pi} | \Pi_t(\pi_t)))\right) - \right.$$
$$\left. \frac{1}{\eta} \ln \mathbb{E}_{\tilde{\pi} \sim \omega_t} \exp(\eta \tilde{g}_t(\tilde{\pi} | \Pi_t(\pi_t))) \right] - \gamma \mathbb{E}_{\tilde{\pi} \sim u} \tilde{g}_t(\tilde{\pi} | \Pi_t(\pi_t)). \quad (45)$$

**Step 2: Bounding the first term of Eq. 45.** First, we show that irrespective of the choice of $\pi_t$,

$$\eta \tilde{g}_t(\pi | \Pi_t(\pi_t)) \leq 1 \,\forall \pi \in \Pi.$$

- $m(\pi_t) = 0, \pi \in \Pi_t^*$

$$\eta \tilde{g}_t(\pi | \Pi_t(\pi_t)) = \eta \left( \frac{g_t(\pi) + \beta}{\sum_{\tilde{\pi} \in \Pi_t^*} p_t(\tilde{\pi})} + \beta \right)$$

$$\leq \frac{\eta(1+2\beta)}{\sum_{\tilde{\pi} \in \Pi_t^*}((1-\gamma)w_t(\tilde{\pi}) + \gamma \frac{1}{K})} \quad (\because (1+2\beta)\eta K \leq \gamma \text{ and } g_t(\pi) \in [0,1])$$

$$\leq 1$$

- $m(\pi_t) = 0, \pi \in (\Pi_t^*)^c; m(\pi_t) = 1, \pi \in \Pi_t(\pi_t)$

$$
\begin{aligned}
\eta \tilde{g}_t(\pi | \Pi_t(\pi_t)) &= \eta \left( g_t(\pi) + \frac{\beta}{\sum_{\tilde{\pi} \leq \pi} p_t(\tilde{\pi})} \right) \\
&\leq \eta \left( \frac{g_t(\pi) + \beta}{\sum_{\tilde{\pi} \leq \pi} p_t(\tilde{\pi})} \right) \\
&\leq \frac{\eta(1 + \beta)}{\sum_{\tilde{\pi} \leq \pi}((1 - \gamma)w_t(\tilde{\pi}) + \gamma \frac{1}{K})} \quad (\because (1 + 2\beta)\eta K \leq \gamma \text{ and } g_t(\pi) \in [0, 1]) \\
&\leq 1
\end{aligned}
$$

- $m(\pi_t) = 1, \pi \in \Pi_t(\pi_t)^c$

$$
\begin{aligned}
\eta \tilde{g}_t(\pi | \Pi_t(\pi_t)) &= \eta \left( \tilde{g}_t(\pi) + \frac{\beta}{p_t(\pi)} \right) \\
&\leq \eta \left( \frac{g_t(\pi) + \beta}{p_t(\pi)} \right) \quad (\because \text{Eq. 8, Eq. 12, and Eq. 26}) \\
&\leq \frac{\eta(1 + \beta)}{(1 - \gamma)w_t(\pi) + \gamma \frac{1}{K}} \quad (\because (1 + 2\beta)\eta K \leq \gamma \text{ and } g_t(\pi) \in [0, 1]) \\
&\leq 1
\end{aligned}
$$

Since (1) $\ln x \leq x - 1$, (2) $\exp(x) \leq 1 + x + x^2$ for all $x \leq 1$, and (3) $\eta \tilde{g}_t(\tilde{\pi} | \Pi_t(\pi_t)) \leq 1$,

$$
\begin{aligned}
\ln \mathbb{E}_{\tilde{\pi} \sim \omega_t} & \exp \left( \eta(\tilde{g}_t(\tilde{\pi} | \Pi_t(\pi_t)) - \mathbb{E}_{\tilde{\pi} \sim \omega_t} \tilde{g}_t(\tilde{\pi} | \Pi_t(\pi_t))) \right) \\
&= \ln \mathbb{E}_{\tilde{\pi} \sim \omega_t} \exp(\eta \tilde{g}_t(\tilde{\pi} | \Pi_t(\pi_t))) - \eta \mathbb{E}_{\tilde{\pi} \sim \omega_t} \tilde{g}_t(\tilde{\pi} | \Pi_t(\pi_t)) \\
&\leq \mathbb{E}_{\tilde{\pi} \sim \omega_t} \{\exp(\eta \tilde{g}_t(\tilde{\pi} | \Pi_t(\pi_t))) - 1 - \eta \tilde{g}_t(\tilde{\pi} | \Pi_t(\pi_t))\} \quad (\because \ln x \leq x - 1) \\
&\leq \mathbb{E}_{\tilde{\pi} \sim \omega_t} \eta^2 \tilde{g}_t(\tilde{\pi} | \Pi_t(\pi_t))^2 \quad (\because \exp(x) \leq 1 + x + x^2) \\
&\leq \eta^2 \frac{1 + 2\beta}{1 - \gamma} \sum_{\tilde{\pi} \in \Pi} \tilde{g}_t(\pi | \Pi_t(\pi_t)),
\end{aligned}
\tag{46}
$$

where the last inequality holds due to the following.

- $m(\pi_t) = 0$

$$
\begin{aligned}
\mathbb{E}_{\tilde{\pi} \sim \omega_t} & \eta^2 \tilde{g}_t(\tilde{\pi} | \Pi_t(\pi_t))^2 \\
&= \eta^2 \left\{ \sum_{\tilde{\pi} \in \Pi_t^*} w_t(\tilde{\pi}) \left( \frac{g_t(\tilde{\pi}) + \beta}{\sum_{\pi' \in \Pi_t^*} p_t(\pi')} + \beta \right)^2 + \sum_{\tilde{\pi} \in (\Pi_t^*)^c} w_t(\tilde{\pi}) \left( g_t(\tilde{\pi}) + \frac{\beta}{\sum_{\pi' \leq \tilde{\pi}} p_t(\pi')} \right)^2 \right\} \\
&\leq \eta^2 \left\{ \sum_{\tilde{\pi} \in \Pi_t^*} w_t(\tilde{\pi}) \left( \frac{1 + 2\beta}{p_t(\tilde{\pi})} \right) \tilde{g}_t(\tilde{\pi} | \Pi_t(\pi_t)) + \sum_{\tilde{\pi} \in (\Pi_t^*)^c} w_t(\tilde{\pi}) \left( \frac{1 + 2\beta}{p_t(\tilde{\pi})} \right) \tilde{g}_t(\tilde{\pi} | \Pi_t(\pi_t)) \right\} \quad (\because g_t(\pi) \in [0,1]) \\
&\leq \eta^2 \frac{1 + \beta}{1 - \gamma} \sum_{\tilde{\pi} \in \Pi} \tilde{g}_t(\tilde{\pi} | \Pi_t(\pi_t)) \quad (\because \frac{w_t(\tilde{\pi})}{p_t(\tilde{\pi})} \leq \frac{1}{1 - \gamma})
\end{aligned}
$$

- $m(\pi_t) = 1$

$$
\begin{aligned}
\mathbb{E}_{\tilde{\pi} \sim \omega_t} & \eta^2 \tilde{g}_t(\tilde{\pi} | \Pi_t(\pi_t))^2 \\
&= \eta^2 \left\{ \sum_{\tilde{\pi} \in \Pi_t(\pi_t)} w_t(\tilde{\pi}) \left( g_t(\tilde{\pi}) + \frac{\beta}{\sum_{\pi' \leq \tilde{\pi}} p_t(\pi')} \right)^2 + \sum_{\tilde{\pi} \in \Pi_t(\pi_t)^c} w_t(\tilde{\pi}) \left( \tilde{g}_t(\tilde{\pi}) + \frac{\beta}{p_t(\tilde{\pi})} + \beta \right)^2 \right\} \\
&\leq \eta^2 \left\{ \sum_{\tilde{\pi} \in \Pi_t(\pi_t)} w_t(\tilde{\pi}) \left( \frac{1 + 2\beta}{p_t(\tilde{\pi})} \right) \tilde{g}_t(\tilde{\pi} | \Pi_t(\pi_t)) + \sum_{\tilde{\pi} \in \Pi_t(\pi_t)^c} w_t(\tilde{\pi}) \left( \frac{1 + 2\beta}{p_t(\tilde{\pi})} \right) \tilde{g}_t(\tilde{\pi} | \Pi_t(\pi_t)) \right\} \quad (\because g_t(\pi), \tilde{g}_t(\pi) \in [0,1]) \\
&\leq \eta^2 \frac{1 + \beta}{1 - \gamma} \sum_{\tilde{\pi} \in \Pi} \tilde{g}_t(\tilde{\pi} | \Pi_t(\pi_t)) \quad (\because \frac{w_t(\tilde{\pi})}{p_t(\tilde{\pi})} \leq \frac{1}{1 - \gamma})
\end{aligned}
$$

**Step 3: Summing.** Let $\tilde{G}_0(\tilde{\pi}) = 0$. Then, combining Eq. 45-Eq. 46 and summing over $t$ yield

$$-\sum_{t=1}^{T} \mathbb{E}_{\tilde{\pi} \sim p_t} \tilde{g}_t(\tilde{\pi} \mid \Pi_t(\pi_t))$$

$$\leq (1+2\beta)\eta \sum_{t=1}^{T} \sum_{\tilde{\pi} \in \Pi} \tilde{g}_t(\tilde{\pi}|\Pi_t(\pi_t)) - \frac{1-\gamma}{\eta} \sum_{t=1}^{T} \ln\left(\sum_{\tilde{\pi} \in \Pi} w_t(\tilde{\pi}) \exp(\eta \tilde{g}_t(\tilde{\pi} \mid \Pi_t(\pi_t)))\right)$$

$$= (1+2\beta)\eta \sum_{t=1}^{T} \sum_{\tilde{\pi} \in \Pi} \tilde{g}_t(\tilde{\pi}|\Pi_t(\pi_t)) - \frac{1-\gamma}{\eta} \ln\left(\frac{\sum_{\tilde{\pi} \in \Pi} \exp(\eta \tilde{G}_T(\tilde{\pi}))}{\sum_{\tilde{\pi} \in \Pi} \exp(\eta \tilde{G}_0(\tilde{\pi}))}\right) \ (\because \text{ Definition of } \omega_t(\tilde{\pi}), \tilde{G}_t(\tilde{\pi}))$$

$$\leq (1+2\beta)\eta K \max_{\tilde{\pi} \in \Pi} \tilde{G}_t(\tilde{\pi}) + \frac{\ln K}{\eta} - \frac{1-\gamma}{\eta} \ln\left(\sum_{\tilde{\pi} \in \Pi} \exp(\eta \tilde{G}_T(\tilde{\pi}))\right) \ (\because 1 - \gamma \leq 1 \text{ and } \tilde{G}_0(\tilde{\pi}) = 0)$$

$$\leq -(1 - r - (1+2\beta)\eta K) \max_{\tilde{\pi} \in \Pi} \tilde{G}_t(\tilde{\pi}) + \frac{\ln K}{\eta} \ (\because \text{ Property of log-sum-exponential})$$

$$\leq -(1 - r - (1+2\beta)\eta K) \max_{\tilde{\pi} \in \Pi} \sum_{t=1}^{T} g_t(\tilde{\pi}) + \frac{\ln(K\delta^{-1})}{\beta} + \frac{\ln K}{\eta},$$

(47)

where the last inequality holds due to the Lemma 4 and the initial assumption that $\gamma \leq \frac{1}{2}$ and $(1+2\beta)K\eta \leq \gamma$. Plugging Eq. 47 into Eq. 44, the following holds with probability $1 - \frac{\delta}{K}$ for all $\pi \in \Pi$:

$$R_\pi(T) \leq \frac{\ell_{\max} - \ell_{\min}}{1 + o(T)^{-1}} \left(C\beta T + o(T)^{-1}T + (1 + o(T)^{-1}) \sum_{t=1}^{T} g_t(\pi) - \sum_{t=1}^{T} \mathbb{E}_{\tilde{\pi} \sim p_t} \tilde{g}_t(\tilde{\pi} \mid \Pi_t(\pi_t))\right)$$

$$\leq \frac{\ell_{\max} - \ell_{\min}}{1 + o(T)^{-1}} \left(C\beta T + o(T)^{-1}T + o(T)^{-1}T + \gamma T + (1+2\beta)\eta KT + \frac{\ln(K\delta^{-1})}{\beta} + \frac{\ln K}{\eta}\right)$$

$$\leq (\ell_{\max} - \ell_{\min}) \left(C\beta T + 2o(T)^{-1}T + \gamma T + (1+2\beta)\eta KT + \frac{\ln(K\delta^{-1})}{\beta} + \frac{\ln K}{\eta}\right).$$

Since $\mathbf{Reg}(T) = \max R_\pi(T)$, this completes the proof by taking the union bound.

# H    ADDITIONAL EXPERIMENTAL DETAIL

## H.1    EXPERIMENT 1: IMAGENET

We use a pretrained ResNet-18 (He et al., 2015) as an image classifier, where $p_\theta(y|x)$ is its estimated probability of label $y$ given input $x$. In the i.i.d. setting, we use $f_t(x_t, y) := p_\theta(y|x_t)^{1/3}$ as a scoring function for all $y \in \mathcal{Y}$ and $t \in [T]$. Meanwhile, we simulate the non-i.i.d. setting by changing the functional form of the scoring function for every $10,000$ time steps. Specifically, unlike the i.i.d. setting, we define the scoring function using different exponents of the estimated probability $p_\theta(y|x_t)^{\alpha_t}$ for all $y \in \mathcal{Y}$, where

$$\alpha_t = \begin{cases} 1/6 & t \in [1, 10,000] \\ 1/4 & t \in (10,000, 20,000] \\ 1/2 & t \in (20,000, 30,000] \\ 1/1.2 & t \in (30,000, 40,000] \\ 1/3 & t \in (40,000, 50,000]. \end{cases}$$

Note that the scoring function concentrates to 1 as $\alpha_t$ decreases to 0, simulating the non-i.i.d. setting where the range of the scoring function and the difficulty evolve over time.

## H.2    EXPERIMENT 2: AIRFOIL SELF-NOISE

We use a linear regression model $\hat{y}(x)$, which is trained using the recursive least squares algorithm. For both i.i.d. and non-i.i.d. settings, we use the following normalized residual score as a scoring function:

$$f_t(x_t, y) = \frac{u - |y - \hat{y}(x_t)|}{u - l},$$

where $l$ and $u$ denote lower and upper bounds on the residuals $|y - \hat{y}(x)|$, respectively. Unlike **Experiment 1**, we simulate the non-i.i.d. setting by changing the input distribution after the first one-third of the datastream, as illustrated in Section 5.

For all subsequent experiments, unless otherwise specified, we use the same i.i.d. and non-i.i.d. setups for ImageNet as in **Experiment 1**, and the same i.i.d. and non-i.i.d. setups for Airfoil Self-Noise as in **Experiment 2**.

## H.3 CHOICE OF HYPERPARAMETER $c$

For all experiments, we set $c = 40$ in the loss function $\ell_t(\pi; \alpha, c) = d_t(\pi; \alpha) + a_t(\pi; \alpha, c)$, as defined in Eq. 5, Eq. 6, and Eq. 7.

We report the long-run miscoverage $\textbf{MC}(T)$ and inefficiency $\textbf{Ineff}(T)$, averaged over 50 independent trials, across various $T$ and $\alpha$ settings for both classification and regression tasks. For the discretization level, we use $K = 200$ for ImageNet and $K = 20$ for Airfoil Self-Noise. The results show that our method consistently achieves decent performance for a fixed $c$, uniformly across different datasets, data generating processes, and $T$, $\alpha$ settings (Table 4 and Table 5).

Table 4: Long-run miscoverage $\textbf{MC}(T)$ and inefficiency $\textbf{Ineff}(T)$ on ImageNet for different choices of $T$ and target miscoverage level $\alpha$, under the i.i.d. and non-i.i.d. settings

| $T$ | Metric | i.i.d. | | | | Non-i.i.d. | | | |
|---|---|---|---|---|---|---|---|---|---|
| | | $\alpha=0.1$ | $\alpha=0.2$ | $\alpha=0.3$ | $\alpha=0.4$ | $\alpha=0.1$ | $\alpha=0.2$ | $\alpha=0.3$ | $\alpha=0.4$ |
| 50,000 | $\textbf{MC}(T)$ | 0.131 | 0.230 | 0.303 | 0.340 | 0.156 | 0.250 | 0.318 | 0.352 |
| | $\textbf{Ineff}(T)$ | 46.73 | 16.77 | 8.47 | 6.25 | 56.32 | 27.30 | 18.08 | 14.88 |
| 60,000 | $\textbf{MC}(T)$ | 0.123 | 0.226 | 0.307 | 0.346 | 0.147 | 0.249 | 0.322 | 0.357 |
| | $\textbf{Ineff}(T)$ | 51.29 | 17.69 | 7.99 | 5.77 | 61.20 | 27.41 | 17.39 | 13.88 |
| 70,000 | $\textbf{MC}(T)$ | 0.116 | 0.222 | 0.310 | 0.350 | 0.141 | 0.246 | 0.324 | 0.361 |
| | $\textbf{Ineff}(T)$ | 54.90 | 18.25 | 7.70 | 5.55 | 64.84 | 27.69 | 16.54 | 13.23 |

Table 5: Long-run miscoverage $\textbf{MC}(T)$ and inefficiency $\textbf{Ineff}(T)$ on Airfoil Self-Noise for different choices of $T$ and target miscoverage level $\alpha$, under the i.i.d. and non-i.i.d. settings

| $T$ | Metric | i.i.d. | | | | Non-i.i.d. | | | |
|---|---|---|---|---|---|---|---|---|---|
| | | $\alpha=0.1$ | $\alpha=0.2$ | $\alpha=0.3$ | $\alpha=0.4$ | $\alpha=0.1$ | $\alpha=0.2$ | $\alpha=0.3$ | $\alpha=0.4$ |
| 30,000 | $\textbf{MC}(T)$ | 0.062 | 0.129 | 0.193 | 0.243 | 0.084 | 0.174 | 0.263 | 0.329 |
| | $\textbf{Ineff}(T)$ | 13.29 | 9.62 | 7.84 | 7.12 | 15.37 | 10.90 | 8.82 | 7.51 |
| 40,000 | $\textbf{MC}(T)$ | 0.057 | 0.123 | 0.191 | 0.243 | 0.077 | 0.168 | 0.261 | 0.330 |
| | $\textbf{Ineff}(T)$ | 13.41 | 9.70 | 7.84 | 7.07 | 15.52 | 10.92 | 8.87 | 7.46 |
| 50,000 | $\textbf{MC}(T)$ | 0.054 | 0.119 | 0.189 | 0.244 | 0.072 | 0.162 | 0.258 | 0.332 |
| | $\textbf{Ineff}(T)$ | 13.53 | 9.69 | 7.84 | 7.02 | 15.83 | 11.02 | 8.85 | 7.44 |

# I   EXPERIMENT 3: ABLATION STUDY ON THE LEVEL OF DISCRETIZATION $K$

We provide an ablation study on the discretization level $K \in \mathbb{N}$ in implementing our bandit-based algorithms. In terms of theoretical guarantees, our regret bound is proportional to $K$, so increasing $K$ yields a finer threshold grid at the cost of a larger regret bound. Meanwhile, empirical results suggest that a finer discretization of the hypothesis space is necessary to keep conformal sets compact. To quantify this trade-off, we conduct an ablation study on both classification and regression tasks by varying $K$ while holding all other parameters fixed. For ImageNet, we set the horizon to $T = 50{,}000$, and the target miscoverage level to $\alpha = 0.15$. For Airfoil Self-Noise, we set the horizon to $T = 50{,}000$, and the target miscoverage level to $\alpha = 0.1$.

We report the long-run miscoverage $\mathbf{MC}(T)$ and inefficiency $\mathbf{Ineff}(T)$, averaged over 50 independent trials for classification and regression tasks, respectively (Table 6 and Table 7).

Table 6: Ablation study on the level of discretization ($K$) on ImageNet

| Method | Metric | i.i.d. | | | Non-i.i.d. | | |
|---|---|---|---|---|---|---|---|
| | | $K{=}180$ | $K{=}200$ | $K{=}220$ | $K{=}180$ | $K{=}200$ | $K{=}220$ |
| OCP-Bandit | $\mathbf{MC}(T)$ | 0.212 | 0.214 | 0.215 | 0.228 | 0.228 | 0.229 |
| | $\mathbf{Ineff}(T)$ | 21.02 | 20.25 | 19.84 | 33.16 | 32.77 | 32.47 |
| OCP-Unlock | $\mathbf{MC}(T)$ | 0.209 | 0.211 | 0.212 | 0.225 | 0.226 | 0.227 |
| | $\mathbf{Ineff}(T)$ | 21.65 | 21.05 | 20.45 | 34.54 | 33.76 | 32.86 |
| OCP-Unlock+ | $\mathbf{MC}(T)$ | 0.172 | 0.173 | 0.176 | 0.199 | 0.200 | 0.201 |
| | $\mathbf{Ineff}(T)$ | 31.21 | 29.85 | 28.52 | 40.45 | 39.11 | 38.17 |

Table 7: Ablation study on the level of discretization ($K$) on Airfoil Self-Noise

| Method | Metric | i.i.d. | | | Non-i.i.d. | | |
|---|---|---|---|---|---|---|---|
| | | $K{=}10$ | $K{=}20$ | $K{=}30$ | $K{=}10$ | $K{=}20$ | $K{=}30$ |
| OCP-Bandit | $\mathbf{MC}(T)$ | 0.056 | 0.064 | 0.070 | 0.071 | 0.084 | 0.092 |
| | $\mathbf{Ineff}(T)$ | 13.97 | 14.06 | 14.03 | 16.54 | 15.96 | 15.79 |
| OCP-Unlock | $\mathbf{MC}(T)$ | 0.056 | 0.064 | 0.069 | 0.070 | 0.083 | 0.090 |
| | $\mathbf{Ineff}(T)$ | 13.71 | 13.81 | 13.94 | 16.49 | 15.97 | 15.79 |
| OCP-Unlock+ | $\mathbf{MC}(T)$ | 0.050 | 0.054 | 0.056 | 0.064 | 0.072 | 0.076 |
| | $\mathbf{Ineff}(T)$ | 13.28 | 13.53 | 13.60 | 16.37 | 15.83 | 15.74 |

While it is theoretically expected that a longer time horizon is required to avoid undercoverage when the hypothesis space is more finely discretized, the goal of this experiment is not to maintain performance across different values of $K$, but rather to study the effect of varying $K$ while keeping the other parameters $(\alpha, T)$ fixed.

## J   EXPERIMENT 4: ABLATION STUDY ON OCP-UNLOCK+ AND ITS VARIANTS

We conduct an ablation study comparing three variants—OCP-Bandit, OCP-Unlock, and OCP-Unlock+—in terms of (i) miscoverage rate $\mathbf{MC}(t)$ and (ii) inefficiency $\mathbf{Ineff}(t)$ over time $t \in [T]$. In each plot, the solid curves show the average across independent runs, and the shaded regions indicate the min–max range. Overall, OCP-Unlock+ achieves tighter coverage under both the i.i.d. and non-i.i.d. settings. This improvement comes with a mild cost: OCP-Unlock+ tends to output slightly larger prediction sets, yielding moderately higher $\mathbf{Ineff}(t)$.

For ImageNet, we set the horizon to $T = 50,000$, the target miscoverage level to $\alpha = 0.15$, and the discretization level to $K = 200$. For Airfoil, we set the horizon to $T = 50,000$, the target miscoverage level to $\alpha = 0.1$, and the discretization level to $K = 20$.

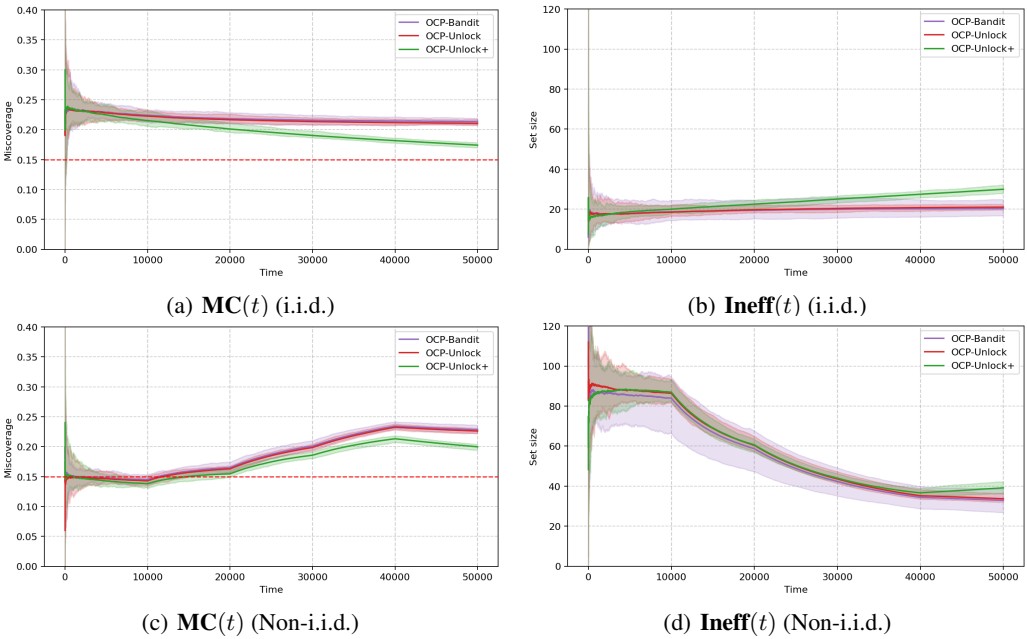

(a) $\mathbf{MC}(t)$ (i.i.d.)        (b) $\mathbf{Ineff}(t)$ (i.i.d.)

(c) $\mathbf{MC}(t)$ (Non-i.i.d.)        (d) $\mathbf{Ineff}(t)$ (Non-i.i.d.)

Figure 5: Ablation study on ImageNet: Long-run miscoverage $\mathbf{MC}(t)$ and inefficiency $\mathbf{Ineff}(t)$ as a function of time $t \in [T]$, under the i.i.d. (top) and non-i.i.d. (bottom) settings. The solid curves denote the average over 50 independent runs, and the shaded regions indicate the min–max range.

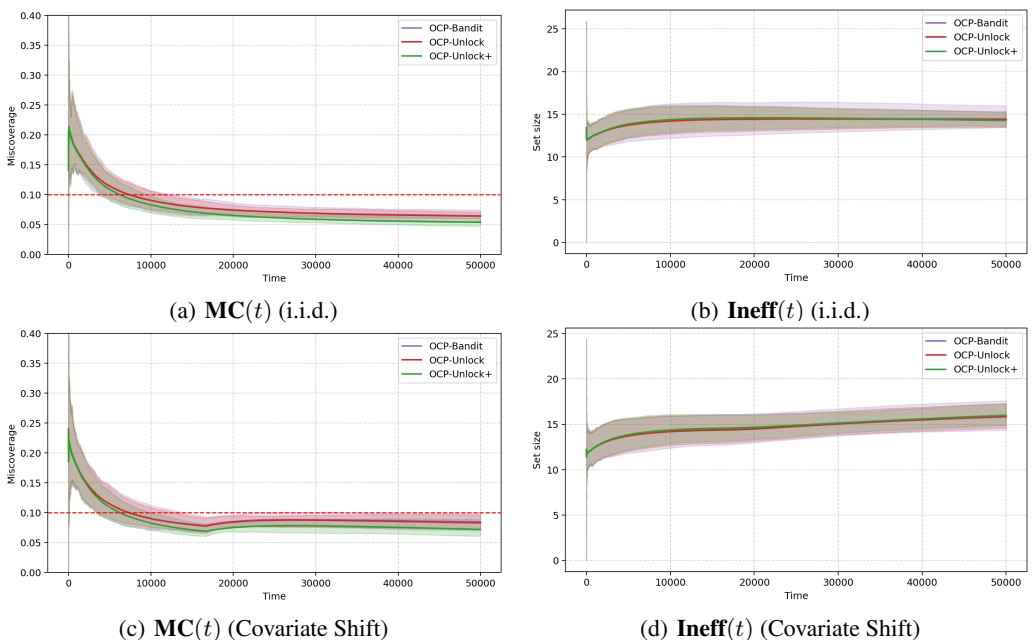

(a) $\mathbf{MC}(t)$ (i.i.d.)  (b) $\mathbf{Ineff}(t)$ (i.i.d.)

(c) $\mathbf{MC}(t)$ (Covariate Shift)  (d) $\mathbf{Ineff}(t)$ (Covariate Shift)

Figure 6: Ablation study on Airfoil Self-Noise: Long-run miscoverage $\mathbf{MC}(t)$ and inefficiency $\mathbf{Ineff}(t)$ as a function of time $t \in [T]$, under the i.i.d. (top) and covariate shift (bottom) settings. The solid curve denotes the average over 50 independent runs, and the shaded region indicates the min–max range.

## K  EXPERIMENT 5: SENSITIVITY OF OCP-UNLOCK+ TO $\alpha$

We study the sensitivity of OCP-Unlock+ to the target miscoverage level $\alpha$. For both ImageNet and Airfoil Self-Noise datasets, we use $\alpha \in \{0.1, 0.2, 0.3, 0.4\}$.

For ImageNet, we set the horizon to $T = 50,000$ and the discretization level to $K = 200$. For Airfoil Self-Noise, we set the horizon to $T = 50,000$ and the discretization level to $K = 20$.

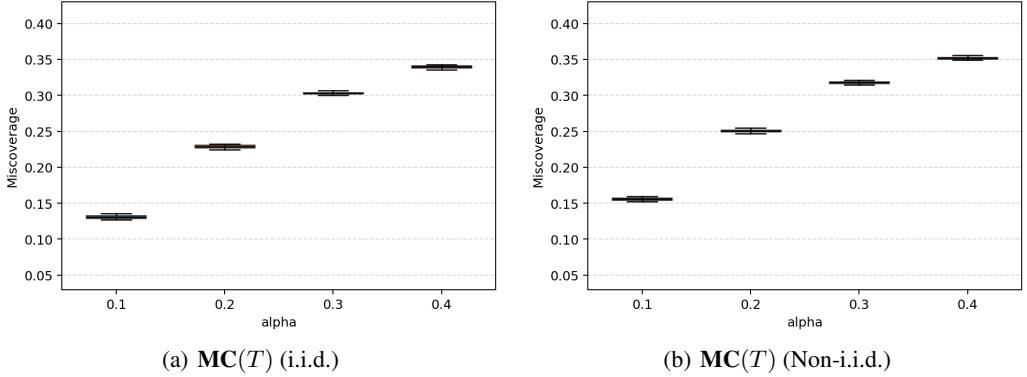

(a) $\mathbf{MC}(T)$ (i.i.d.)             (b) $\mathbf{MC}(T)$ (Non-i.i.d.)

Figure 7: Sensitivity of OCP-Unlock+ to $\alpha$ on ImageNet. Box plots summarize the long-run miscoverage $\mathbf{MC}(T)$ across 50 independent runs for $\alpha \in \{0.1, 0.2, 0.3, 0.4\}$ under the i.i.d. (left) and non-i.i.d. (right) settings. Each box shows the interquartile range with the median marked inside, while the whiskers indicate the spread of the results.

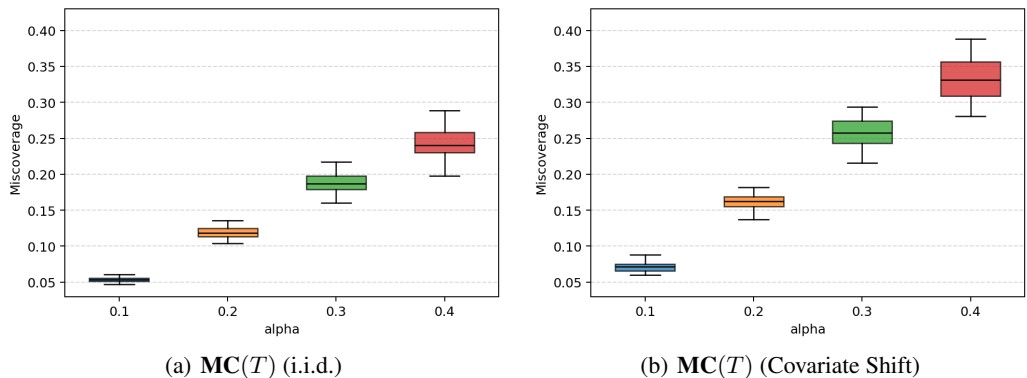

(a) $\mathbf{MC}(T)$ (i.i.d.)             (b) $\mathbf{MC}(T)$ (Covariate Shift)

Figure 8: Sensitivity of OCP-Unlock+ to $\alpha$ on Airfoil Self-Noise. Box plots summarize the long-run miscoverage $\mathbf{MC}(T)$ across 50 independent runs for $\alpha \in \{0.1, 0.2, 0.3, 0.4\}$ under the i.i.d. (left) and covariate shift (right) settings. Each box shows the interquartile range with the median marked inside, while the whiskers indicate the spread of the results.

## L    EXPERIMENT 6: EMPIRICAL ANALYSIS ON $C_{\mathbf{MC}}(T)$

We empirically analyze the additional term $C_{\mathbf{MC}}(T)$ that appears in the upper bound on the deviation of the miscoverage rate $\mathbf{MC}(T)$ from $\alpha$ (Lemma 1) to validate the efficacy of `OCP-Unlock+`.

Following the same experimental setup as in **Experiment 5** (Appendix K), we report the $C_{\mathbf{MC}}(T)$ term, averaged over 50 independent runs (Table 8 and Table 9).

Table 8: $C_{\mathbf{MC}}(T)$ on ImageNet for different choices of target miscoverage level $\alpha$, under the i.i.d. and non-i.i.d. settings

| $T$ | i.i.d. | | | | Non-i.i.d. | | | |
|---|---|---|---|---|---|---|---|---|
| | $\alpha=0.1$ | $\alpha=0.2$ | $\alpha=0.3$ | $\alpha=0.4$ | $\alpha=0.1$ | $\alpha=0.2$ | $\alpha=0.3$ | $\alpha=0.4$ |
| 50,000 | 0.062 | 0.161 | 0.304 | 0.469 | 0.091 | 0.186 | 0.318 | 0.476 |
| 60,000 | 0.050 | 0.149 | 0.299 | 0.461 | 0.079 | 0.178 | 0.312 | 0.466 |
| 70,000 | 0.040 | 0.139 | 0.295 | 0.453 | 0.071 | 0.170 | 0.307 | 0.459 |

Table 9: $C_{\mathbf{MC}}(T)$ on Airfoil Self-Noise for different choices of target miscoverage level $\alpha$, under the i.i.d. and non-i.i.d. settings

| $T$ | i.i.d. | | | | Non-i.i.d. | | | |
|---|---|---|---|---|---|---|---|---|
| | $\alpha=0.1$ | $\alpha=0.2$ | $\alpha=0.3$ | $\alpha=0.4$ | $\alpha=0.1$ | $\alpha=0.2$ | $\alpha=0.3$ | $\alpha=0.4$ |
| 30,000 | 0.020 | 0.131 | 0.291 | 0.500 | 0.025 | 0.130 | 0.276 | 0.451 |
| 40,000 | 0.010 | 0.115 | 0.267 | 0.465 | 0.014 | 0.115 | 0.258 | 0.425 |
| 50,000 | 0.002 | 0.099 | 0.248 | 0.438 | 0.006 | 0.102 | 0.243 | 0.406 |

