# OpenReview forum: "Online Conformal Prediction with Adversarial Semi-bandit Feedback via Regret Minimization"
_ICLR.cc/2026/Conference — ICLR 2026 Poster_

### Official Review · Reviewer_7ZrS · 2025-10-30

**Soundness:** 2
**Presentation:** 2
**Contribution:** 2
**Rating:** 4
**Confidence:** 4

**Summary:**

The paper tackles online conformal prediction with semi-bandit feedback and adaptive adversaries by reducing threshold selection to an adversarial bandit problem. It designs a loss that trades off deviation from a target miscoverage level with efficiency (tuned by λ) and proves a conversion from bandit regret to miscoverage guarantees. Implementations with EXP3.P and an “unlocking” variant that reuses feedback via monotonicity provide high-probability bounds, and experiments on classification and regression show accurate coverage tracking with compact prediction sets, including under covariate shift.

**Strengths:**

The paper introduces an adversarial semi bandit formulation for online conformal prediction and reduces threshold selection to an adversarial bandit with a transparent bridge from regret control to miscoverage control, together with an unlocking mechanism that exploits monotonicity to reuse feedback. It offers high probability guarantees and a clear analysis pipeline from loss design to regret bounds to miscoverage guarantees, implemented with practical EXP3.P algorithms. Experiments on classification and regression show close tracking of target miscoverage with strong sample efficiency, consistent with the theory. The writing is clear, and the modular design lets other bandit learners plug in, increasing impact in retrieval, selective prediction, and human in the loop systems.

**Weaknesses:**

See details in Questions.

**Questions:**

1. In both experiments, the choice of the hyperparameter $\lambda$ is not specified. How is $\lambda$ selected, and how sensitive are the results to different values of $\lambda$? The paper lacks discussion on this critical design choice.
2. What is the performance of the proposed method under low feedback rates (i.e., when semi-bandit feedback is rarely triggered)? The current experiments do not analyze or discuss scenarios with sparse feedback.
3. Could there exist a more direct threshold-updating mechanism (e.g., standard online quantile tracking or reinforcement learning methods) that also achieves adversarial miscoverage control? Why is the bandit framework preferred here, and what are its concrete advantages over such alternatives?
4. In the abstract (line 025), the word "micoverage" should be corrected to "miscoverage".
5. In line 127, the definition of “size” is unclear. Why is it claimed to be proportional to exp(-$\pi$)? Please provide a reasonable justification or reference.
6. In line 172 (Equation 3), the definition of $a_t(\pi_t)$ appears, but later discussions refer to $a_t(\pi)$. Are they the same quantity? A unified and explicit definition for $a_t(\pi)$ is missing.
7. In line 213 under EXP3.P-CP, the parameters $\delta$and $K$ are used without prior introduction or definition. Please clarify their meanings.
8. In line 225, what is the valid range of the probability distribution $p_t$? It would be helpful to explicitly state its domain or constraints.
9. In Figure 1, the blue curve (MVP) is barely visible or unidentifiable. Please consider adjusting the visualization for clarity.
10. In line 568, there is an unexplained symbol or placeholder "(?)", which appears to be a typo.
11. On the final page of the appendix (Page 18), there are punctuation issues in one of the equations. A thorough proofreading of all punctuation throughout the paper is recommended.

---

> ### Author Response · Authors · 2025-11-17
> **Question 1**
>
> ### **1. In both experiments, the choice of the hyperparameter is not specified. How is selected, and how sensitive are the results to different values of? The paper lacks discussion on this critical design choice.**
>
> ***In short, experimental results empirically show that our methods are robust under different values of $\lambda$.***
>
> Specifically, we have conducted an ablation study in both experiments by varying $\lambda$, while controlling the other hyperparameters to be fixed.
>
> [Ablation Study on Robustness of Our Method on Different Values of $\lambda$]
>
> [Experiment 1: ImageNet (i.i.d.)]
> - Hyperparameter set-up: $\alpha=0.1, K=200, T = 50, 000$
> - The following results are average results over 50 trials.
>
> [$\texttt{EXP3.P-CP}$]
> | $\lambda$ | 0.1 | 0.5 | 1.0 | 2.0 | 5.0 | 10.0 |
> |----------|----------|----------|----------|----------|----------|----------|
> | Coverage | 0.857 | 0.856 | 0.855 | 0.853 | 0.850 | 0.850 |
> | Width    | 41.75 | 38.95 | 39.21 | 37.64 | 36.71 | 35.61 |
>
> [$\texttt{EXP3.P-CP-UNLOCK}$]
> | $\lambda$ | 0.1 | 0.5 | 1.0 | 2.0 | 5.0 | 10.0 |
> |----------|----------|----------|----------|----------|----------|----------|
> | Coverage | 0.898 | 0.897 | 0.896 | 0.894 | 0.893 | 0.892 |
> | Width    | 63.61 | 61.77 | 60.51 | 59.21 | 57.08 | 56.12 |
>
> [Experiment 2: Airfoil (i.i.d.)]
> - Hyperparameter set-up: $\alpha=0.1, K=20, T = 1, 127$
> - The following results are average results over 100 trials.
>
> [$\texttt{EXP3.P-CP}$]
> | $\lambda$ | 0.1 | 0.5 | 1.0 | 2.0 | 5.0 | 10.0 |
> |----------|----------|----------|----------|----------|----------|----------|
> | Coverage | 0.897 | 0.896 | 0.895 | 0.894 | 0.893 | 0.892 |
> | Width    | 14.51 | 14.44 | 14.41 | 14.36 | 14.29 | 14.28 |
>
> [$\texttt{EXP3.P-CP-UNLOCK}$]
> | $\lambda$ | 0.1 | 0.5 | 1.0 | 2.0 | 5.0 | 10.0 |
> |----------|----------|----------|----------|----------|----------|----------|
> | Coverage | 0.908 | 0.907 | 0.906 | 0.905 | 0.903 | 0.903 |
> | Width    | 15.12 | 15.04 | 14.98 | 14.90 | 14.78 | 14.73 |
>
> [Experiment 3: Airfoil (Covariate Shift)]
> - Hyperparameter set-up: $\alpha=0.1, K=20, T = 1, 127$
> - The following results are average results over 100 trials.
>
> [$\texttt{EXP3.P-CP}$]
> | $\lambda$ | 0.1 | 0.5 | 1.0 | 2.0 | 5.0 | 10.0 |
> |----------|----------|----------|----------|----------|----------|----------|
> | Coverage | 0.841 | 0.840 | 0.839 | 0.838 | 0.836 | 0.836 |
> | Width    | 14.85 | 14.76 | 14.73 | 14.67 | 14.61 | 14.60 |
>
> [$\texttt{EXP3.P-CP-UNLOCK}$]
> | $\lambda$ | 0.1 | 0.5 | 1.0 | 2.0 | 5.0 | 10.0 |
> |----------|----------|----------|----------|----------|----------|----------|
> | Coverage | 0.848 | 0.847 | 0.846 | 0.844 | 0.843 | 0.842 |
> | Width    | 15.13 | 15.07 | 15.03 | 15.01 | 14.90 | 14.86 |
>
> As the above ablation study shows, our method shows robust performance irrespective of the choice of $\lambda$, which in return has led us to set $\lambda = 1$ uniformly across all experiment set-ups.
>
> ***Nevertheless, we totally agree with your ample feedback on the absence of the discussion on the design choice of $\lambda$.***
>
> ***Moreover, we have further improved our $\texttt{EXP3.P-CP-UNLOCK}$ algorithm that provides users with $\lambda$ in a closed form as a function of the user-specified parameters $T$ (terminal time horizon) and $\alpha$ (target miscoverage rate), while enjoying the same theoretical guarantee.***
>
> We will update this modification in the updated version of our draft.

---

> ### Author Response · Authors · 2025-11-17
> **Question 2**
>
> ### **2. What is the performance of the proposed method under low feedback rates (i.e., when semi-bandit feedback is rarely triggered)? The current experiments do not analyze or discuss scenarios with sparse feedback.**
>
> In short, ***while we cannot pre-specify the rate of semi-bandit feedback to be low***, we conducted ***experiments on the same dataset (ImageNet) under different data streaming set-ups by varying levels of difficulty.***
>
> [1. Why We Cannot Specify Low Semi-bandit Feedback Rate]
>
> Semi-bandit feedback is generated if the chosen conformal set ($C_t(x_t)$) contains the scoring function of the true label ($y_t$), i.e., $m_t(\pi_t) = 0$, where our algorithm is guaranteed to achieve the target coverage rate ($1 - \alpha$) under this set-up.
>
> Roughly speaking, among $T$ iterations, we will receive around $(1 - \alpha) \times T$ times of semi-bandit feedback.
>
> ***In other words, since the target miscoverage rate ($\alpha$) determines the rate of semi-bandit feedback, we could not keep semi-bandit feedback rates to be low unless we set $\alpha$ to be high (which is not of our research interest).***
>
> [2. Ablation Study by Varying Levels of Difficulty]
>
> While we cannot specify the rate of semi-bandit feedback to be low, we have considered different data steaming set-ups on ImageNet by varying the levels of difficulty as the following:
>
> (a) Stochastic set-up (i.i.d.)
>
> (b) Non-stochastic set-up: We have modified the functional form of the scoring function every 10,000 time steps.
>
> The following are experiment results for each data-streaming set-up.
>
> [Ablation Study on ImageNet by Varying Levels of Difficulty]
>
> [Experiment 1: ImageNet (i.i.d.)]
> - Hyperparameter set-up: $\alpha=0.15, K=1, 000, T = 50, 000$
> - The following results are average results over 50 trials.
>
> | Method | $\texttt{MVP}$ | $\texttt{SCP}$ | $\texttt{EXP3.P-CP}$ | $\texttt{EXP3.P-CP-UNLOCK}$ |
> |----------|----------|----------|----------|----------|
> | Coverage | 0.836 | 0.850 | 0.809 | 0.844 |
> | Width    | 2.016 | 2.162 | 20.90 | 32.68
>
> [Experiment 2: ImageNet (Multiple Shifts)]
> - Hyperparameter set-up: $\alpha=0.15, K=1, 000, T = 50, 000$
> - The following results are average results over 50 trials.
> - We have scaled the original scoring function differently for every $10,000$ iterations to the power of $1/6 ~ (t= \sim 10\text{k}), 1/2 ~ (t=10\text{k} \sim 20\text{k}), 1/4 ~ (t=20\text{k} \sim 30\text{k}), 1/2.5 ~ (t=30\text{k} \sim 40\text{k}), ~ \text{and} ~ 1/1.2 ~ (t=40\text{k} \sim 50\text{k})$, respectively. Roughly speaking, it becomes more challenging as the scale is close to 0, since all the scoring functions are concentrated toward 1, making a true scoring function hard to distinguish from others.
>
> | Method | $\texttt{MVP}$ | $\texttt{SCP}$ | $\texttt{EXP3.P-CP}$ | $\texttt{EXP3.P-CP-UNLOCK}$ |
> |----------|----------|----------|----------|----------|
> | Coverage | 0.821 | 0.727 | 0.795 | 0.826 |
> | Width    | 1.975 | 1.238 | 34.28 | 40.26
>
> ***The more the data-streaming set-up deviates from the i.i.d. scenario, the larger the average size of the conformal set becomes, indicating that the learner is having difficulty in terms of constructing a compact conformal set containing the true scoring function.***
>
> [3. Final Remark on the Low Semi-bandit Feedback Rate Scenario]
>
> We want to make a final remark that $\texttt{EXP3.P-CP}$, an algorithm that directly applies $\texttt{EXP3.P}$ algorithm to online conformal prediction problems with semi-bandit feedback, can be understood as a proxy to the scenario under the low semi-bandit feedback rate scenario.
>
> Specifically, since $\texttt{EXP3.P-CP}$ learner makes a decision solely based on the chosen conformal set at each time step without any exploitation of unlocking mechanisms, it can be understood as an approximation to low semi-bandit feedback rate scenario.
>
> ***In conclusion, requiring additional time steps in terms of approaching the target coverage compared to the learner exploiting semi-bandit feedback ($\texttt{EXP3.P-CP-UNLOCK}$) implies the importance of incorporating unlocking mechanisms into the algorithm (coverage (stochastic): 0.809 vs. 0.844; coverage (non-stochastic): 0.804 vs. 0.838).***

---

> ### Author Response · Authors · 2025-11-17
> **Question 3 - Question 7**
>
> ### **3. Could there exist a more direct threshold-updating mechanism (e.g., standard online quantile tracking or reinforcement learning methods) that also achieves adversarial miscoverage control? Why is the bandit framework preferred here, and what are its concrete advantages over such alternatives?**
>
> While there are existing direct threshold-updating mechanisms that tackle online conformal prediction problems under arbitrary distribution shifts [1, 2, 3], the main difference is in the specific problem set-up.
>
> Specifically, while existing methods typically assume ***full feedback*** set-up where evaluation over all possible conformal sets is possible for every time step, our work considers ***partial feedback (semi-bandit feedback)*** set-up [4], a more challenging set-up where evaluation over all possible conformal sets is feasible only when the chosen conformal set contains the true label.
>
> Recalling that [4] assumes online conformal prediction problems under the i.i.d. data generating process, we want to highlight that ***our work is the first time to tackle online conformal prediction problems under both (1) arbitrary distribution shifts and (2) partial feedback set-up.***
>
> As a side note, as illustrated in **Line 113** - **Line 115**, we would like to note that the data stream we consider is “almost” arbitrary, in the sense that we consider an adaptive adversary.
>
> [1] [Gibbs and Candes., NeurIPS, 2021] Adaptive Conformal Inference Under Distribution Shift
>
> [2] [Gibbs and Candes, JMLR, 2024] Conformal Inference for Online Prediction with a Arbitrary Distribution Shifts
>
> [3] [Bastani et al., NeurIPS, 2022] Practical Adversarial Multivalid Conformal Prediction
>
> [4] [Ge et al., ICML, 2025] Stochastic Online Conformal Prediction with Semi-bandit Feedback
>
> ### **4. In the abstract (Line 025), the word "micoverage" should be corrected to "miscoverage".**
>
> Thank you for catching the typo.
>
> We have corrected from “micoverage” to “miscoverage” in the abstract, and carefully checked for similar typos elsewhere.
>
> ### **5. In Line 127, the definition of “size” is unclear. Why is it claimed to be proportional to $\text{exp}(-\pi)$? Please provide a reasonable justification or reference.**
>
> We agree that our original explanation was unclear.
>
> ***Formally speaking, we define ``size’’ of a given conformal set as its cardinality.***
>
> We have used the expression that ``size is proportional to $\text{exp}(-\pi)$’’ in the sense that $\text{exp}(-\pi)$ is a decreasing function in $\pi$.
>
> More specifically, since the cardinality of a conformal set decreases as its threshold parameter ($\pi$) increases, ***$\text{exp}(-\pi)$ can be understood as a smooth surrogate of ``size’’ that encourages the learner to construct more compact conformal sets.***
>
> Indeed, the surrogate of ``size’’ can be of any form as long as (1) it is a decreasing function in $\pi$ and (2) bounded by [0, 1] ($\because ~ \text{Lemma 1}$).
>
> We once again appreciate your valuable feedback, and will update **Line 127** accordingly.
>
> ### **6. In Line 172 (Equation 3), the definition of $a_t(\pi_t)$ appears, but later discussions refer to $a_t(\pi)$. Are they the same quantity? A unified and explicit definition for $a_t(\pi)$ is missing.**
>
> We apologize for the confusion due to the inconsistent notation usage.
>
> In the revised version, we will provide a unified and explicit definition for $a_t(\pi)$ as the following.
>
> Specifically, we define $a_t (\pi)$ as the loss term that encourages small ``size’’ (**Question 5**) at time step $t$, which is evaluated on a conformal set parameterized by $\pi$.
>
> Accordingly, $a_t (\pi_t)$ will be used as the loss term related to ``size’’ evaluated on the specific chosen conformal set by our learner at time step $t$.
>
> We appreciate once again for your ample feedback on improving the readability of our paper.
>
> ### **7. In Line 213 under $\texttt{EXP3.P-CP}$, the parameters $\delta$ and $K$ are used without prior introduction or definition. Please clarify their meanings.**
>
> In the revised version, we will first define the terms $\delta$ and $K$ prior to using them in our algorithm description in **Line 213**.
>
> 1. $\delta$: The confidence level that appears in the high-probability miscoverage guarantee. Built upon the $\texttt{EXP3.P}$ algorithm, we want to highlight that our algorithm holds for any confidence level $\delta$.
>
> 2. $K$: The number of candidate thresholds parametrizing each conformal set, which comprise our hypothesis space. Each threshold corresponds to the arm in the bandit learning set-up.

---

> ### Author Response · Authors · 2025-11-17
> **Question 8 - Question 11**
>
> ### **8. In Line 225, what is the valid range of the probability distribution $p_t$? It would be helpful to explicitly state its domain or constraints.**
>
> We will explicitly state the domain of $p_t$ in our revised version.
>
> $p_t \in \mathbb{R}^K$ is defined as a probability vector defined on our hypothesis space of $K$ different threshold parameters, from which $\pi_t$ is sampled at time step $t$.
>
> Specifically, $p_t$ belongs to the $K$-dimensional probability simplex, where (1) each component of $p_t$ is non-negative and (2) the sum of all components of $p_t$ equals 1.
>
> ### **9. In Figure 1, the blue curve (MVP) is barely visible or unidentifiable. Please consider adjusting the visualization for clarity.**
>
> Thank you for your invaluable feedback.
>
> In the revised version, we will adjust the visualization for clarity.
>
> Specifically, our coloring scheme and line styles will be updated to make the blue curve identifiable.
>
> ### **10. In Line 568, there is an unexplained symbol or placeholder "(?)", which appears to be a typo.**
>
> We thank you once again for your feedback.
>
> This symbol occurred due to the missing reference.
>
> We will make sure to update it in our revised version.
>
> ### **11. On the final page of the appendix (Page 18), there are punctuation issues in one of the equations. A thorough proofreading of all punctuation throughout the paper is recommended.**
>
> We appreciate your ample feedback, and will thoroughly proofread all punctuations throughout our paper, and revise it accordingly.
>
> We will make sure that the updated version addresses all of these errors for clarity and completeness.

---

### Official Review · Reviewer_d5vm · 2025-10-30

**Soundness:** 3
**Presentation:** 2
**Contribution:** 3
**Rating:** 6
**Confidence:** 3

**Summary:**

The paper explores how regret minimization techniques from bandit and sequential prediction literature can be adapted to online conformal prediction under partial feedback. It establishes a theoretical link by deriving an upper bound on the miscoverage rate in terms of the learner’s regret, enabling the use of these methods in this setting. Building on this, the paper demonstrates the application of the EXP3.P algorithm and introduces a novel variant, EXP3.P-CP with Unlocking, which exploits additional partial feedback and the monotonicity of prediction set construction. For both methods, the paper provides miscoverage guarantees and evaluates their empirical performance on regression and classification tasks.

**Strengths:**

- The partial feedback setting in online conformal prediction is highly relevant, offering practical value to both the conformal prediction research community and real-world practitioners.
- The paper’s connection to regret minimization methods is particularly compelling, as it unlocks a rich toolbox of established techniques that can be effectively applied to conformal prediction. I am not an expert in this specific area of conformal prediction, and I am surprised this connection has not appeared in the literature before, but to the best of my knowledge, it is novel.
- The paper is generally clear and well-structured, making it easy to follow. The proofs are also well detailed and, I believe, sound.

**Weaknesses:**

- The experiments are somewhat limited in the sense that the only baselines considered operate under a different setting. This makes it hard to evaluate the effectiveness of the proposed method. Are there no other conformal prediction methods designed for partial feedback, or at least methods that could be adapted to this setting?
- The presentation could be improved as there are a few minor typographic errors and occasional non-idiomatic phrasing (see minor issues below).

### Minor issues

- Line 16: “with fewer data generation assumptions” instead of “with less data generation assumption”
- Line 18: "mature literature" or "established literature" instead of “matured literatures”
- Line 41:  I am not sure “enjoyable” is a good word choice in “enjoyable guarantee”. Maybe just “guarantee” would read better.
- Line 126: “minimizes” should be “minimize”.
- Line 133: The sentence “due to exploiting the achievement of adversarial bandits under an adaptive adversary” is not clear. What does the “achievement of adversarial bandits” mean?
- Line 159: “which we leverage this algorithm as our baseline” does not read well. Probably better to add a full stop and then “We leverage this algorithm as our baseline”.
- Line 208: “method” instead of “methods”.
- Line 209: “to use” instead of “to using”.
- Line 568: Missing reference.

**Questions:**

1. After Lemma 1, it says that “the corresponding miscoverage rate converges to a desired level $\alpha$”. However, we do not really see that in the results of Figure 1. Why is that?
2. How are the set of candidate thresholds chosen? How does that affect the results in both theoretical and practical terms? I imagine that the resolution of the threshold might affect the convergence of the regret, for example.
3. The results focus on the absolute coverage gap, treating undercoverage and overcoverage equally. Would it be possible to extend the analysis to penalize undercoverage more heavily than overcoverage? Relatedly, EXP3.P-CP with Unlocking appears more conservative than EXP3.P-CP, leading to overcoverage in Figure 1. Is this behavior expected?

---

> ### Author Response · Authors · 2025-11-18
> **Weakness 1 - Weakness 2**
>
> ### **Weakness 1. The experiments are somewhat limited in the sense that the only baselines considered operate under a different setting. This makes it hard to evaluate the effectiveness of the proposed method. Are there no other conformal prediction methods designed for partial feedback, or at least methods that could be adapted to this setting?**
> We appreciate your invaluable feedback on additional experiments on baselines that can be adapted to partial feedback set-up.
>
> To the best of our knowledge, $\texttt{SPS}$ algorithm proposed by [1] is the only method that tackles online conformal prediction problems under partial feedback set-up.
>
> While we couldn’t attach experiment results on $\texttt{SPS}$ in our initial draft due to its lack of public code repository, we have inquired about access to implementation code from original authors and attached corresponding results as below.
>
> To compare $\texttt{SPS}$ with our method, we conducted experiments on the same dataset (ImageNet) under two different data streaming set-ups (i.i.d., multiple shifts), where the theoretical guarantee of $\texttt{SPS}$ is valid only under the i.i.d. set-up.
>
> [ImageNet (i.i.d.)]
>
> - Hyperparameter set-up: $\alpha=0.15, K=1, 000, T = 50, 000$
>
> - The following results are average results over 50 trials.
>
> | Method | $\texttt{SPS}$ | $\texttt{EXP3.P-CP}$ | $\texttt{EXP3.P-CP-UNLOCK}$ |
> |----------|----------|----------|----------|
> | Coverage | 0.878 | 0.809 | 0.844 |
> | Width    | 13.00 | 20.90 | 32.68 |
>
> [ImageNet (multiple shifts)]
>
> - Hyperparameter set-up: $\alpha=0.15, K=1, 000, T = 50, 000$
>
> - The following results are average results over 50 trials.
>
> - We have scaled the original scoring function differently for every $10,000$ iterations to the power of $1/6 ~ (t= \sim 10\text{k}), 1/2 ~ (t=10\text{k} \sim 20\text{k}), 1/4 ~ (t=20\text{k} \sim 30\text{k}), 1/2.5 ~ (t=30\text{k} \sim 40\text{k}), ~ \text{and} ~ 1/1.2 ~ (t=40\text{k} \sim 50\text{k})$, respectively. Roughly speaking, it becomes more challenging as the scale is close to 0, since all the scoring functions are concentrated toward 1, making a true scoring function hard to distinguish from others.
>
> | Method | $\texttt{SPS}$ | $\texttt{EXP3.P-CP}$ | $\texttt{EXP3.P-CP-UNLOCK}$ |
> |----------|----------|----------|----------|
> | Coverage | 0.761 | 0.795 | 0.826 |
> | Width    | 11.84 | 34.28 | 40.47 |
>
> While $\texttt{SPS}$ achieves target coverage with compact conformal set size in the i.i.d. set-up, $\texttt{SPS}$ has difficulty in reaching the target coverage as expected (0.761), while our method ($\texttt{EXP3.P-CP-UNLOCK}$) achieves quite reasonable empirical coverage (0.826).
>
> [1] [Ge et al., ICML, 2025] Stochastic Online Conformal Prediction with Semi-bandit Feedback
>
>
> ### **Weakness 2. The presentation could be improved as there are a few minor typographic errors and occasional non-idiomatic phrasing (see minor issues below).**
>
> We appreciate your feedback on revising typographic errors and non-idiomatic phrasing.
>
> We will make sure they are all  addressed in the revised version to improve the readability and clarity of our paper.

---

> ### Author Response · Authors · 2025-11-18
> **Question 1 - Question 2**
>
> ### **Question 1. After Lemma 1, it says that “the corresponding miscoverage rate converges to a desired level $\alpha$”. However, we do not really see that in the results of Figure 1. Why is that?**
>
> Before answering **Question 1**, we will adopt the expressions "absolute coverage gap", "undercoverage", and "overcoverage" in **Question 3** for consistency and clarity.
>
> We first apologize for our imprecise expression "the corresponding miscoverage rate converges to a desired level $\alpha$", causing a confusion that **Lemma 1** upper bounds "absolute coverage gap".
>
> Indeed, **Lemma 1** states that the difference between the miscoverage ($\text{\textbf{MC}}_T$) and the target miscoverage ($\alpha$) is upper bounded by $\text{Reg}(T)/T$, where $\text{Reg}(T)$ of our algorithm is of order $\mathcal{O}(\sqrt{K \text{ln}K T})$ (**Theorem 1**), implying that $\text{\textbf{MC}}_T - \alpha$ is roughly less than or equal to 0 for long time horizon ($T$).
>
> In other words, combined with **Theorem 1**, **Lemma 1** provides a guarantee of our algorithm on the "overcoverage" (not on the "absolute coverage gap"), which is aligned with results of **Figure 1**.
>
> We thank you once again for ample feedback on our imprecise use of the expression, and will make sure to update it in our revised version.
>
>
> ### **Question 2. How are the set of candidate thresholds chosen? How does that affect the results in both theoretical and practical terms? I imagine that the resolution of the threshold might affect the convergence of the regret, for example.**
>
> We appreciate your feedback on the effect of the choice of $K$ on the results both in terms of the theoretical and empirical manner.
>
> In terms of the ***theoretical guarantee***, our regret bound is of rate $\mathcal{O} ( \sqrt{K \text{ln}K T} )$, so increasing $K$ makes the threshold grid finer at the cost of increasing the regret bound.
>
> Meanwhile, in terms of the ***empirical results***, experimental results empirically show that finer discretization of the hypothesis space ($\uparrow K$) is required for keeping size of the chosen conformal set to be compact.
>
> Specifically, we have conducted an ablation study in both experiments by varying $K$, while controlling the other hyperparameters to be fixed.
>
> [Experiment 1: ImageNet (i.i.d.)]
>
> - Hyperparameter set-up: $\alpha=0.1, \lambda=1, T = 50, 000$
>
> - The following results are average results over 50 trials.
>
> [$\texttt{EXP3.P-CP}$]
> | $K$ | 200 | 500 | 1000 |
> |----------|----------|----------|----------|
> | Coverage | 0.854 | 0.828 | 0.811 |
> | Width    | 36.84 | 26.01 | 21.58 |
>
> [$\texttt{EXP3.P-CP-UNLOCK}$]
> | $K$ | 200 | 500 | 1000 |
> |----------|----------|----------|----------|
> | Coverage | 0.896 | 0.867 | 0.842 |
> | Width    | 60.60 | 40.15 | 30.54 |
>
> [Experiment 2: Airfoil (i.i.d.)]
>
> - Hyperparameter set-up: $\alpha=0.1, \lambda=1, T = 1, 127$
>
> - The following results are average results over 100 trials.
>
> [$\texttt{EXP3.P-CP}$]
> | $K$ | 20 | 40 | 60 |
> |----------|----------|----------|----------|
> | Coverage | 0.894 | 0.869 | 0.852 |
> | Width    | 14.42 | 13.61 | 13.21 |
>
> [$\texttt{EXP3.P-CP-UNLOCK}$]
> | $K$ | 20 | 40 | 60 |
> |----------|----------|----------|----------|
> | Coverage | 0.904 | 0.880 | 0.865 |
> | Width    | 14.88 | 14.11 | 13.58 |
>
> [Experiment 3: Airfoil (Covariate Shift)]
>
> - Hyperparameter set-up: $\alpha=0.1, \lambda=1, T = 1, 127$
>
> - The following results are average results over 100 trials.
>
> [$\texttt{EXP3.P-CP}$]
> | $K$ | 20 | 40 | 60 |
> |----------|----------|----------|----------|
> | Coverage | 0.834 | 0.807 | 0.789 |
> | Width    | 14.62 | 13.75 | 13.21 |
>
> [$\texttt{EXP3.P-CP-UNLOCK}$]
> | $K$ | 20 | 40 | 60 |
> |----------|----------|----------|----------|
> | Coverage | 0.842 | 0.816 | 0.800 |
> | Width    | 14.91 | 14.05 | 13.55 |

---

> ### Author Response · Authors · 2025-11-18
> **Question 3**
>
> ### **Question 3. The results focus on the absolute coverage gap, treating undercoverage and overcoverage equally. Would it be possible to extend the analysis to penalize undercoverage more heavily than overcoverage? Relatedly, $\texttt{EXP3.P-CP}$ with Unlocking appears more conservative than $\texttt{EXP3.P-CP}$, leading to overcoverage in Figure 1. Is this behavior expected?**
>
> [First Part of **Question 3**]
>
> We believe that an answer to **Question 1** will also be an answer to this part.
>
> [Second Part of **Question 3**]
>
> In addition, we believe that the following remark in addition to the answer to the first part of **Question 3** will answer the second part of **Question 3**.
>
> Specifically, $\texttt{EXP3.P-CP}$, an algorithm that directly applies $\texttt{EXP3.P}$ algorithm to online conformal prediction problems with semi-bandit feedback, makes ***a decision solely based on the chosen conformal set*** at each time step without any exploitation of unlocking mechanisms as $\texttt{EXP3.P-CP-UNLOCK}$ does.
>
> This requires for $\texttt{EXP3.P-CP}$ to have additional time steps in terms of approaching the target coverage compared to the learner exploiting semi-bandit feedback ($\texttt{EXP3.P-CP-UNLOCK}$), which made the behavior in **Figure 1-(a)**, **Figure 1-(c)** expected to occur before running an experiment.

---

> > ### Comment · Reviewer_d5vm · 2025-11-24
> >
> > Thank you for addressing my concerns; they were all solved satisfactorily. In particular, the new experimental results and the discussion around the trade-offs involved in setting $K$ are valuable additions to the paper that should be included in the final version. I will update my score accordingly.

---

> > > ### Author Response · Authors · 2025-11-25
> > >
> > > We sincerely appreciate once again your valuable feedback on our paper and the update of the score. We will make sure to include the additional experimental results, as well as all the other feedback you provided, in the final version of our draft.

---

### Official Review · Reviewer_KKfP · 2025-10-31

**Soundness:** 3
**Presentation:** 2
**Contribution:** 3
**Rating:** 6
**Confidence:** 3

**Summary:**

The paper proposes a new framework for online conformal prediction under partial and adversarial feedback, where the learner only observes limited information about prediction correctness.  Conformal prediction has been connected to adversarial bandit learning by framing conformal set selection as an arm-selection problem. They develop a conversion lemma linking regret bounds to miscoverage guarantees, allowing the use of existing bandit algorithms for online conformal prediction.

**Strengths:**

- Establishes a clear connection between regret minimization and coverage guarantees in conformal prediction.

- The introduction of the EXP3.P-CP-UNLOCK algorithm effectively adapts the classic EXP3.P framework to the unique structure of conformal prediction feedback.

- Provides regret and miscoverage bounds with detailed proofs.

**Weaknesses:**

- The review of online conformal prediction is insufficient. Important recent works are missing, including:
   - Angelopoulos et al. (2023), Conformal PID Control for Time Series Prediction;
   - Angelopoulos et al. (2024), Online Conformal Prediction with Decaying Step Sizes;
   - Gibbs & Candès (2024), Conformal Inference for Online Prediction with Arbitrary Distribution Shifts;
   - Zhang et al. (2025), Online Differentially Private Conformal Prediction for Uncertainty Quantification.


- As $m_t$ is an indicator function, the regret formulation may not distinguish between observing or not observing $y_t$, potentially causing information loss. Clarification on how the true label feedback is used is needed.

- A formal theorem explicitly stating the miscoverage rate bound would make the theoretical results clearer.

- Please clarify whether specific techniques are used to address non-i.i.d. or non-exchangeable data.

- The differences from Ge et al. (2025) should be highlighted more clearly in terms of assumptions,method, and theoretical guarantees.

**Questions:**

- Should it be “minimize the miscoverage rate” in line 128 on page 3?

- There are no comparisons with other recent partial-feedback uncertainty quantification frameworks.

- Please discuss how sensitive the results are to the choice of $λ$ and the discretization level $K$.

---

> ### Author Response · Authors · 2025-11-18
> **Weakness 1 - Weakness 3**
>
> ### **Weakness 1. The review of online conformal prediction is insufficient. Important recent works are missing.**
>
> In the revised version, we will update the **Related Work** section to ***(1) introduce and reference all of your suggested papers on online conformal prediction*** and ***(2) compare them to our paper.***
>
> While both existing methods and our work are tackling online conformal prediction problems under arbitrary distribution shifts, the main difference is in the specific problem set-up.
>
> Specifically, while existing methods typically assume ***full feedback*** set-up where evaluation over all possible conformal sets are possible for every time step, our work considers ***partial feedback (semi-bandit feedback)*** set-up [1], a more challenging set-up where evaluation over all possible conformal sets are feasible only when the chosen conformal set contains the true label.
>
> Recalling that the work from [1] assumes online conformal prediction problems under the i.i.d. data generating process, we want to highlight that ***our work is the first time to tackle online conformal prediction problems under both (1) arbitrary distribution shifts and (2) partial feedback set-up.***
>
> [1] [Ge et al., ICML, 2025] Stochastic Online Conformal Prediction with Semi-bandit Feedback
>
> ### **Weakness 2. As $m_t$ is an indicator function, the regret formulation may not distinguish between observing or not observing $y_t$, potentially causing information loss. Clarification on how the true label feedback is used is needed.**
>
> We first appreciate your ample feedback on the lack of clarity in the description of our method.
>
> In short, ***the observability of the true label ($y_t$), i.e., whether $m_t(\pi_t) = 0$ or $m_t(\pi_t) = 1$, determines the degree of the feedback unlocking.***
>
> Here, the degree of the feedback unlocking is defined as the accessibility to gains on other candidate conformal sets except for the chosen one ($C_{\pi_t}(x_t)$).
>
> Specifically, under the partial feedback scenario we consider, $y_t$ is observed only when $C_{\pi_t}(x_t)$ contains $y_t$, i.e., $m_t(\pi_t)=0$. Since we can evaluate the true score function ($f_t(x_t, y_t)$) after observing $y_t$, we can evaluate gains for all possible conformal sets on whether each conformal set contains $y_t$.
>
> In other words, the partial feedback scenario is equivalent to the full feedback scenario when $m_t(\pi_t)=0$. Please refer to **Line 9** - **Line 10** of **Algorithm 1** ($\texttt{EXP3.P-CP-UNLOCK}$).
>
> Indeed, additional information on gains of the other conformal sets except for $C_{\pi_t}(x_t)$ is the reason why $\texttt{EXP3.P-CP-UNLOCK}$ achieves the target coverage faster than $\texttt{EXP3.P-CP}$ (**Figure 1-(a)**, **Figure 1-(c)**), where $\texttt{EXP3.P-CP}$ updates the distribution over conformal sets only using the gain of $C_{\pi_t}(x_t)$ at each time step.
>
> In conclusion, we will revise the initial draft to clarify ***how the observability of the true label ($y_t$) makes achieving the target coverage faster through the feedback unlocking***, which could not have been achieved by directly applying the $\texttt{EXP3.P}$ algorithm ($\texttt{EXP3.P-CP}$).
>
> ### **Weakness 3. A formal theorem explicitly stating the miscoverage rate bound would make the theoretical results clearer.**
>
> We totally agree that a modification of **Theorem 1** to a formal theorem on the upper bound of the miscoverage rate will improve the readability.
>
> Specifically, combined with **Lemma 1** which connects the miscoverage rate and the regret, ***we will either (1) update Theorem 1 or (2) introduce an additional corollary to make the statement be explicitly stated as the high probability upper bound of the miscoverage rate, not of the regret as in the original draft***.

---

> ### Author Response · Authors · 2025-11-18
> **Weakness 4 - Weakness 5**
>
> ### **Weakness 4. Please clarify whether specific techniques are used to address non-i.i.d. or non-exchangeable data.**
>
> As formalized in [1], we consider the online conformal prediction problem under the adversarial setting, where (1) $(X_1, Y_1), (X_2, Y_2), \dots$ to be an arbitrary sequence of (input, answer) pairs in $\mathcal{X} \times \mathcal{Y}$, and (2) $f_1(\cdot, \cdot), f_2 (\cdot, \cdot), \dots$ to be an arbitrary sequence of scoring functions which map from $\mathcal{X} \times \mathcal{Y}$ to $[0, 1]$.
>
> More precisely, the data stream we consider is “almost” arbitrary, in the sense that we consider an adaptive adversary (**Line 113** - **Line 115**).
>
> Built upon the $\texttt{EXP3.P}$ algorithm, both of our algorithms $\texttt{EXP3.P-CP}$ and $\texttt{EXP3.P-CP-UNLOCK}$ are tailored to the online conformal prediction problem under the adversarial setting, which provides the guarantee on the sub-linear regret with respect to the time horizon ($T$).
>
> The only condition for the theoretical guarantee is on ***the boundedness of loss function for every time step***, as the original $\texttt{EXP.3.P}$ algorithm is (**Line 249** in **Theorem 1**).
>
> We will clarify in the revised version that ***our guarantee holds under the adversarial setting***, without any assumptions on both the data generating process and the functional form of the scoring function.
>
> [1] [Angelopoulos et al., ICML, 2024] Online Conformal Prediction with Decaying Step Sizes
>
> ### **Weakness 5. The differences from Ge et al. (2025) should be highlighted more clearly in terms of assumptions,method, and theoretical guarantees.**
>
> We fully appreciate that elaborating the differences between our method and Ge et al. (2025) in terms of (1) assumptions, (2) methodology, and (3) guarantees will clarify our method and its contribution.
>
> [1. Assumption]
>
> First, motivated by Ge et al. (2025), our method assumes partial (semi-bandit) feedback set-up where evaluation over all candidate conformal sets are possible only when the chosen conformal set ($C_{\pi_t}(x_t))$) contains the true label ($y_t$), i.e., $m_t(\pi_t)=0$.
>
> The major difference lies in ***the assumption on the data streams***. While Ge et al. (2025) assumes the stream of i.i.d. data sequences, we consider arbitrary data streams without any assumptions on both the data generating process and the functional form of the scoring function. Specifically, as addressed in **Question 4**, the data stream we consider is “almost” arbitrary, in the sense that we consider an adaptive adversary (**Line 113** - **Line 115**).
>
> [2. Methodology]
>
> While Ge et al. (2025) ***directly updates the conformal set*** at each time step in an online manner, our method ***updates the probability mass function*** ($p_t$) defined on the hypothesis space of size $K$, from which the conformal set ($C_{\pi_t}(x_t)$) is chosen at time step $t$.
>
> [3. Theoretical Guarantee]
>
> As also illustrated in **[1. Assumption]**, the major difference in terms of theoretical guarantee is ***under which data streams the guarantee is valid***.
>
> While the theoretical guarantee from Ge et al. (2025) is valid under the i.i.d. data streams, our guarantee is valid under arbitrary data streams by an adaptive adversary.
>
> Meanwhile, in terms of ***the actual content of the theoretical guarantee***, Ge et al. (2025) provides stronger theoretical guarantee in terms of two aspects.
>
> First of all, their coverage guarantee is in ***expectation-sense***. While our guarantee controls the miscoverage in terms of empirical average, that of Ge et al. (2025) is provided in terms of expectation.
>
> However, we would like to mention that theoretical guarantees from existing works on online conformal prediction under arbitrary data streams are also stated in terms of the empirical miscoverage (e.g., [1]).
>
> Second, the theoretical guarantee from Ge et al. (2025) is ***any-time guarantee***, in the sense that the constructed conformal set ($C_{\pi_t}(x_t)$) contains the true label with probability at least $1-\texttt{miscoverage}$ for all time-step $t \in [T]$ with high probability ($1 - 2/T$).
>
> We thank you once again for your ample feedback, and make sure that additional differences (e.g., technical assumptions) will be updated in the revised version.
>
> [1] [Gibbs and Candes., NeurIPS, 2021] Adaptive Conformal Inference Under Distribution Shift

---

> ### Author Response · Authors · 2025-11-18
> **Question 1 - Question 2**
>
> ### **Question 1. Should it be “minimize the miscoverage rate” in Line 128 on Page 3?**
> We appreciate catching the typo of our paper which has caused confusion.
>
> We will correct the phrase in **Line 128** on **Page 3** to ``minimize the miscoverage rate’’ in the updated version.
>
> ### **Question 2. There are no comparisons with other recent partial feedback uncertainty quantification frameworks.**
>
> To the best of our knowledge, $\texttt{SPS}$ algorithm proposed by Ge et al. (2025) is the only method that tackles online conformal prediction problems under partial feedback set-up.
>
> While we couldn’t attach experiment results on $\texttt{SPS}$ in our initial draft due to its lack of public code repository, we have inquired about access to implementation code from original authors and attached corresponding results as below.
>
> To compare $\texttt{SPS}$ with our method, we conducted experiments on the same dataset (ImageNet) under two different data streaming set-ups (i.i.d., multiple shifts), where the theoretical guarantee of $\texttt{SPS}$ is valid only under the i.i.d. set-up.
>
> [ImageNet (i.i.d.)]
> - Hyperparameter set-up: $\alpha=0.15, K=1, 000, T = 50, 000$
>
> - The following results are average results over 50 trials.
>
> | Method | $\texttt{SPS}$ (Ge et al., 2025) | $\texttt{EXP3.P-CP}$ | $\texttt{EXP3.P-CP-UNLOCK}$ |
> |----------|----------|----------|----------|
> | Coverage | 0.878 | 0.809 | 0.844 |
> | Width    | 13.00 | 20.90 | 32.68 |
>
> [ImageNet (multiple shifts)]
> - Hyperparameter set-up: $\alpha=0.15, K=1, 000, T = 50, 000$
>
> - The following results are average results over 50 trials.
>
> - We have scaled the original scoring function differently for every $10,000$ iterations to the power of $1/6 ~ (t= \sim 10\text{k}), 1/2 ~ (t=10\text{k} \sim 20\text{k}), 1/4 ~ (t=20\text{k} \sim 30\text{k}), 1/2.5 ~ (t=30\text{k} \sim 40\text{k}), ~ \text{and} ~ 1/1.2 ~ (t=40\text{k} \sim 50\text{k})$, respectively. Roughly speaking, it becomes more challenging as the scale is close to 0, since all the scoring functions are concentrated toward 1, making a true scoring function hard to distinguish from others.
>
> | Method | $\texttt{SPS}$ (Ge et al., 2025) | $\texttt{EXP3.P-CP}$ | $\texttt{EXP3.P-CP-UNLOCK}$ |
> |----------|----------|----------|----------|
> | Coverage | 0.761 | 0.795 | 0.826 |
> | Width    | 11.84 | 34.28 | 40.47 |
>
> While $\texttt{SPS}$ achieves target coverage with compact conformal set size in the i.i.d. set-up, $\texttt{SPS}$ has difficulty in reaching the target coverage as expected (0.761), while our method ($\texttt{EXP3.P-CP-UNLOCK}$) achieves quite reasonable empirical coverage (0.826).

---

> ### Author Response · Authors · 2025-11-18
> **Question 3 (Ablation Study on $K$)**
>
> ### **Question 3. Please discuss how sensitive the results are to the choice of $\lambda$ and the discretization level $K$.**
>
> We first thank you for your invaluable feedback on the lack of ablation study on the sensitivity of hyperparameters.
>
> We elaborate ***the ablation study for each hyperparameter*** as the following.
>
> [Ablation Study on Robustness of Our Method on Different Values of $K$]
>
> In terms of the ***theoretical guarantee***, our regret bound is of rate $\mathcal{O} ( \sqrt{K \text{ln}K T} )$, so increasing $K$ makes the threshold grid finer at the cost of increasing the regret bound.
>
> Meanwhile, in terms of the ***empirical results***, experimental results empirically show that finer discretization of the hypothesis space ($\uparrow K$) is required for keeping the size of the chosen conformal set compact.
>
> Specifically, we have conducted an ablation study in both experiments by varying $K$, while controlling the other hyperparameters to be fixed.
>
> [Experiment 1: ImageNet (i.i.d.)]
>
> - Hyperparameter set-up: $\alpha=0.1, \lambda=1, T = 50, 000$
>
> - The following results are average results over 50 trials.
>
> [$\texttt{EXP3.P-CP}$]
> | $K$ | 200 | 500 | 1000 |
> |----------|----------|----------|----------|
> | Coverage | 0.854 | 0.828 | 0.811 |
> | Width    | 36.84 | 26.01 | 21.58 |
>
> [$\texttt{EXP3.P-CP-UNLOCK}$]
> | $K$ | 200 | 500 | 1000 |
> |----------|----------|----------|----------|
> | Coverage | 0.896 | 0.867 | 0.842 |
> | Width    | 60.60 | 40.15 | 30.54 |
>
> [Experiment 2: Airfoil (i.i.d.)]
> - Hyperparameter set-up: $\alpha=0.1, \lambda=1, T = 1, 127$
>
> - The following results are average results over 100 trials.
>
> [$\texttt{EXP3.P-CP}$]
> | $K$ | 20 | 40 | 60 |
> |----------|----------|----------|----------|
> | Coverage | 0.894 | 0.869 | 0.852 |
> | Width    | 14.42 | 13.61 | 13.21 |
>
> [$\texttt{EXP3.P-CP-UNLOCK}$]
> | $K$ | 20 | 40 | 60 |
> |----------|----------|----------|----------|
> | Coverage | 0.904 | 0.880 | 0.865 |
> | Width    | 14.88 | 14.11 | 13.58 |
>
> [Experiment 3: Airfoil (Covariate Shift)]
> - Hyperparameter set-up: $\alpha=0.1, \lambda=1, T = 1, 127$
>
> - The following results are average results over 100 trials.
>
> [$\texttt{EXP3.P-CP}$]
> | $K$ | 20 | 40 | 60 |
> |----------|----------|----------|----------|
> | Coverage | 0.834 | 0.807 | 0.789 |
> | Width    | 14.62 | 13.75 | 13.21 |
>
> [$\texttt{EXP3.P-CP-UNLOCK}$]
> | $K$ | 20 | 40 | 60 |
> |----------|----------|----------|----------|
> | Coverage | 0.842 | 0.816 | 0.800 |
> | Width    | 14.91 | 14.05 | 13.55 |

---

> ### Author Response · Authors · 2025-11-18
> **Question 3 (Ablation Study on $\lambda$)**
>
> [Ablation Study on Robustness of Our Method on Different Values of $\lambda$]
>
> ***In short, experimental results empirically show that our methods are robust under different values of $\lambda$.***
>
> Specifically, we have conducted an ablation study in both experiments by varying $\lambda$, while controlling the other hyperparameters to be fixed.
>
> [Experiment 1: ImageNet (i.i.d.)]
>
> - Hyperparameter set-up: $\alpha=0.1, K=200, T = 50, 000$
>
> - The following results are average results over 50 trials.
>
> [$\texttt{EXP3.P-CP}$]
> | $\lambda$ | 0.1 | 0.5 | 1.0 | 2.0 | 5.0 | 10.0 |
> |----------|----------|----------|----------|----------|----------|----------|
> | Coverage | 0.857 | 0.856 | 0.855 | 0.853 | 0.850 | 0.850 |
> | Width    | 41.75 | 38.95 | 39.21 | 37.64 | 36.71 | 35.61 |
>
> [$\texttt{EXP3.P-CP-UNLOCK}$]
> | $\lambda$ | 0.1 | 0.5 | 1.0 | 2.0 | 5.0 | 10.0 |
> |----------|----------|----------|----------|----------|----------|----------|
> | Coverage | 0.898 | 0.897 | 0.896 | 0.894 | 0.893 | 0.892 |
> | Width    | 63.61 | 61.77 | 60.51 | 59.21 | 57.08 | 56.12 |
>
> [Experiment 2: Airfoil (i.i.d.)]
> - Hyperparameter set-up: $\alpha=0.1, K=20, T = 1, 127$
> - The following results are average results over 100 trials.
>
> [$\texttt{EXP3.P-CP}$]
> | $\lambda$ | 0.1 | 0.5 | 1.0 | 2.0 | 5.0 | 10.0 |
> |----------|----------|----------|----------|----------|----------|----------|
> | Coverage | 0.897 | 0.896 | 0.895 | 0.894 | 0.893 | 0.892 |
> | Width    | 14.51 | 14.44 | 14.41 | 14.36 | 14.29 | 14.28 |
>
> [$\texttt{EXP3.P-CP-UNLOCK}$]
> | $\lambda$ | 0.1 | 0.5 | 1.0 | 2.0 | 5.0 | 10.0 |
> |----------|----------|----------|----------|----------|----------|----------|
> | Coverage | 0.908 | 0.907 | 0.906 | 0.905 | 0.903 | 0.903 |
> | Width    | 15.12 | 15.04 | 14.98 | 14.90 | 14.78 | 14.73 |
>
> [Experiment 3: Airfoil (Covariate Shift)]
> - Hyperparameter set-up: $\alpha=0.1, K=20, T = 1, 127$
> - The following results are average results over 100 trials.
>
> [$\texttt{EXP3.P-CP}$]
> | $\lambda$ | 0.1 | 0.5 | 1.0 | 2.0 | 5.0 | 10.0 |
> |----------|----------|----------|----------|----------|----------|----------|
> | Coverage | 0.841 | 0.840 | 0.839 | 0.838 | 0.836 | 0.836 |
> | Width    | 14.85 | 14.76 | 14.73 | 14.67 | 14.61 | 14.60 |
>
> [$\texttt{EXP3.P-CP-UNLOCK}$]
> | $\lambda$ | 0.1 | 0.5 | 1.0 | 2.0 | 5.0 | 10.0 |
> |----------|----------|----------|----------|----------|----------|----------|
> | Coverage | 0.848 | 0.847 | 0.846 | 0.844 | 0.843 | 0.842 |
> | Width    | 15.13 | 15.07 | 15.03 | 15.01 | 14.90 | 14.86 |
>
> As the above ablation study shows, our method shows robust performance irrespective of the choice of $\lambda$, which in return has led us to set $\lambda = 1$ uniformly across all experiment set-ups.
>
> ***Nevertheless, we totally agree with your ample feedback on the absence of the discussion on the design choice of $\lambda$.***
>
> ***Moreover, we have further improved our $\texttt{EXP3.P-CP-UNLOCK}$ algorithm that provides users with $\lambda$ in a closed form as a function of the user-specified parameters $T$ (terminal time horizon) and $\alpha$ (target miscoverage rate), while enjoying the same theoretical guarantee.***
>
> We will update this modification in the updated version of our draft.

---

> > ### Comment · Reviewer_KKfP · 2025-11-26
> >
> > Thank you to the authors for their efforts in addressing my concerns. Most of the issues have been satisfactorily resolved. However, the sensitivity analysis indicates that, across a wide range of tuning parameters, the proposed method exhibits under-coverage. I am maintaining my original score, as I am not fully convinced that the paper has reached the acceptance threshold.

---

> ### Author Response · Authors · 2025-11-26
> **Clarification on the Sensitivity Analysis**
>
> We sincerely appreciate your valuable and insightful feedback on our additional experimental results.
>
> In brief, we would like to clarify that ***the under-coverage phenomenon observed in our sensitivity analysis does not arise from instability or inefficacy of our algorithm. Rather, it is primarily due to the relatively short time horizon ($T$) used in the experiments***.
>
> ---
>
> [1. Additional Experiments and Comments on the Sensitivity Analysis of $K$]
>
> To validate our claim, we first reproduced the same sensitivity analysis with respect to the discretization level ($K$) (**Experiment 2** and **Experiment 3**) using $\texttt{MVP}$, an online conformal prediction baseline with an access to the full feedback, which effectively serves as an oracle.
>
> [Experiment 2: Airfoil (i.i.d.)]
>
> - Hyperparameter set-up: $\alpha=0.1, \lambda=1, T = 1, 127$
>
> - The following results are average results over 100 trials.
>
> [$\texttt{MVP}$ (Full Feedback, Oracle)]
> | $K$ | 20 | 40 | 60 |
> |----------|----------|----------|----------|
> | Coverage | 0.890 | 0.888 | 0.883 |
> | Width    | 9.18 | 9.09 | 8.99 |
>
> [$\texttt{EXP3.P-CP-UNLOCK}$ (Partial Feedback, Ours)]
> | $K$ | 20 | 40 | 60 |
> |----------|----------|----------|----------|
> | Coverage | **0.904** | 0.880 | 0.865 |
> | Width    | 14.88 | 14.11 | 13.58 |
>
> [Experiment 3: Airfoil (Covariate Shift)]
>
> - Hyperparameter set-up: $\alpha=0.1, \lambda=1, T = 1, 127$
>
> - The following results are average results over 100 trials.
>
> [$\texttt{MVP}$ (Full Feedback, Oracle)]
> | $K$ | 20 | 40 | 60 |
> |----------|----------|----------|----------|
> | Coverage | 0.874 | 0.858 | 0.845 |
> | Width    | 11.86 | 11.63 | 11.41 |
>
> [$\texttt{EXP3.P-CP-UNLOCK}$ (Partial Feedback, Ours)]
> | $K$ | 20 | 40 | 60 |
> |----------|----------|----------|----------|
> | Coverage | 0.842 | 0.816 | 0.800 |
> | Width    | 14.91 | 14.05 | 13.55 |
>
> ***Even though $\texttt{MVP}$ assumes a full feedback setting, it exhibits under-coverage patterns those observed in our method ($\texttt{EXP3.P-CP-UNLOCK}$).***
>
> To further support our claim, we also repeated the sensitivity analysis while increasing the time-horizon ($T$).
>
> [Experiment 2: Airfoil (i.i.d.)]
>
> - Hyperparameter set-up: $\alpha=0.1, \lambda=1$
>
> - The following results are average results over 100 trials.
>
> [$\texttt{EXP3.P-CP-UNLOCK}$ ($T=1,127$)]
> | $K$ | 20 | 40 | 60 |
> |----------|----------|----------|----------|
> | Coverage | **0.904** | 0.880 | 0.865 |
> | Width    | 14.88 | 14.11 | 13.58 |
>
> [$\texttt{EXP3.P-CP-UNLOCK}$ ($T=3,381$)]
> | $K$ | 20 | 40 | 60 |
> |----------|----------|----------|----------|
> | Coverage | **0.940** | **0.925** | **0.914** |
> | Width    | 16.27 | 15.54 | 15.14 |
>
> [Experiment 3: Airfoil (Covariate Shift)]
>
> - Hyperparameter set-up: $\alpha=0.1, \lambda=1$
>
> - The following results are average results over 100 trials.
>
> [$\texttt{EXP3.P-CP-UNLOCK}$ ($T=1,127$)]
> | $K$ | 20 | 40 | 60 |
> |----------|----------|----------|----------|
> | Coverage | 0.842 | 0.816 | 0.800 |
> | Width    | 14.91 | 14.05 | 13.55 |
>
> [$\texttt{EXP3.P-CP-UNLOCK}$ ($T=11,270$)]
> | $K$ | 20 | 40 | 60 |
> |----------|----------|----------|----------|
> | Coverage | **0.925** | **0.910** | **0.900** |
> | Width    | 17.61 | 16.87 | 16.45 |
>
> As expected, the under-coverage patterns disappear across candidate values of $K$, confirming that the short time-horizon ($T$) is the cause of the under-coverage phenomenon.
>
> This phenomenon is aligned with our theoretical guarantee on the miscoverage rate ($MC_T$).
>
> Specifically, we have shown that the gap between $MC_T$ and the target miscoverage level ($\alpha$), *i.e.*, $MC_T - \alpha$, is bounded by $\mathcal{O}\sqrt{K \text{ln}K T^{-1}}$ ($\because$ **Lemma 1** and **Theorem 1**).
>
> ***In other words, once the number of time steps is sufficiently large ($T \gg K \text{ln}K$), under-coverage is unlikely to occur.***
>
> Therefore, as a practical guideline, a large $K$ is preferable when dealing with long data streams ($\uparrow T$), as it yields more compact conformal sets.
>
> In contrast, while a smaller $K$ may be preferable in short-horizon scenarios ($\downarrow T$), there is by no means to prevent the under-coverage phenomenon when $T$ is too small, which is also observed in baselines under the full feedback scenario.
>
> As a final remark, we want to additionally note that longer time-horizon ($\uparrow T$) is required to reach the target coverage under the non-i.i.d. setting, when compared to the i.i.d. scenario.

---

> ### Author Response · Authors · 2025-11-26
> **Clarification on the Sensitivity Analysis (cont.)**
>
> [2. Additional Experiments and Comments on the Sensitivity Analysis of $\lambda$]
>
> As shown in our initial sensitivity analysis on $\lambda$,
> the experimental results demonstrate that ***our method is robust across different values of $\lambda$***.
>
> ***Regarding the under-coverage issue, increasing the time horizon ($\uparrow T$) resolves it in the same manner as in the sensitivity analysis on $K$.***
>
> [Experiment 2: Airfoil (i.i.d.)]
>
> - Hyperparameter set-up: $\alpha=0.1, K=20$
>
> - The following results are average results over 100 trials.
>
> [$\texttt{EXP3.P-CP-UNLOCK}$ ($T=1,127$)]
> | $\lambda$ | 0.1 | 0.5 | 1.0 | 2.0 | 5.0 | 10.0 |
> |----------|----------|----------|----------|----------|----------|----------|
> | Coverage | 0.897 | 0.896 | 0.895 | 0.894 | 0.893 | 0.892 |
> | Width    | 14.50 | 14.44 | 14.41 | 14.36 | 14.28 | 14.28 |
>
> [$\texttt{EXP3.P-CP-UNLOCK}$ ($T=3,381$)]
> | $\lambda$ | 0.1 | 0.5 | 1.0 | 2.0 | 5.0 | 10.0 |
> |----------|----------|----------|----------|----------|----------|----------|
> | Coverage | **0.942** | **0.941** | **0.940** | **0.940** | **0.939** | **0.938** |
> | Width    | 16.43 | 16.34 | 16.27 | 16.21 | 15.99 | 15.95 |
>
> [Experiment 3: Airfoil (Covariate Shift)]
>
> - Hyperparameter set-up: $\alpha=0.1, K=20$
>
> - The following results are average results over 100 trials.
>
> [$\texttt{EXP3.P-CP-UNLOCK}$ ($T=1,127$)]
> | $\lambda$ | 0.1 | 0.5 | 1.0 | 2.0 | 5.0 | 10.0 |
> |----------|----------|----------|----------|----------|----------|----------|
> | Coverage | 0.841 | 0.840 | 0.839 | 0.838 | 0.836 | 0.836 |
> | Width    | 14.85 | 14.76 | 14.73 | 14.67 | 14.61 | 14.60 |
>
> [$\texttt{EXP3.P-CP-UNLOCK}$ ($T=11,270$)]
> | $\lambda$ | 0.1 | 0.5 | 1.0 | 2.0 | 5.0 | 10.0 |
> |----------|----------|----------|----------|----------|----------|----------|
> | Coverage | **0.927** | **0.926** | **0.925** | **0.924** | **0.923** | **0.922** |
> | Width    | 17.81 | 17.69 | 17.61 | 17.46 | 17.34 | 17.28 |
>
> ---
>
> [3. Conclusion]
>
> Finally, we would like to emphasize that ***our initial sensitivity analyses intentionally fixed the time horizon ($T$) while varying only the hyperparamers. This design allowed us to isolate and assess the individual effects of each hyperparameter under controlled conditions***.
>
> In particular, while we have expected in advance from the theoretical result that longer time-horizon ($\uparrow T$) is required to prevent from undercoverage for finer discretization of the hypothesis space ($\uparrow K$), our primary objective in the initial analyses was to study the effect of varying $K$ while keeping others factors ($\alpha, \lambda, T$) fixed, not to maintain performance across different $K$ values.
>
> We couldn't report additional experiments under the **Experiment 1** setting due to the computation resource issue.
>
> While it may take some time, we will upload the result upon your request.
>
> ---
>
> We once again thank you for your thoughtful and invaluable feedback, and we look forward to your response.

---

### Author Response · Authors · 2025-12-02
**Summary of Our Paper and Rebuttal Process**

Dear **Reviewers**, **Area Chairs**, **Senior Area Chairs**, and **Program Chairs**,

We sincerely appreciate all three reviewers for their constructive comments.

Along with ***our work's strengths, including practicality, theoretical guarantee, high modularity, and clear paper writing that reviewers acknowledged***, we are encouraged that ***the reviewers' comments have greatly helped us further refine and improve our paper***.


We respectfully submit ***(1) a concise summary of our paper*** and ***(2) an overview of the reviewers' initial comments along with our rebuttal responses***.

---

## **[1. Summary of Our Paper]**

### **[Problem]**

We propose a method for online conformal prediction with adversarial partial feedback, where a true label is revealed only when it lies within a chosen conformal set.

### **[Main Contribution]**

While existing works on online conformal prediction consider either (1) adversarial full feedback or (2) stochastic (i.i.d.) partial feedback settings, our work is the first principled method that can jointly encompass both settings.

### **[Method]**

We formulate our problem as a multi-armed adversarial bandit problem by treating each conformal set candidate as an arm. As such, we leverage the $\texttt{EXP3.P}$ algorithm, an algorithm that provides a sublinear regret under adversarial bandit environments.

### **[Theoretical Guarantee]**

By designing a loss function tailored to conformal prediction, we provide an explicit learner-agnostic relationship between a regret from a learner and its miscoverage rate.

This relationship ensures the long-run coverage to achieve the target level $1 - \alpha$ for any learner that achieves a sublinear regret.

### **[Technical Contribution]**

We further improve the algorithm by fully exploiting (1) information available in the partial feedback scenario and (2) the monotonicity property of a threshold-parameterized conformal set with respect to the miscoverage.

### **[Empirical Results]**

We empirically show that our method approaches long-run coverage while maintaining a moderate average conformal set size in both i.i.d. and non-i.i.d. settings, achieving performance comparable to online conformal prediction baselines with adversarial full feedback.

---

## **[2. Reviewer's Assessments]**

First of all, we are particularly encouraged by ***the reviewer's recognition of our paper's strengths*** as the following.

### **Strength 1: Significance of the Problem and Practicality**
- **Reviewer dv5m**: The partial feedback setting in online conformal prediction is ***highly relevant***, offering ***practical value*** to both the conformal prediction research community and real-world practitioners.
- **Reviewer 7zrS**: ... the modular design lets other bandit learners plug in, ***increasing impact in retrieval, selective prediction, and human-in-the-loop systems***.

### **Strength 2: Theoretical Guarantee**
- **Reviewer KKfP**: Establishes a ***clear connection between regret minimization and coverage guarantees*** in conformal prediction.
- **Reviewer 7zrS**: It offers high probability guarantees and a ***clear analysis pipeline from loss design to regret bounds to miscoverage guarantees***, ...

### **Strength 3: High Modularity**
- **Reviewer dv5m**: The paper’s connection to regret minimization methods is particularly compelling, as it ***unlocks a rich toolbox of established techniques*** that can be effectively applied to conformal prediction.
- **Reviewer 7zrS**: ... and ***the modular design lets other bandit learners plug in***, ...

### **Strength 4: Clear Paper Writing with Detailed Proofs**
- **Reviewer KKfP**: Provides regret and miscoverage bounds with detailed proofs.
- **Reviewer d5vm**: The paper is generally clear and well-structured, making it easy to follow. The proofs are also well detailed and, I believe, sound.
- **Reviewer 7zrS**: The writing is clear, ...

---

> ### Author Response · Authors · 2025-12-02
> **Summary of Our Paper and Rebuttal Process (cont.)**
>
> ***Moreover, during the rebuttal phase, we would like to demonstrate that we have thoroughly addressed all the concerns from the reviewers as summarized below.***
>
> We would like to emphasize that one of the reviewers raised their score from 6 to 8 in the early rebuttal phase and that we have made additional experiments accompanied by detailed explanations on the comment asking for further clarifications on our initial response (**Comment 6**).
>
> Here, **W** stands for the **Weakness**, and **Q** for the **Question**.
>
> ---
>
> First, we summarize ***comments that were raised by multiple reviewers*** as the following.
>
> ### **Comment 1: Additional Literature Review** (**Reviewer KKfp**: **W1**, **W5**; **Reviewer 7ZrS**: **Q3**)
>
> => **Response**: We have summarized existing works on online conformal prediction with adversarial full feedback or stochastic (i.i.d.) partial feedback, thereby highlighting that our work is the first to jointly address both settings.
>
> ### **Comment 2: Clarifications on the Contents of Our Paper** (**Reviewer KKfp**: **W2**, **W3**, **W4**; **Reviewer d5vm**: **Q1**, **Q3**; **Reviewer 7ZrS**: **Q5**, **Q6**, **Q7**, **Q8**, **Q9**)
>
> => **Response**: We have clarified the contents of our paper, specifically on the notation, proposed method, theoretical guarantee, and experimental results.
>
> ### **Comment 3: Typos** (**Reviewer KKfp**: **Q1**; **Reviewer d5vm**: **W2**, **Reviewer 7ZrS**: **Q4**, **Q10**, **Q11**)
>
> => **Response**: We have ensured that all the typos will be removed in our final draft for better clarity.
>
> ### **Comment 4: Comparison with Additional Baselines Under Partial Feedback Setting** (**Reviewer KKfp**: **Q2**; **Reviewer d5vm**: **W1**)
>
> => **Response**: We have run $\texttt{SPS}$ algorithm [1], the only method that tackles online conformal prediction under partial feedback setting. We could not run experiments at the time of submission due to the absence of a public code repository. While $\texttt{SPS}$ struggles to achieve the target coverage under the distribution shift setting, our method ($\texttt{EXP3.P-CP-UNLOCK}$) achieves reasonably good empirical coverage.
>
> ***Additional details and experimental results can be found in the following [anonymized link](https://anonymous.4open.science/api/repo/OCP-PDFs-37CE/file/comment4.pdf?v=16c16599).***
>
> ### **Comment 5: Ablation Study on Hyperparameters** (**Reviewer KKfp**: **Q3**, **Reviewer d5vm**: **Q2**, **Reviewer 7ZrS**: **Q1**)
>
> => **Response**: We have conducted the ablation study on the discretization level ($K$) and $\lambda$. Regarding $K$, we have shown that there is a trade-off between the size of the conformal set and the regret bound, which aligns with our theoretical results. Besides, the results are robust to the choice of $\lambda$. Nevertheless, we have further improved our $\texttt{EXP3.P-CP-UNLOCK}$ algorithm that provides $\lambda$ in a closed form to users as a function of the user-specified parameters $T$ (total time horizon) and $\alpha$ (target miscoverage rate), while enjoying the same theoretical guarantee.
>
> ***Additional details and experimental results can be found in the following [anonymized link](https://anonymous.4open.science/api/repo/OCP-PDFs-37CE/file/comment5.pdf?v=061524a3).***
>
> [1] [Ge et al., ICML, 2025] Stochastic Online Conformal Prediction with Semi-bandit Feedback

---

> ### Author Response · Authors · 2025-12-02
> **Summary of Our Paper and Rebuttal Process (cont.)**
>
> ***The remaining comments were raised by a single reviewer as the following.***
>
> ### **Comment 6: Additional comment on the under-coverage phenomenon in ablation study on hyperparameters** (**Reviewer KKfP**)
>
> => **Response**: We have shown through extensive additional experiments that the under-coverage phenomenon observed in our sensitivity analysis does not stem from instability or ineffectiveness of our algorithm. Rather, it is primarily due to the relatively short time horizon ($T$) used in the experiments. Indeed, our oracle baseline $\texttt{MVP}$, an online conformal prediction method that exploits full feedback, also suffers from under-coverage phenomenon for the same $T$, and the under-coverage phenomenon is removed as we increase $T$. More importantly, our initial sensitivity analyses intentionally fixed the time horizon ($T$) while varying only the hyperparameters. This design allowed us to isolate and assess the individual effects of each hyperparameter under controlled conditions. Finally, guidelines on the choice of $K$ are provided, which additionally supplements **Comment 5** above.
>
> ***Additional details and experimental results can be found in the following [anonymized link](https://anonymous.4open.science/api/repo/OCP-PDFs-37CE/file/comment6.pdf?v=464ac159).***
>
> ### **Comment 7: Additional experiments under low feedback rates** (****Reviewer 7ZrS**: Q2**)
>
> => **Response**: While we cannot pre-specify the rate of semi-bandit feedback to be low (which is clarified in detail in our response), we conducted experiments on the same dataset (ImageNet) under different data streaming set-ups by varying the levels of difficulty. The results further emphasized the importance of fully exploiting the information under the semi-bandit feedback setting, which is achieved by incorporating unlocking mechanisms into our algorithm.
>
> ***Additional details and experimental results can be found in the following [anonymized link](https://anonymous.4open.science/api/repo/OCP-PDFs-37CE/file/comment7.pdf?v=b2342be2).***
>
> ---
>
> We believe that our work represents ***a meaningful step toward building practical online conformal prediction methods in real-world settings***.
>
> ***We have also revised the paper to enhance its clarity and presentation in response to the reviewers’ feedback. Major modifications are highlighted in blue in the revised manuscript.***
>
> We once again deeply appreciate all the reviewers for their time and for overseeing the review process.
>
> Sincerely,
>
> The Authors

---

### Meta-Review · Area_Chair_SyWp · 2026-01-07

**Summary:**

The main concerns raised by the reviewers include:
1. Insufficient review of related work on online conformal prediction (KKfP)
2. Insufficient clarification of differences from SPS (Ge et al., 2025) (KKfP)
3. Limited set of experimental baselines (d5vm)
4. Unclear effect of hyperparameter settings (KKfP, d5vm, 7ZrS)

**Reviewer Concerns:**

The authors addressed each of these concerns during the rebuttal phase.
1. The authors revised the paper to include additional discussion of related work suggested by reviewers.
2. The authors clarified the distinction between the proposed method and SPS (Ge et al., 2025). In particular, the proposed method handles arbitrary distribution shifts under partial feedback.
3. The authors added SPS as an experimental baseline to provide a direct comparison.
4. The authors added ablations that explore the affect of hyperparameters $\lambda$ and $K$ on the results.

**Reviewer Scores:**

- KKfP is very likely to keep their score. Their initial review was positive and the author rebuttal addressed remaining concerns. In particular, the authors added discussion about key related work, clarified the distinction between the proposed method and SPS, and added SPS as an experimental baseline. The authors also provided ablations demonstrating the effect of $\lambda$ and $K$ on the results.
- d5vm is likely to increase their score. To address the concerns about experiments, the authors added SPS (Ge et al., 2025) as a baseline. Reviewer d5vm also requested more information about how the set of candidate thresholds were chosen. The authors provided ablations demonstrating how discretization of the hypothesis space affects results.
- 7ZrS is very likely to increase their score. The author rebuttal addressed sensitivity to hyperparameters by adding an ablation study for $\lambda$. They also sufficiently addressed remaining reviewer concerns about performance under low feedback and the motivation for the bandit framing.

---

### Decision · Program_Chairs · 2026-01-26

Accept (Poster)